# Human hippocampal CA3 damage disrupts both recent and remote episodic memories

Thomas D Miller[1,2], Trevor T-J Chong[3], Anne M Aimola Davies[4,5],
Michael R Johnson[6], Sarosh R Irani[1], Masud Husain[1,4], Tammy WC Ng[7],
Saiju Jacob[8], Paul Maddison[9], Christopher Kennard[1], Penny A Gowland[10]*,
Clive R Rosenthal[1]*

[1]Nuffield Department of Clinical Neurosciences, University of Oxford, Oxford, United Kingdom; [2]Department of Neurology, Royal Free Hospital, London, United Kingdom; [3]Monash Institute of Cognitive and Clinical Neurosciences, Monash University, Clayton, Australia; [4]Department of Experimental Psychology, University of Oxford, Oxford, United Kingdom; [5]Research School of Psychology, Australian National University, Canberra, Australia; [6]Division of Brain Sciences, Imperial College London, London, United Kingdom; [7]Department of Anaesthestics, Royal Free Hospital, London, United Kingdom; [8]Neurology Department, Queen Elizabeth Neuroscience Centre, University Hospitals of Birmingham, Birmingham, United Kingdom; [9]Neurology Department, Queen's Medical Centre, Nottingham, United Kingdom; [10]Sir Peter Mansfield Magnetic Resonance Centre, School of Physics and Astronomy, University of Nottingham, Nottingham, United Kingdom

*For correspondence:
penny.gowland@nottingham.ac.
uk (PAG);
clive.rosenthal@clneuro.ox.ac.uk
(CRR)

Competing interest: See
page 31

Reviewing editor: Morgan
Barense, University of Toronto,
Canada

**Abstract** Neocortical-hippocampal interactions
support new episodic (event) memories, but there is conflicting evidence about the dependence of remote episodic memories on the hippocampus. In line with systems consolidation and computational theories of episodic memory, evidence from model organisms suggests that the cornu ammonis 3 (CA3) hippocampal subfield supports recent, but not remote, episodic retrieval. In this study, we demonstrated that recent and remote memories were susceptible to a loss of episodic detail in human participants with focal bilateral damage to CA3. Graph theoretic analyses of 7.0-Tesla resting-state fMRI data revealed that CA3 damage disrupted functional integration across the medial temporal lobe (MTL) subsystem of the default network. The loss of functional integration in MTL subsystem regions was predictive of autobiographical episodic retrieval performance. We conclude that human CA3 is necessary for the retrieval of episodic memories long after their initial acquisition and functional integration of the default network is important for autobiographical episodic memory performance.

## Introduction

Neurobiological theories of episodic (i.e., event) memory differ in the proposed duration of hippocampal support for episodic retrieval. In standard systems consolidation and computational-based theories of episodic memory, new hippocampal-dependent episodic memories become reorganized into a distributed neocortical network, such that remote memories are no longer dependent on the hippocampus (*Bayley et al., 2005*; *Bontempi et al., 1999*; *Dudai and Morris, 2013*; *Kim and Fanselow, 1992*; *Kirwan et al., 2008*; *McClelland et al., 1995*; *Squire and Bayley, 2007*; *Takashima et al., 2009*). By contrast, according to multiple trace theory, the transformation

hypothesis, contextual binding theory, and scene construction theory, episodic memory is hypothesized to be continuously dependent on the hippocampus for as long as the memory retains spatial detail and context-specific episodic content (*Barry and Maguire, 2019*; *Maguire and Mullally, 2013*; *Moscovitch et al., 2016*; *Moscovitch et al., 2005*; *Winocur et al., 2010*; *Yonelinas et al., 2019*).

Recent work has centered on understanding the differing anatomical connectivity, firing properties, and functional contribution of hippocampal subfields (the cornu Ammonis [CA] CA1–3, dentate gyrus, and subiculum) (*Dalton et al., 2019*; *Dalton et al., 2018*; *Kesner and Rolls, 2015*; *Rebola et al., 2017*). Little is known, however, about how long each human hippocampal subfield remains necessary for episodic retrieval. Lesion and molecular imaging studies in rodents indicate that CA1 and CA3 enable the rapid storage and retrieval of recent (<1 month) contextual fear memories, whereas remote memories depend on CA1 but not CA3 (*Denny et al., 2014*; *Guzman et al., 2016*; *Kesner and Rolls, 2015*; *Leutgeb et al., 2007*; *Lisman, 1999*; *Lux et al., 2016*; *McNaughton and Morris, 1987*; *Rebola et al., 2017*). In humans, damage to CA1, associated with transient global amnesia (lasting, 8.3 ± 1.9 hr), impairs recent and remote autobiographical episodic retrieval (*Bartsch et al., 2011*). These results are consistent with the role of CA1 as the primary output node from the hippocampus to the neocortex (*Witter and Amaral, 2004*). In a recent study of a single case involving focal damage to the human dentate gyrus and a portion of CA3, secondary to hypoxic-ischaemic brain injury, deficits were also found in recent and remote memories (*Baker et al., 2016*; *Kwan et al., 2015*).

To our knowledge, it has not yet been possible to study the causal role of human CA3 in episodic memory retrieval over extended retention intervals at a group level, because the anatomical specificity of damage involving the hippocampus is seldom restricted to a single subfield. If human hippocampal subfields do not share a common duration of involvement in episodic memory retrieval, as suggested by the evidence from model organisms (*Denny et al., 2014*; *Guzman et al., 2016*; *Kesner and Rolls, 2015*; *Leutgeb et al., 2007*; *Lisman, 1999*; *Lux et al., 2016*; *McNaughton and Morris, 1987*; *Rebola et al., 2017*), then investigating the duration of involvement as a function of each human subfield may help to resolve the divergence in experimental studies and theoretical accounts that are based on averaging across the hippocampus.

Here, we tested the contribution of human hippocampal area CA3 to the retrieval of recent and remote autobiographical memories by assessing 16 human participants (age: 64.2 ± 4.81 years [mean ± s.e.m.], female = 3) with hippocampal damage (bilateral volume loss confined to CA3, mean reduction = -29%, *Figures 1* and *2*, *Table 1*, and *Figure 2—figure supplement 1*), secondary to a rare, single aetiology, leucine-rich glycine-inactivate-1 antibody-complex limbic encephalitis (LGI1-antibody-complex LE) (*Dalmau and Rosenfeld, 2014*; *Irani et al., 2013*; *Irani et al., 2010*; *Miller et al., 2017*). The amnesic group was compared against 16 control participants (62.3 ± 3.23 years, female = 6). In prior 3.0-Tesla and 7.0-Tesla MRI based studies, we showed that anatomical damage associated with the chronic phase of the LGI1-antibody-complex LE phenotype did not lead to gray matter volume loss outside of the hippocampus (*McCormick et al., 2016*; *McCormick et al., 2017*; *McCormick et al., 2018b*; *Miller et al., 2017*) (*Figure 3*). Such evidence of focal hippocampal damage aligns with the results from other laboratories on the anatomical sequalae of LGI1-antibody-complex LE (*Finke et al., 2017*; *Wagner et al., 2015b*; *Wagner et al., 2015a*).

In order to assess episodic memory retrieval, recent and remote autobiographical memories for personal events were interrogated using an objective, parametric, text-based method, the Autobiographical Interview (AI) (*Levine et al., 2002*). Autobiographical memories were sampled across five intervals, covering recent memories post CA3 damage and remote memories up to ~60 years prior to the CA3 damage. Responses were scored to obtain quantitative measures of internal detail (i.e., re-experiencing a remembered event acquired in a discrete context) and external detail (i.e., non-episodic, general 'semantic' facts about a remembered event and evaluative comments) associated with each interval. Evidence of impaired recent (~1 year) autobiographical episodic but intact semantic memory following CA3 damage is consistent with most accounts of the hippocampal role in episodic memory consolidation (*Miller et al., 2017*). Here, we hypothesized that CA3 damage would lead to a loss of internal details for both recent and remote memories, because there is mounting evidence from functional neuroimaging to indicate that human CA3 is engaged in the retrieval of both recent and remote vivid episodic memories (*Bonnici et al., 2013*; *Chadwick et al., 2014*). By contrast, if CA3 is not required for remembering remote memories, then the amnesic group would be expected to exhibit deficits in recent but not remote memories (i.e., individuals with amnesia would exhibit the

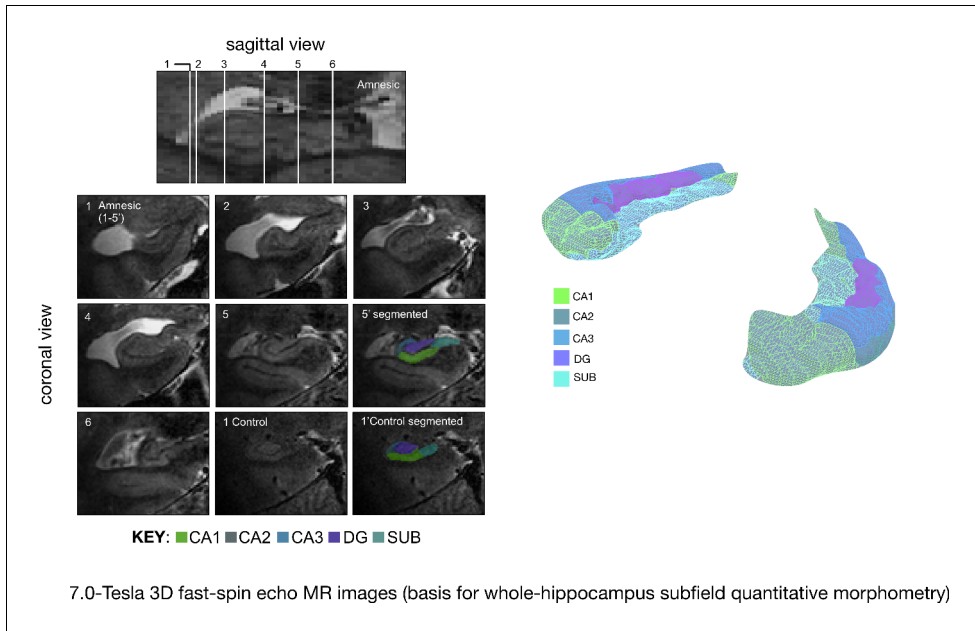

**Figure 1.** Quantitative three-dimensional whole-hippocampal manual volumetry of five hippocampal subfields (CA1, CA2, CA3, the dentate gyrus, and the subiculum). Left panel: native coronal images from whole-hippocampal 7.0-Tesla 3-D fast-spin echo sequence (0.39 × 0.39 × 1.0 mm$^3$ spatial resolution). Quantitative three-dimensional whole-hippocampal manual volumetry of five hippocampal subfields (CA1, CA2, CA3, the dentate gyrus, and the subiculum) was conducted along the full longitudinal axis of participants in the amnesic group (N = 15) and control group (N = 15). Colored shading on the coronal images provides examples from applying the manual hippocampal subfield segmentation protocol in a participant at the chronic phase of the LGI1-antibody-complex LE phenotype (5') and in a control (1'). Each of the white lines (1–6) on the sagittal view of the hippocampus corresponds to six example coronal locations along the anterior–posterior axis. 7.0-Tesla 3-D fast-spin echo scans and results from manual volumetry were reported in our previous study on 18 participants assessing the chronic phase of the LGI1-antibody-complex LE phenotype (**Miller et al., 2017**). Right panel: example 3-D rendering (Paraview v4.10; www.paraview.org) obtained from the output of hippocampal subfield segmentation generated for a participant from the amnesic group using ITK-SNAP v3.2 (http://www.itksnap.org). The color key under the 3-D FSE coronal images and color key 3-D render corresponds to CA1, CA2, CA3, DG (dentate gyrus), and SUB (subiculum) hippocampal subfields. Adapted from **Miller et al. (2017)**, published under CC BY license, http://creativecommons.org/licenses/by/4.0/.

phenomenon of temporally graded retrograde amnesia). Such a pattern of loss would align with a systems consolidation based interpretation, whereby recent memories are more vulnerable to hippocampal damage than older, remote memories that have been reorganized into a form that is supported by the neocortex (**Bayley et al., 2005**; **Bontempi et al., 1999**; **Dudai and Morris, 2013**; **Kim and Fanselow, 1992**; **Kirwan et al., 2008**; **Squire and Bayley, 2007**; **Takashima et al., 2009**).

CA3 damage was also hypothesized to affect the functional connectivity of regions that have been implicated in autobiographical episodic memory, because the hippocampus acts as a major 'hub', linking different subnetworks that are involved in memory retrieval. Lesions involving hubs such as the hippocampus or other regions within the medial temporal lobe (MTL) lead to disruption in the functional connectivity of non-local brain regions (**Backus et al., 2016**; **Crossley et al., 2014**; **Mišić et al., 2014**). These effects are consistent with models of connectomic diaschisis, whereby functional changes are hypothesized to occur in areas that are not directly linked to a damaged area (**Carrera and Tononi, 2014**). Less is known about how damage to single subfields of the human hippocampus affects the functional connectivity of regions associated with large-scale resting brain networks. Evidence from the inhibition of rodent CA1 indicates that selective compensatory changes can occur in anterior cingulate cortex activity (**Goshen et al., 2011**). CA3 is not considered a main output structure, because it receives sparse, orthogonalized input via the mossy fibers from the DG, but nonetheless has outputs via the fimbria that can bypass CA1 (**van Strien et al., 2009**; **Witter and Amaral, 2004**), which suggests that the effects of CA3 damage are likely to involve

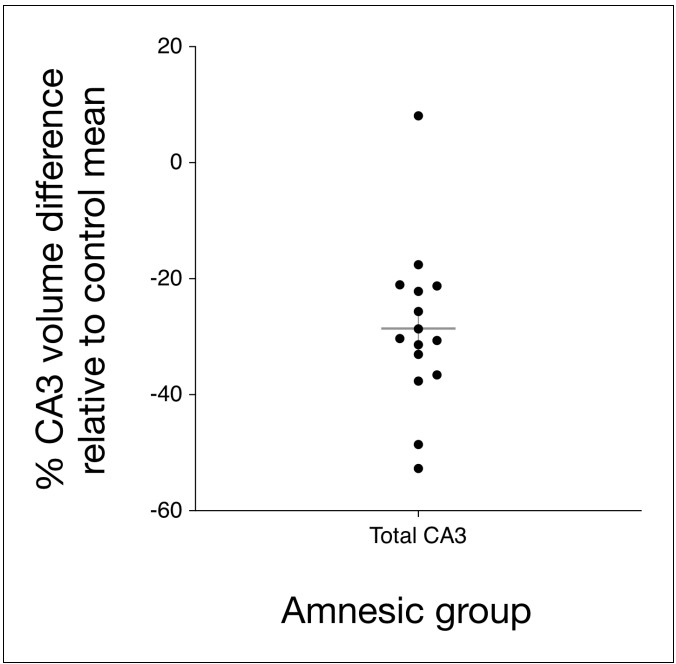

**Figure 2.** Bilateral hippocampal CA3 volume loss in the amnesic group. The graph depicts the reduction in CA3 subfield volume (CA3 subfield volume was corrected for total intracranial volume) relative to the control group mean. Error bars correspond to the s.e.m. and the horizontal line corresponds to the mean. A three-way mixed-model ANOVA, with two within-subjects factors (subfield and side) and one between-subjects factor (group), was used to test for differences in hippocampal subfield volumes between the amnesic (N = 15) and control (N = 15) groups. The assumption of sphericity was violated for subfield ($\chi^2_{(9)}$ = 52.46, p<0.0001) and for the interaction between subfield and side ($\chi^2_{(9)}$ = 63.48, p<0.0001), so degrees of freedom were corrected using Greenhouse-Geisser correction ($\varepsilon$ = 0.551). Significant two-way interactions were found between group and subfield ($F_{(5.30,61.74)}$ = 5.30, p=0.006), and between side and subfield ($F_{(2.02,56.55)}$=14.15, p<0.0001), but not between group and side ($F_{(1,28)}$ = 1.25, p=0.272). The three-way interaction was not significant ($F_{(2.02,56.55)}$ = 0.43, p=0.66). Subfield volumes were collapsed across left and right due to the absence of the significant three-way interaction. Significant bilateral CA3 volume loss was seen in the amnesic group relative to the control group ($F_{(1,28)}$ = 14.52, p=0.001, Cohen's *d* = 1.39; mean reduction = –29%), whereas the differences in CA1, CA2, subiculum, and dentate gyrus volumes were not statistically significant at the Holm-Bonferroni alpha criterion corrected for multiple comparisons. Mean normalized total volumes for all segmented subfields are reported in *Table 1* and subfield volumes for individual participants are plotted in *Figure 2—figure supplement 1*. 7.0-Tesla 3-D fast-spin echo scans and results from manual volumetry are a subgroup of those reported in our previous clinical study involving 18 participants at the chronic phase of the LGI1-antibody-complex LE phenotype (*Miller et al., 2017*).

The online version of this article includes the following figure supplement(s) for figure 2:

**Figure supplement 1.** Plot of mean (normalized) total hippocampal subfield volumes (mm³) for participants in the amnesic group and control group.

changes in the functional connectivity of non-local brain regions. Therefore, to examine how damage to human CA3 disrupts functional connectivity, we acquired functional MRI data at ultra-high field strength (7.0-Tesla) from the resting (i.e., task-free) brain (rs-fMRI) of participants in the amnesic group and the control group.

We applied graph theoretic analyses of the rs-fMRI data to investigate the scalar extent and topological properties of interconnected network nodes (brain regions-of-interest) affected by the CA3 damage (*Bressler and Menon, 2010*; *Fornito et al., 2015*; *van den Heuvel and Sporns, 2013*). Nodes and their pairwise ties or edges (the quantification of functional connectivity) were focused on the default network (DN) (*Buckner et al., 2008*; *Buckner and DiNicola, 2019*; *Raichle et al., 2001*), because the DN is associated with episodic retrieval (*Raichle, 2015*; *Spreng and Grady, 2010*). The DN also overlaps with a network of regions associated with autobiographical memory; namely, parietal regions such as posterior cingulate cortex/retrosplenial cortex, the medial prefrontal cortex, the MTL,

**Table 1.** Hippocampal subfield volumes (means (mm³)), ± standard error of the mean (SEM), standard deviation (SD)) in the amnesic group and control group.

Volumes were normalized to the total intracranial volumes obtained from the VBM analyses. Volumes were collapsed across the left and right hippocampi because there was no significant interaction term between group (amnesic, control), side (left, right), and subfield (CA1, CA2, CA3, DG, and SUB) ($F_{(2.02,56.55)}$ = 0.43, p=0.66, $\eta^2_p$=0.015; *Figure 2*). See *Figure 2—figure supplement 1* for hippocampal subfield volumes for individual participants.

| Hippocampal subfield | Mean total subfield volumes (mm³), SEM and SD | |
| --- | --- | --- |
| | Amnesic (LGI1-complex-antibody LE) group | Control group |
| CA1 | 961 (±63, 243) | 1149 (±41, 157) |
| CA2 | 169 (±9, 37) | 179 (±10, 40) |
| CA3* | 377 (±19, 75) | 528 (±34,134) |
| DG | 625 (±50,194) | 659 (±23,95) |
| SUB | 526 (±34,131) | 611 (±25,95) |

*Significant at the alpha criterion based on Holm-Bonferroni correction for multiple comparisons, following mixed model ANOVA. All other subfields were non-significant, when assessed at the alpha criterion corrected for multiple comparisons. CA1, cornu ammonis 1; CA2, cornu ammonis 2; CA3, cornu ammonis 3; DG, dentate gyrus; SUB, subiculum. Total intracranial volume (TIV) was derived by applying the sequence of unified segmentation, as implemented in SPM12, to the whole-brain T₁-weighted images that were also acquired from each participant.

and the lateral temporal cortex (*Andrews-Hanna et al., 2010*; *Andrews-Hanna et al., 2014*; *Buckner and Carroll, 2007*; *Cabeza and St Jacques, 2007*; *Greenberg et al., 2005b*; *Schacter et al., 2012*; *Spreng et al., 2009*; *Svoboda et al., 2006*).

Applications of graph theoretic analyses and clustering analysis techniques to rs-fMRI have revealed that the DN can be fractionated into a midline core (posterior cingulate cortex and anterior medial prefrontal cortex [amPFC]), a MTL subsystem (ventral medial prefrontal cortex, posterior inferior parietal lobule, retrosplenial cortex, parahippocampal cortex, and hippocampal formation), and a dorsomedial prefrontal cortex (dmPFC) subsystem (dmPFC, temporo-parietal junction, lateral temporal cortex, and temporal pole) (*Andrews-Hanna et al., 2010*). Here, these DN components were examined by computing graph theoretic measures of functional integration (average path length and global efficiency), functional segregation (clustering coefficient and local efficiency), and local measures of node centrality (degree and betweenness centrality). Functional integration examines the capacity of nodes to combine information from distributed regions, whereas functional segregation is a proxy for the capacity to support specialized processing within densely interconnected groups of regions (*van den Heuvel and Sporns, 2013*). Perturbations of DN topology were hypothesized to be largely confined to the MTL subsystem given its a priori association with episodic remembering, the evidence in our prior work showing that deficits associated with the chronic phase of the LGI1-antibody-complex LE phenotype on standardized neuropsychological assessment did not extend beyond tests of memory (*Miller et al., 2017*), and evidence that simulated damage and anatomical lesions involving hub regions can have non-extensive effects that do not necessarily affect whole-brain network organization (*Gratton et al., 2012*; *He et al., 2009*; *Honey and Sporns, 2008*).

In summary, we tested the prediction that human CA3 is necessary for remote as well as recent autobiographical episodic memory, and hypothesized that the effects of CA3 damage on neurobiologically meaningful network properties would be expressed in the topological properties of nodes that comprise the MTL subsystem of the DN. Importantly, in order to understand the relevance of anatomical damage and alterations in functional connectivity for behavior, we also examined whether CA3 volume and between-group differences in the topological properties of affected DN nodes were predictive of autobiographical episodic memory performance on the AI.

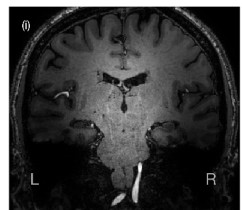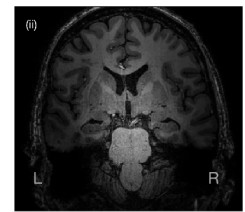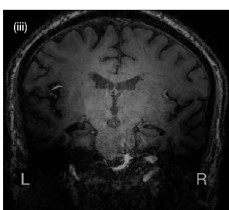

7.0-Tesla T$_1$-weighted anatomical images used as the basis for whole-brain voxel-based morphometry

**Figure 3.** 7.0-Tesla whole-brain anatomical magnetic resonance imaging. Whole-brain 7.0-Tesla T$_1$-weighted anatomical (600 μm isotropic spatial resolution) coronal images from three participants in the amnesic group (i–iii), illustrating significant in vivo volume loss in the hippocampus. T$_1$-weighted anatomical images from participants in the amnesic and control groups were used to conduct whole-brain voxel-based morphometry (VBM). The 7.0-Tesla T$_1$-weighted anatomical images are a subgroup re-analysis of those reported in our previous clinical study involving 18 participants at the chronic phase of the LGI1-antibody-complex LE phenotype (*Miller et al., 2017*). Normalized gray matter in the amnesic group and in the control group participants, derived from the whole-brain VBM analysis, were contrasted using a two-sample *t*-test and thresholded at p<0.05 family-wise error corrected for multiple comparisons with SPM12. No evidence of significant gray matter volume loss was found outside of the hippocampus in the amnesic group relative to the control group (see 'Results' section), which is in agreement with the results from VBM reported in our previous study (*Miller et al., 2017*). Adapted from *Miller et al. (2017)*, published under CC BY license, http://creativecommons.org/licenses/by/4.0/.

## Results

All data were collected at the chronic phase of the LGI1-antibody-complex LE phenotype (time between symptoms onset and study examination: median = 4 years, range = 7), suggesting that the outcomes were unlikely to be mediated by short-term compensatory processes. Clinical and laboratory characteristics, neuropsychological assessment, and quantitative measures of damage based on anatomical 7.0-Tesla MRI data have been previously published (*Miller et al., 2017*). New data on autobiographical memory for remote events and functional connectivity based assessments of resting-state fMRI data are reported here.

In brief, we first report autobiographical memory performance by examining the data obtained from administering the AI to participants in the amnesic and control groups. Differences in the retrieval of internal detail over time between the amnesic and control group participants were examined to assess: (a) whether there was a loss of internal detail in the amnesic group, and (b) if present, how the loss changed over time relative to the control group (*Figures 4* and *5*). Second, we report the results from standardized neuropsychological tests administered to the amnesic group in order to assess intelligence, attention, executive function, language, visuomotor skills, visuoconstructive skills, verbal memory, visual memory, and recognition memory (*Figure 6* and *Supplementary file 1a*). Third, anatomical MRI data acquired at 7.0-Tesla field strength were used to assess: (a) which hippocampal subfield volumes were affected in the amnesic group relative to the control group (*Figures 1* and *2*, *Figure 2—figure supplement 1*, and *Table 1*), and (b) whole-brain gray matter volume in the amnesic group relative to that in the control group (*Figure 3*). Fourth, we characterized the impact of CA3 damage on functional connectivity within the DN by testing for between-group differences in the topological properties of DN nodes defined by the *Andrews-Hanna et al. (2010)* parcellation scheme. Specifically, the graph theoretic measures were applied to investigate functional integration, functional segregation, and local measures of centrality (*Figures 7* and *8*). Fifth, in order to determine the scalar extent of altered topology, the same graph theoretic measures were applied to examine the topological properties of five other large-scale brain networks (somatomotor network, visual network, dorsal attention network, ventral attention network, and salience network). Sixth, functional connectivity was assessed using seed-to-voxel and region-of-interest-to-region-of-interest (ROI-to-ROI) based analyses involving left and right hippocampal seed ROIs and ROIs in the DN. Finally, the relevance of observed differences in CA3 volume and in the topological properties of the affected nodes for autobiographical episodic memory performance were assessed using

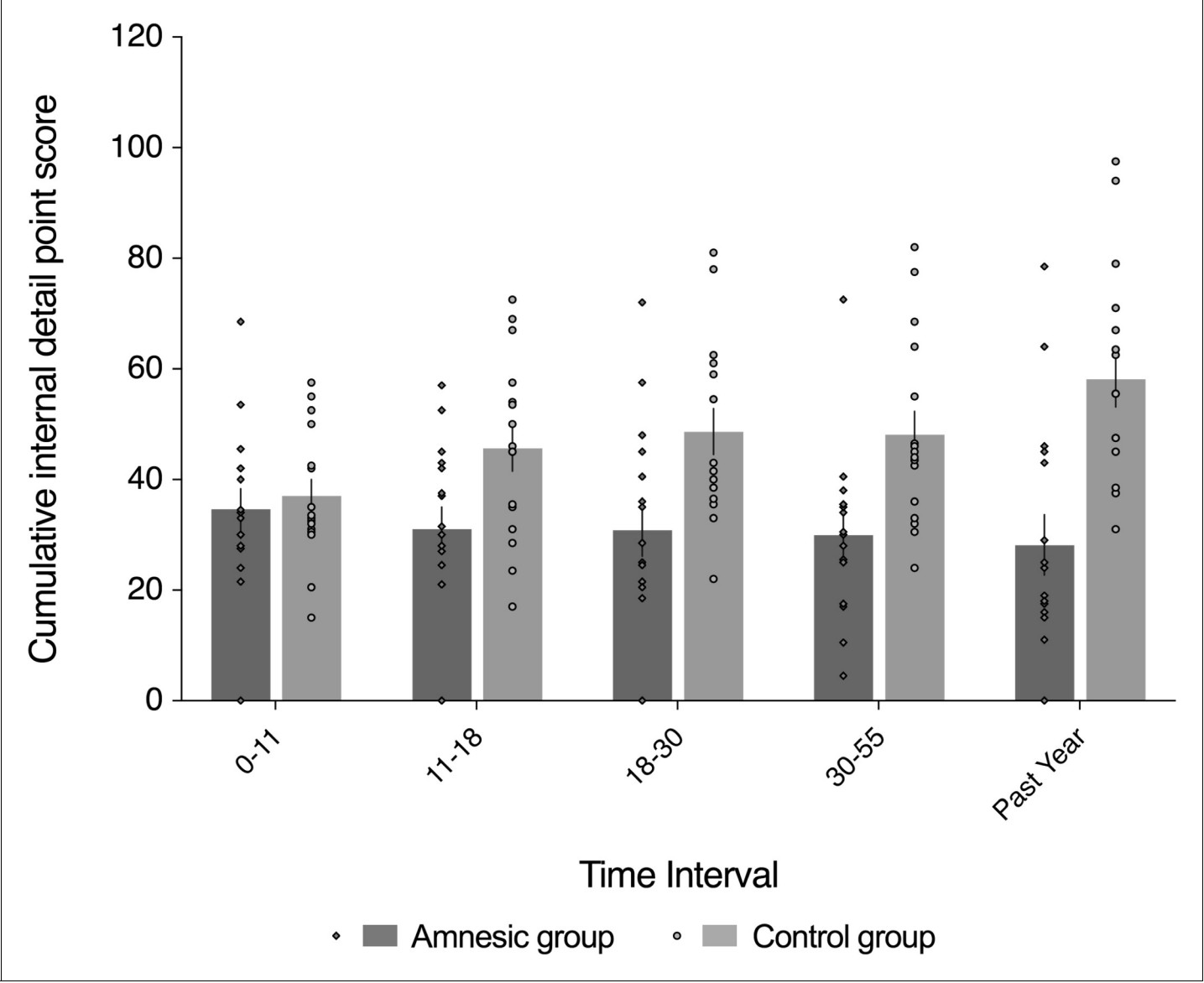

**Figure 4.** Loss of internal (episodic) detail in the amnesic group for recent and remote memories (up to ~50 years prior to the CA3 damage). Plot depicts mean cumulative number (summed across the general and specific probes) of internal (episodic) details generated on the AI across the five sampled intervals, as a function of group (amnesic group, N = 16; control group, N = 16). With the exception of the past year, each time interval refers to the age of the participant at the time of the remembered event; for example, 18–30 years refers to an event that occurred when each participant was between 18–30 years-of-age. A significant interaction between group and time (across all five sampled intervals) suggests that the loss of internal (episodic) detail was time-sensitive ($F_{(2.67,80.22)}$ = 3.91, p=0.015, $\eta^2_p$ = 0.115). Post hoc analyses revealed that the earliest remote memory (0–11 years) was intact ($F_{(1,30)}$ = 0.250, p=0.621), whereas there was temporally ungraded loss of internal detail across the remaining remote and recent memories (group: $F_{(1,30)}$ = 23.25, p<0.0001, $\eta^2_p$ = 0.437; group x time: $F_{(2.62,78.44)}$ = 1.51, p=0.222, $\eta^2_p$ = 0.048; time: $F_{(2.62,78.44)}$ = 0.604, p=0.592, $\eta^2_p$ = 0.020), extending up to ~50 years prior to the CA3 damage (11–18 year interval for internal detail, $F_{(1,30)}$ = 6.43, p=0.017). Error bars correspond to the s.e.m. The online version of this article includes the following figure supplement(s) for figure 4:

**Figure supplement 1.** Scatterplots of CA3 volume against total internal detail.

robust multiple regression based analyses. Inferences were two-sided at an alpha level of 0.05, with correction for multiple comparisons.

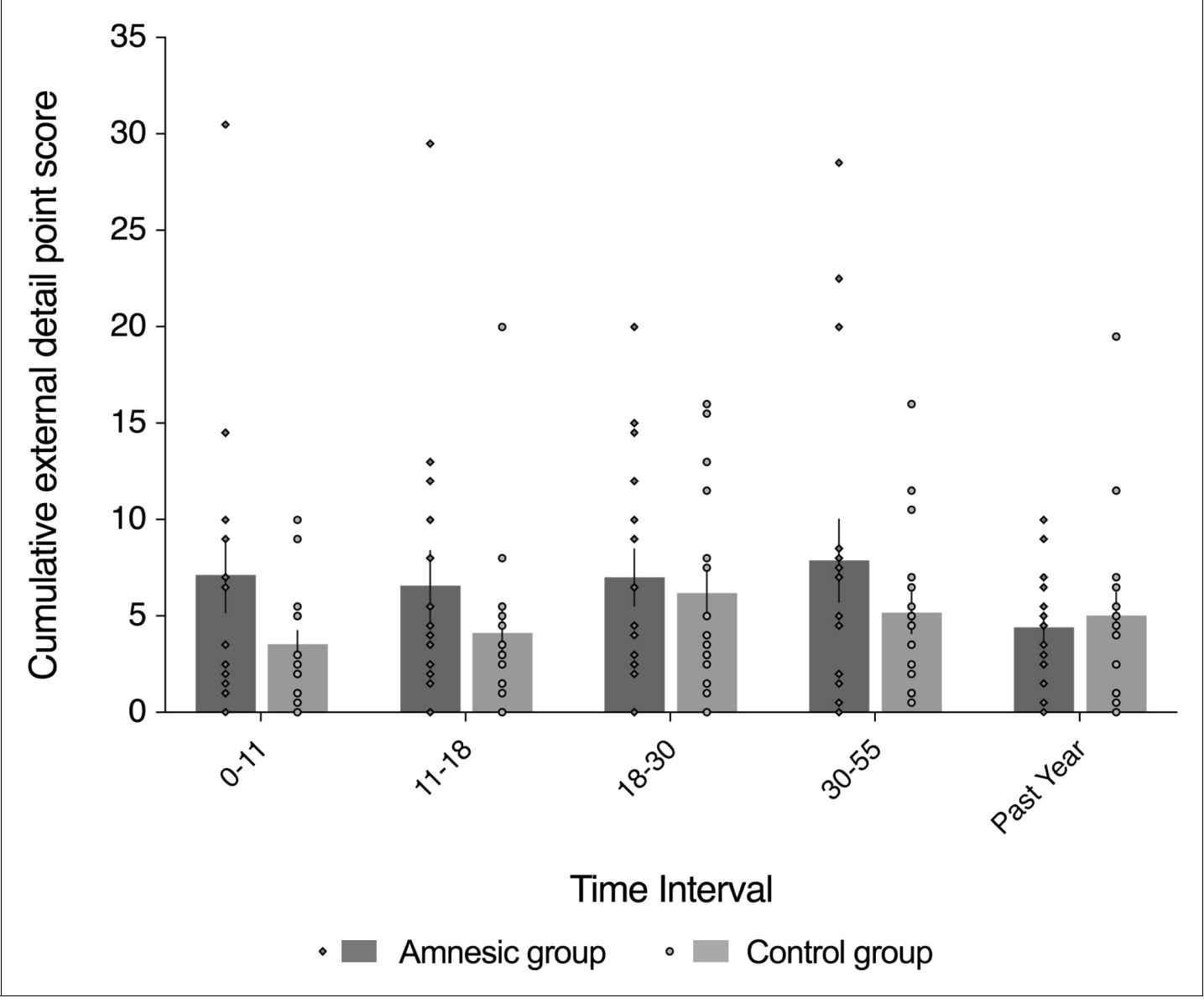

**Figure 5.** External (non-episodic, mainly personal semantic) detail was intact for recent and remote memories (~1–60 year interval). The plot depicts mean cumulative (summed across the general and specific probes) number of external details generated on the AI across the five sampled intervals, as a function of group (amnesic group, N = 16; control group, N = 16). With the exception of the past year, each time interval refers to age of the participants at the time of the remembered event; for example, 18–30 years refers to an event that occurred when each participant was between 18 and 30 years-of-age. The specificity of the deficit in internal (episodic) detail was revealed by the absence of a significant between-group difference in the amount of external (semantic) detail remembered over the five internals ($F_{(1,30)}$ = 1.24, p=0.275, $\eta^2_p$ = 0.040), and the interaction between group and time was not significant ($F_{(4,120)}$ = 1.46, p=0.218, $\eta^2_p$ = 0.046). Evidence for a null group difference in external detail aligns with the more general preservation of associative semantic memory (Camel and Cactus Test) (mean z-score = 0.20, s.e.m. = 0.30, $t_{(14)}$ = 0.67, p=0.514, two-tailed one-sample t-test) (**Bozeat et al., 2000**). Error bars correspond to the s.e.m.

## Cognition

### Autobiographical memory: amnesia for recent and remote episodic detail

*Figures 4* and *5* depict the results of statistical analyses conducted on the internal (episodic) and external (non-episodic, semantic) event details, respectively, acquired by administering the AI to the amnesic group (N = 16) and control group (N = 16), as a function of when each event memory occurred. In accordance with the standard administration of the AI, five intervals were acquired in all participants: the past year (termed, anterograde); 30–55, 18–30, 11–18, and 0–11 years (all termed,

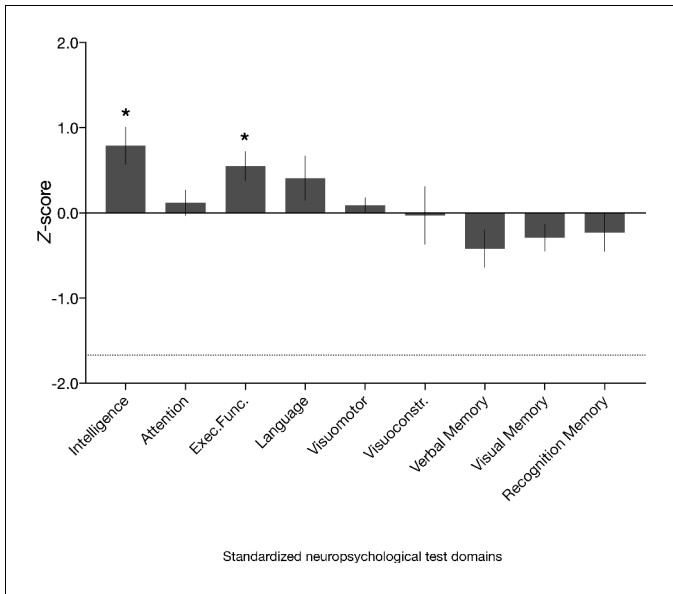

**Figure 6.** Neuropsychological domain performance in the amnesic group. Comprehensive assessment using standardized neuropsychological tests revealed that the scores for the amnesic group were comparable or significantly above normative data on composite measures of intelligence, attention, executive function, language, visuomotor skills, visuoconstruction skills, verbal memory, visual memory, and recognition memory (see 'Materials and methods' for individual subtest tests underlying the domain scores; *Supplementary file 1a* contains detailed results and N for each domain). Delayed verbal recall performance (which contributed to the verbal memory domain) was significantly different from normative data (N = 16, average z-score = −0.77, s.e.m. = 0.24, $t_{(15)}$ = −3.16, p=0.006), but above the threshold that typically indicates severe impairment (−1.67). Delayed verbal recall was comprised of Logical Memory II, Logical Memory II themes and Word Lists II (Wechsler Memory Scale–third edition [WMS-III]) and Doors and People, People Recall Test. By contrast, delayed visual recall (comprised of Rey Delayed Recall) was intact (average z = −0.08, s.e.m. = 0.20, $t_{(15)}$ = 0.41, p=0.685). Visuomotor = visuomotor skils; Visuoconstr. = visuoconstruction skills. Error bars correspond to the s.e.m.

retrograde). With the exception of the past year, these intervals refer to the age of the participant at the time of the remembered event; for example, 11–18 years refers to an event that occurred when the participant was between 11–18 years-of-age. As is convention (*Esopenko and Levine, 2017*), we report cumulative performance on the AI analyzed across the three levels of cueing used in the standardized administration. All responses on the AI were segmented and scored by two trained raters to obtain quantitative measures of internal (episodic) detail and external (non-episodic, semantic) detail. The raters were blinded to the identity and group membership of each transcript. Composite internal detail and external detail scores were computed for each participant by averaging the five response categories (i.e., event details, time, place, perceptual details, and thoughts and emotions) scores. The inter-rater correlation coefficient was calculated to be 0.97, across the internal and external detail scores, which is in line with the index study (*Levine et al., 2002*).

To preface the main results, a significant interaction between group and time (across all five sampled intervals) indicates that the loss of internal (episodic) detail loss was time-sensitive ($F_{(2.67,80.22)}$ = 3.91, p=0.015, $\eta^2_p$=0.115). Post hoc analyses revealed that the earliest remote memory (0–11 years) was intact ($F_{(1,30)}$ = 0.250, p=0.621), whereas there was a loss of internal detail across the remaining remote and recent memories that did not change over time (group: $F_{(1,30)}$ = 23.25, p<0.0001, $\eta^2_p$ = 0.437; group x time: $F_{(2.62,78.44)}$ = 1.51, p=0.222, $\eta^2_p$ = 0.048; time: $F_{(2.62,78.44)}$ = 0.604, p=0.592, $\eta^2_p$ = 0.020), extending up to ~50 years prior to the CA3 damage (11–18 year interval for internal detail, $F_{(1,30)}$ = 6.43, p=0.017) (*Figure 4*).

An omnibus 2 (group: amnesic, control) x 2 (detail type: internal, external) x 5 (time: past year [i.e., anterograde interval]; 30–55, 18–30, 11–18, and 0–11 years [i.e., retrograde intervals]) mixed-model factorial ANOVA on the units of information acquired on the AI revealed a significant interaction between group, detail type, and time ($F_{(3.50,105.11)}$ = 2.83, p=0.034, $\eta^2_p$ = 0.086; see Appendix

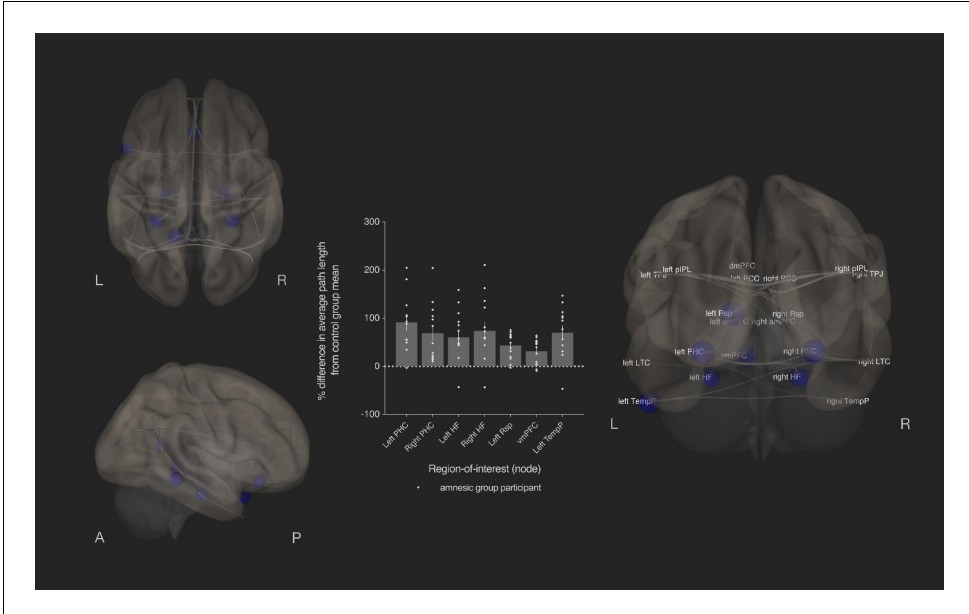

**Figure 7.** Bilateral damage to human CA3 disrupted integration of the MTL subsystem. Results from graph theoretic analyses of the DN in the amnesic and control groups, derived from 4-D rs-fMRI EPI images acquired at 7.0-Tesla MRI field strength. 3-D rendered brain depicts nodes (DN vertices/brain regions-of-interest) and their associated edges (paths between nodes) used to define the DN. The size of a node represents the beta values for that node. Network edges (adjacency matrix threshold): $z > 0.84$, one-sided (positive); analysis threshold: p-FDR corrected <0.05 (two-sided). Increased average path length of left and right parahippocampal cortex (MNI co-ordinates $-28,-40,-12$, $\beta = -1.41$, $t = -3.51$, p-FDR = 0.013; $28,-40,-12$, $\beta = -1.40$, $t = -3.53$, p-FDR = 0.013, respectively), left retrosplenial cortex (MNI co-ordinates $-14,-52,8$, $\beta = -0.75$, $t = -3.55$, p-FDR = 0.013), left and right hippocampal formation (MNI co-ordinates $-22,-20,-26$, $\beta = -0.93$, $t = -2.72$, p-FDR = 0.033 and MNI co-ordinates $22,-20,26$, $\beta = -1.18$, $t = -2.99$, p-FDR = 0.020, respectively), ventromedial prefrontal cortex (vmPFC) (MNI co-ordinates $0,26,-18$, $\beta = -1.52$, $t = -3.32$, p-FDR = 0.020), and left temporal pole of the dmPFC subsystem (MNI co-ordinates $-50,14,-40$, $\beta = -1.18$, $t = -3.34$, p-FDR = 0.013) in the amnesic group compared to the control group. The plot depicts differences in average path length at these MTL subsystem nodes and at the left temporal pole when comparing the amnesic group participants against the control group mean. Error bars correspond to the s.e.m.. *, p<0.05. The differences in average path length of the left PHC, left hippocampal formation, right hippocampal formation, left retrosplenial cortex, vmPFC, and left temporal pole were predictive of the retrieval of episodic details (i.e., mean composite internal details scored across all intervals). Comparable between-group differences in network topology were observed when an alternate threshold was used to test for functional connections (see section on stability of the effects and *Supplementary files 1h –1m*). Renders are depicted at the same threshold as those used to assess significance (i.e., p-FDR corrected <0.05 [two-sided]). The online version of this article includes the following figure supplement(s) for figure 7:

**Figure supplement 1.** Between-group hippocampal seed-to-voxel functional connectivity, at a lenient, *p*-uncorrected <0.05 cluster-size threshold.

**Figure supplement 2.** Left and right hippocampal seed-to-voxel functional connectivity, as a function of group.

---

1 for full three-way ANOVA results; *Figures 4* and 5). In order to explore the between-group differences in internal (episodic) detail as a function of the age of the memory, we conducted a post hoc 2 (group: amnesic, control) x 5 (time: past year [i.e., anterograde interval]; 30–55, 18–30, 11–18, and 0–11 years [i.e., retrograde intervals]) mixed-model ANOVA on the cumulative internal detail scores. Mauchly's test of sphericity was significant for time ($\chi^2_{(9)} = 31.84$, p<0.0001). Degrees of freedom were corrected using Greenhouse-Geisser estimates ($\varepsilon = 0.669$). In addition to a significant main effect of group ($F_{(1,30)} = 16.37$, p<0.0001, $\eta^2_p = 0.353$), there was a significant two-way interaction between group and time ($F_{(2.67,80.22)} = 3.91$, p=0.015, $\eta^2_p = 0.115$). The main effect of time was not significant ($F_{(2.67,80.22)} = 1.13$, p=0.337, $\eta^2_p = 0.036$). Crucially, these results suggest that the loss of internal (episodic) detail in the amnesic group relative to the control group changed across the five sampled intervals. In addition, the profile of loss in the control group is consistent with a recency

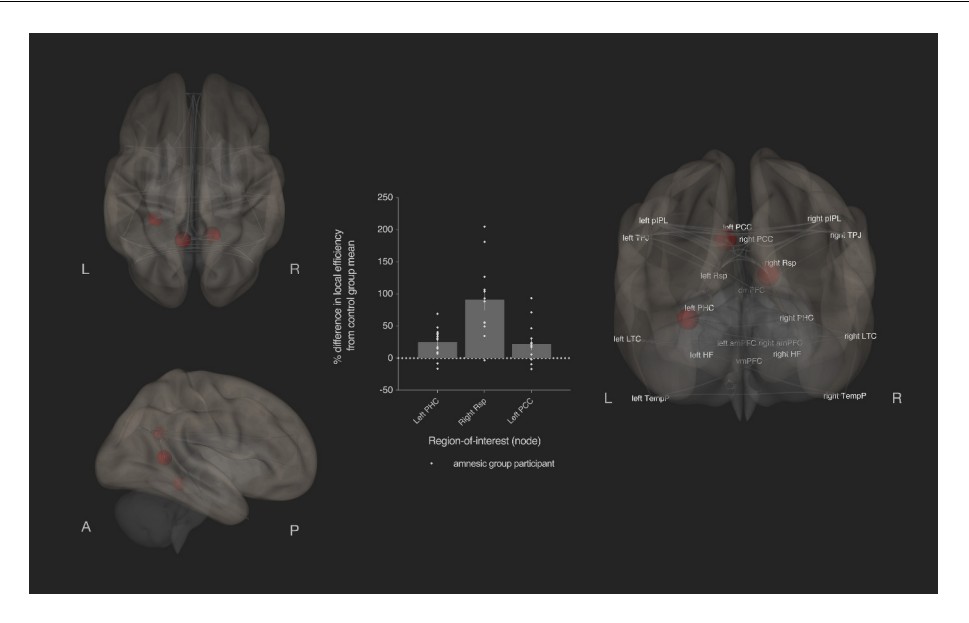

**Figure 8.** Bilateral damage to human CA3 increased local efficiency in three DN nodes. Results from graph theoretic analyses of the DN in the amnesic and control groups, derived from 4-D rs-fMRI EPI acquired at 7.0-Tesla MRI field strength. 3-D rendered brain depicts nodes (DN vertices/brain regions-of-interest) and their associated edges (paths between nodes) used to define the DN. The size of a node represents the beta value for that node. Network edges (adjacency matrix threshold): $z > 0.84$, one-sided (positive); and, analysis threshold: two-sided p-FDR corrected <0.05 (two-sided). Local efficiency was significantly increased in three nodes: left posterior cingulate cortex (left PCC) (MNI co-ordinates $-8,-56,26$, $\beta = 0.35$, $t = 3.49$, p-FDR = 0.020), left parahippocampal cortex (left PHC) (MNI $-28,-40,-12$, $\beta = 0.45$, $t = 3.25$, p-FDR = 0.037) and right retrosplenial cortex (right Rsp) (MNI 14,–52,8, $\beta = 0.36$, $t = 3.45$, p-FDR = 0.020). Plot depicts differences in local efficiency in amnesic group participants from the mean of the control group. Differences in local efficiency at these nodes from the mean of the control group were not predictive of internal (episodic) detail performance on the AI (i.e., mean composite internal details score across all intervals) (***Supplementary file 1q***). Renders are depicted at the same threshold as that used to assess significance (i.e., p-FDR corrected <0.05 [two-sided]). Comparable between-group differences in network topology were observed when an alternate threshold was used to test for functional connections (see section on stability of the effects and ***Supplementary files 1h–1m***). Plot depicts mean local efficiency at these three nodes. Error bars correspond to the s.e.m.. *p<0.05.

effect for episodic detail, such that episodic detail decreased as a function of the age of the memory (***Noulhiane et al., 2007***; ***Piolino et al., 2009***; ***Rubin and Schulkind, 1997***).

Visual inspection of ***Figure 4*** suggests a null difference in internal (episodic) detail generation for earliest remote memory (i.e., 0–11 year interval). A post hoc direct group comparison revealed a null difference at the 0–11 year interval ($F_{(1,30)} = 0.25$, p=0.621). Early memories have been described as gist-like (***Hardt et al., 2013***; ***Richards and Frankland, 2017***; ***Sadeh et al., 2014***), and are arguably qualitatively different from other remote memories along several dimensions (***Barclay and Wellman, 1986***; ***Cermak, 1984***; ***Sekeres et al., 2018***; ***Winocur and Moscovitch, 2011***), lose contextual specificity over time, and can be supported by extra-hippocampal regions such as the medial prefrontal cortex (***Clewett et al., 2019***; ***Wiltgen et al., 2010***; ***Winocur et al., 2010***; ***Winocur et al., 2007***). By inference, the earliest, intact remote memory may not be hippocampal (CA3)-dependent. In order to assess whether the earliest remote memory was qualitatively different from the other remote memories, a post hoc one-way ANOVA with time (past year, 30–55, 18–30, 11–18, and 0–11 years) as the repeated-measures variable was conducted to assess whether or not the ratio of external detail was elevated relative to internal detail in the control group for the earliest remote memory. Mauchly's test of sphericity was not significant for time ($\chi^2_{(5)} = 15.86$, p=0.071). The main effect of time was not significant ($F_{(4,60)} = 1.04$, p=0.396, $\eta^2_p = 0.065$). Hence, the earliest remote memories were not detectably schematized to a state that rendered them qualitatively different from the more recent

(remote) memories, at least when assessed by examining the ratio between external and internal detail.

*Figure 4* also points to loss of internal (episodic) detail at all of the other sampled intervals (i.e., the past year, 30–55, 18–30, and 11–18). To assess the profile of retrograde amnesia across these intervals, we conducted a post hoc 2 (group: amnesic, control) x 4 (time: past year, 30–55, 18–30, and 11–18 years) mixed-model ANOVA on the cumulative internal detail scores. Mauchly's test of sphericity was significant for time ($\chi^2_{(5)}$ = 11.93, p=0.036), so degrees of freedom were corrected using Huynh-Feldt estimates ($\varepsilon$ = 0.872). The loss of internal detail was evident in the main effect of group ($F_{(1,30)}$ = 23.25, p<0.0001, $\eta^2_p$ = 0.437), whereas neither the two-way interaction between group and time nor the main effect of time were significant ($F_{(2.62,78.44)}$ = 1.51, p=0.222, $\eta^2_p$ = 0.048; $F_{(2.62,78.44)}$ = 0.604, p=0.592, $\eta^2_p$ = 0.020, respectively). Hence, this post hoc analysis revealed that the loss of internal detail in the amnesic group did not change across these recent and remote memories (i.e., it was temporally ungraded/a flat gradient), and spanned up to ~50 years prior to CA3 damage (11–18 year interval for internal detail, $F_{(1,30)}$ = 6.43, p=0.017), when the loss was assessed without the earliest, intact remote memory. Together, these post hoc analyses revealed that the loss of internal detail was time-sensitive: retrograde amnesia was temporally ungraded across recent and remote memories and spanned up to ~50 years prior to CA3 damage, whereas the earliest remote memory was intact.

The specificity of the deficit in episodic (internal) detail was revealed by a companion post hoc 2 (group: amnesic, control) x 5 (time: past year, 30–55, 18–30, 11–18, and 0–11 years) mixed-model ANOVA conducted on the cumulative external detail scores. Mauchly's test of sphericity was not significant for time ($\chi^2_{(9)}$ = 4.85, p=0.848). There was no significant between-group difference in the amount of external (semantic) detail over the five intervals ($F_{(1,30)}$ = 1.24, p=0.275, $\eta^2_p$ = 0.040) (*Figure 5*). Moreover, the interaction between group and time for external detail was not significant ($F_{(4,120)}$ = 1.46, p=0.218, $\eta^2_p$ = 0.046), nor was the main effect of time ($F_{(4,120)}$ = 1.41, p=0.234, $\eta^2_p$ = 0.045). Importantly, the evidence showing that amnesic group participants generated comparable external (non-episodic, semantic) details to those of control group participants suggests that the loss of internal (episodic) detail across the lifespan did not reflect an impairment in the ability to generate detail per se. The preservation of personal semantic detail in the amnesic group aligns with the more general preservation of associative semantic memory (Camel and Cactus Test) (mean *z*-score = 0.20, s.e.m. = 0.30, $t_{(14)}$ = 0.67, p=0.514, two-tailed one-sample *t*-test) (*Bozeat et al., 2000*).

## No evidence of impairment in susceptibility to tangents

The absence of impairment in the retrieval of external (non-episodic, semantic) detail, intact verbal fluency on the Graded Naming Test (mean *z*-score = 0.75, s.e.m. = 0.28, $t_{(15)}$ = 2.71, p=0.016, two-tailed one-sample *t*-test), and the use of a similar number of words to describe the episodes (amnesic group, total number of words = 106,047, mean number of words per interview = 6628, s.e.m. = 696; control group, total number of words = 97,654, mean number of words per interview = 6103, s.e.m = 500; $t_{(15)}$ = 0.59, p=0.56) suggests that the amnesic group autobiographical episodic memory deficit did not reflect a general impairment in verbal output (*Barnett et al., 2000*). However, it has also been suggested that participants with amnesia are susceptible to losing track of their narratives (*Dede et al., 2016*). Minimizing the frequency of such so-called tangents with 'supportive questioning' has been shown to lead to intact autobiographical memory in a group of six participants with amnesia for all sampled intervals except the near past (*Dede et al., 2016*).

Tangents were operationalized in accordance with the protocol described by *Dede et al. (2016)*. In particular, each detail was assigned a relevance rating from 1 to 4, where 1 corresponded to highly relevant and 4 corresponded to an irrelevant detail for the narrative of the central event. The generation of three or more consecutive details assigned a relevance rating of 4 was scored as a tangent. A return to the narrative of the central event, following an irrelevant detail, was recorded when one or more relevant details was produced either before the completion of the narrative or before being prompted by the experimenter (*Dede et al., 2016*), p. 13,478). Results from the analysis revealed that the amnesic group participants were comparable to the control group participants in their susceptibility to tangents during narrative construction (cumulative total tangents across five intervals: amnesic group = 0.20, control group = 0.63), which suggests that the deficits in the

recollection of episodic detail were unlikely to reflect insufficiently supportive questioning during the adminstration of the AI.

## Other cognitive functions: general neuropsychological assessment

It can be challenging to disentangle the impact of anterograde cognitive pathology on retrograde deficits. However, in line with prior studies from our laboratory and other laboratories on the chronic phase of the LGI1-antibody-complex LE phenotype (*Argyropoulos et al., 2019*; *Frisch et al., 2013*; *Malter et al., 2014*; *McCormick et al., 2016*; *McCormick et al., 2017*; *McCormick et al., 2018b*; *Miller et al., 2017*), extensive neuropsychological assessment revealed no evidence of dysfunction in the amnesic group on standardized neuropsychological tests outside of memory (median = 4 years post-onset, range = 7). In particular, domain indices of intelligence, executive function, attention, language, visuomotor skills, and visuoconstruction skills were comparable to normative values (*Figure 6* and *Supplementary file 1a*). Hence, the deficits in recent and remote memory are unlikely to be secondary to impairments in cognitive faculties that are necessary for autobiographical retrieval, such as attention, language, and executive function.

In terms of standardized tests of memory, there were no significant deficits evident on the composite indices outside of delayed verbal recall (comprised of Logical Memory II, Logical Memory II themes and Word Lists II [WMS-III] and the People Recall Test) (n = 16, average z-score = −0.77, s.e. m. = 0.24, $t_{(15)}$ = –3.16, p=0.006), which is in line with evidence that delayed recall is sensitive to hippocampal damage (*Aggleton and Shaw, 1996*; *Mayes et al., 2002*). By contrast, delayed visual recall (comprised of Rey Delayed Recall) was intact (average z = −0.08, s.e.m. = 0.20, $t_{(15)}$ = 0.41, p=0.685). Prior studies have shown that the chronic phase of the LGI1-antibody-complex LE phenotype is associated with persistent memory deficits (*Bettcher et al., 2014*; *Butler et al., 2014*; *Malter et al., 2014*), particularly in episodic verbal memory (*Finke et al., 2017*; *McCormick et al., 2016*; *McCormick et al., 2017*; *McCormick et al., 2018b*; *Miller et al., 2017*). The evidence of intact recognition memory suggests that the deficit in autobiographical episodic detail in the amnesic group was not due to a general inability to remember (*Figure 6* and *Supplementary file 1a*).

The specificity of impairment accompanying CA3 damage is broadly consistent with the hypothesis that damage to hub regions, which underpin subnetworks relevant for specific cognitive functions, can generate specific cognitive deficits (*Gratton et al., 2012*). Nonetheless, the neuropsychological profile of participants in studies that have examined functional connectivity rs-fMRI in hippocampal amnesia is quite variable. In a recent study reported by *Henson et al. (2016)*, two of the six participants with amnesia exhibited retrograde and anterograde amnesia alongside generally preserved cognition on neuropsychological assessment outside of memory. Another participant exhibited deficits on test of episodic memory, but largely preserved autobiographical memory (with the exception of details near the time of injury), and preserved cognition outside of memory. In a study by *Hayes et al. (2012)*, all three participants with MTL damage (involving regions that included the hippocampus, parahippocampal gyrus, lateral temporal cortex, amygdala, and temporal pole) were impaired on Wechsler Memory Scale-III indices of immediate and delayed episodic memory, exhibited anterograde and retrograde amnesia, and had intact working memory performance. Cognitive domains outside of memory were not reported. It is conceivable that broader cognitive deficits are more likely to occur when damage involves connector regions (which co-ordinate between multiple subnetworks) (*Gratton et al., 2012*). More focal damage limited to human CA1 in a group of 16 participants, secondary to acute transient global amnesia (lasting 8.3 ± 1.9 hr), was associated with deficits on verbal and visuoconstructive memory tests, whereas naming and conceptual knowledge, general intellectual abilities, and visual attention were intact (*Bartsch et al., 2011*).

## 7.0-Tesla MRI

### Anatomical and resting-state functional connectivity MRI

Anatomical and functional MRI data acquisition was conducted at 7.0-Tesla field strength in 16 participants at the chronic phase of the LGI1-antibody-complex LE phenotype and in 15 of the age- and education-matched control participants in order to conduct quantitative hippocampal subfield volumetric morphometry (390 μm in-plane spatial resolution), to perform whole-brain voxel-by-voxel

morphometry (600 μm isotropic spatial resolution), and to measure functional network properties in whole-brain resting-state networks (2 mm³ isotropic resolution). It was not possible to scan one of the 16 participants in the control group because of a technical issue. As noted in the introduction, the anatomical MRI data were reported in our previous study, where we found that the chronic phase of the LGI1-antibody-complex LE phenotype was associated with damage limited to bilateral CA3 (*Miller et al., 2017*). In the amnesic group, all MRI data were acquired several years after disease onset (median 4 years post-autoimmune encephalitis onset; range = 7).

## Quantitative hippocampal subfield morphometry

Dice similarities (amnesic group median = 0.79, control group median = 0.76) and intra-class correlation-coefficient-based metrics (amnesic group median = 0.98, control group median = 0.97) across lateralized subfield volumes demonstrated a high degree of reliability in the output of two full repetitions of the segmentation protocol in the entire dataset. Inter-rater Dice Similarity Indices (DSIs) of two independent raters for the amnesic group and control group were also reliable (median amnesic group and control group DSIs across all subfields: 0.75 and 0.74, respectively).

In line with our previous study on 18 participants at the chronic phase of the LGI1-antibody-complex LE phenotype (*Miller et al., 2017*), quantitative three-dimensional whole-hippocampal volumetry of five hippocampal subfields (CA1–3, DG, and SUB) conducted on the 7.0-Tesla 3-D fast-spin echo images indicated that participants in the amnesic group (15 of the participants reported here were included our previous study) had volume loss confined to bilateral CA3 when compared with the matched control group ($F_{(1,28)}$ = 14.52, p=0.001, Cohen's *d* = 1.39; *Figures 1* and *2*, *Table 1*, and *Figure 2—figure supplement 1*). Full results from a three-way mixed-model ANOVA conducted to examine these effects are reported in Appendix 1 (see *Figure 2—figure supplement 1* for a plot showing CA1–3, DG, and SUB subfield volumes for each amnesic group and control group participant). Notably, left (mean volume loss = -29%, SEM = 0.04) and right (mean volume loss = -29%, SEM = 0.03) CA3 exhibited comparable volume loss in the amnesic group (N = 15) when contrasted against the corresponding control mean CA3 volumes (N = 15). A relative reduction in CA3 volumes to their matched control was observed in all amnesic group participants. Also in line with our prior study of 18 amnesic participants with LGI1-antibody-complex LE, CA1 volume loss was not significant when the alpha criterion was corrected for multiple comparisons (mean volume loss = -16%, $F_{(1,28)}$ = 5.25, p-uncorrected = 0.019, Cohen's *d* = 0.91).

Auto-antibodies to the two principal antigenic components of the voltage-gated potassium channel (VGKC)-complex—LGI1 and CASPR2 proteins—are preferentially expressed in CA3 and CA1 (*Irani et al., 2010*). As noted, unlike volume loss in CA3, volume loss in CA1 did not reach statistical significance when corrected for multiple comparisons. The observed selectivity of CA3 volume loss is consistent with the anatomical localization of enrichment of LGI1 gene transcripts in CA3 of the adult human brain (*Hawrylycz et al., 2012*), the expression of LGI1 gene transcripts in mouse CA3 (*Herranz-Pérez et al., 2010*), and evidence of greater neuronal loss in CA3 compared to CA1 following seizures in homozygous LGI1 knockout mice (*Chabrol et al., 2010*). When compared to CA1, CA3 exhibits particular vulnerability to excitotoxic lesions associated with seizures, given that IgG-containing LGI1 antibodies induce population epileptiform discharges either in CA3 pyramidal neurons in vitro (*Lalic et al., 2011*), or from complement-mediated fixation of bound antibodies (*Bien et al., 2012*). No other lesions were detected in any of the amnesic group participants. Evidence of selective anatomical damage associated with LGI1 pathogenesis suggests that the chronic phase of the disease represents a compelling lesion model for studying the causal role of human CA3 in the hippocampal network.

Results from one participant in the amnesic group were not available, because it was not possible to segment all five hippocampal subfields across the entire longitudinal axes of both hippocampi, due to insufficient contrast for the delineation of subfield boundaries on each coronal slice. The same participant exhibited bilateral hippocampal volume loss compared to the control group mean and met all a priori inclusion criteria for participation in the study. Re-examination of the results from the AI revealed that the findings held when this participant was removed from the main analyses (see Appendix 1).

Of note, the CA3 volume of another participant in the amnesic group was 8% greater in total volume than the control group mean (*Figure 2*). When considered against the matched control for the

participant, CA3 volume loss (-18%) was within the range of the other amnesic group participants. In addition, the participant exhibited memory impairment on the AI that was characteristic of the amnesic group participants; specifically, internal detail was 52.9% below control group mean and 62% below the internal detail remembered by the matched control participant. The main results from the AI were replicated when the data were reanalyzed without including this participant (see Appendix 1). The participant also met all a priori inclusion criteria for participation in the study and also exhibited altered functional connectivity.

## Whole-brain voxel-by-voxel morphometry

Whole-brain voxel-by-voxel morphometry (VBM) and diffeomorphic anatomical registration using the exponentiated Lie algebra (DARTEL) registration method (*Ashburner and Friston, 2009*) were conducted on the 7.0-Tesla (0.6 × 0.6 × 0.6 mm$^3$ spatial resolution) T$_1$-weighted anatomical images (*Figure 3*; amnesic group, N = 15, control group N = 15). SPM12 did not register one of the amnesic group participants and so the scan was removed from further VBM analyses. The same participant was excluded from all resting-state functional connectivity analyses because the motion parameters did not meet the minimum mean framewise displacement threshold (<0.5 mm). The voxel-by-voxel contrast of normalized gray matter across the whole-brain was conducted using a two-sample *t*-test thresholded at p<0.05, with family-wise error correction for multiple comparisons. The results revealed no suprathreshold clusters of gray matter volume loss in amnesic group participants relative to the control group. The absence of gray matter volume loss elsewhere in the brain corroborates our prior results, conducted on an independent 3.0-Tesla dataset and our previous 7.0-Tesla-based study involving 18 participants at the chronic phase of the LGI1-antibody-complex LE phenotype (*McCormick et al., 2016*; *McCormick et al., 2017*; *McCormick et al., 2018b*; *Miller et al., 2017*), and those from other laboratories (testing over 130 chronic phase individuals; for an exception, see *Argyropoulos et al., 2019*) that have conducted volumetric studies of LGI1-antibody-complex LE with either comparable VBM analyses or with Freesurfer-based whole-brain segmentation (http://surfer.nmr.mgh.harvard.edu/) (*Finke et al., 2017*; *Hanert et al., 2019*; *Wagner et al., 2015a*; *Wagner et al., 2015b*).

## Functional connectivity

Functional MRI data for the resting-state analysis were collected at 7.0-Tesla field strength in one sequential acquisition of 200 volumes optimized for functional connectivity analysis, using a echo-planar imaging (EPI) sequence, providing blood-oxygen-level dependent contrast images (2 × 2 × 2 mm, TR = 2500 s, TE = 25 ms, FOV 192×150×120, flip angle = 90° (nominal), 60 slices, slice gap = 0, slice thickness = 2 mm). Echo-planar images from one participant in the amnesic group were excluded because of the poor quality of the data collected. In particular, the level of motion in the EPI images did not meet the minimum mean framewise displacement threshold (i.e., <0.5 mm) for inclusion in the resting-state functional connectivity analyses, which was estimated from the reference (i.e., realignment) volume threshold. Voxelwise group effects were considered significant at a p-False Discovery Rate (FDR)-corrected threshold set at <0.05 (*Benjamini and Hochberg, 1995*).

## Integration within the MTL subsystem was reduced by damage to the human CA3 network

The rs-fMRI data were interrogated to describe the topological organization of the two subsystems and mainline core of the DN, defined by the parcellation scheme proposed by *Andrews-Hanna et al. (2010)*. Network topological properties (i.e., the arrangement of nodes and edges) of the nodes that comprised the two subsystems and the mainline core were examined by computing each graph theoretic measure for each network node (8 mm spherical regions-of-interest and their pairwise edges). In line with previous studies, we computed measures of functional integration (average path length and global efficiency), measures of functional segregation (clustering coefficient and local efficiency), and local measures that consider the centrality of nodes (degree and betweenness centrality). Functional integration examined the capacity of nodes within a network to combine information from distributed regions, whereas the measures of functional segregation were a proxy for the capacity for specialized processing within densely interconnected groups of regions. It is important to consider graph theoretic measures together, because, for example, an increase in global

efficiency that is accompanied by a reduction in clustering coefficient could reflect an imbalance between functional integration and segregation.

Average path length expresses the average value of the shortest path lengths in a graph and is inversely related to the (integrative) global efficiency of information exchange over a network. A smaller path length thus represents greater integration. Removal of connections in functional hub regions reduces global efficiency (*Hwang et al., 2013*), reflecting a loss of network integration (i.e., a loss of efficient communication). Clustering coefficient estimates the extent to which connectivity is clustered around a node, independently of its membership of a particular module, and reflects the functional specificity of regional brain areas. Degree (the number of edges maintained by a node) and betweenness centrality (the number of short communication paths of which a node is a member) were computed as local measures. For a detailed interpretation of these graph theoretic measures, see *Bullmore and Bassett (2011)* and *Rubinov and Sporns (2010)*. Descriptions of the equations for these graph theoretic measures can be found at www.nitrc.org/projects/conn.

Weights in the connectivity matrices represent the *z*-scores of Pearson correlations and were computed with a thresholding approach that included *z*-scores > 0.84 (*Harrington et al., 2015*). The approach to thresholding was designed to balance statistical evidence of connectivity with avoiding less reliable sparse networks (*Rubinov and Sporns, 2011*; *Wang et al., 2011*). Negative *z*-scores were excluded because these can reduce the reliability of graph theoretic measures (*Wang et al., 2011*). Tests for between-group differences in all graph theoretic measures and functional networks were FDR-corrected for multiple comparisons ($p<0.05$).

Compared to the control group, the amnesic group exhibited a loss of integration (increased average path length) in nodes of the MTL subsystem (*Figure 7*); namely, left and right parahippocampal cortex (MNI co-ordinates $-28,-40,-12$, $\beta = -1.41$, $t = -3.51$, p-*FDR* = 0.013 and MNI co-ordinates $28,-40,-12$, $\beta = -1.40$, $t = -3.53$, p-FDR = 0.013, respectively), left retrosplenial cortex (MNI co-ordinates -14,-52,8, $\beta = -0.75$, $t = -3.55$, p-FDR = 0.013), left and right hippocampal formation (MNI co-ordinates $-22,-20,-26$, $\beta = -0.93$, $t = -2.72$, p-FDR = 0.033 and MNI co-ordinates $22,-20,26$, $\beta = -1.18$, $t = -2.99$, p-FDR = 0.020, respectively), and ventromedial prefrontal cortex (vmPFC) (MNI co-ordinates 0, 26,$-18$, $\beta = -1.52$, $t = -3.32$, p-FDR = 0.020). Increased average path length was also observed in the left temporal pole of the dmPFC subsystem (MNI co-ordinates $-50,14,-40$, $\beta = -1.18$, $t = -3.34$, p-FDR = 0.013). No other regions survived the p-FDR <0.05 threshold. At an uncorrected analysis threshold (i.e., p-uncorrected <0.05), there were between-group differences in the left and right posterior cingulate cortices (MNI co-ordinates $-8,-56,26$, $\beta = -0.50$, $t = -2.11$, p-FDR = 0.098, *p*-uncorrected = 0.044 and MNI co-ordinates $8,-56,26$, $\beta = -0.61$, $t = -2.39$, p-FDR = 0.059, p-uncorrected = 0.024, respectively). No other regions were significant at p-uncorrected <0.05 (see *Supplementary file 1b*).

Local efficiency was altered in the left parahippocampal cortex (MNI co-ordinates $-28,-40,-12$, $\beta = 0.45$, $t = 3.25$, p-FDR = 0.037), right retrosplenial cortex (MNI co-ordinates 14,$-52,8$, $\beta = 0.36$, $t = 3.45$, p-FDR = 0.020), and left posterior cingulate cortex (MNI $-8,-56,26$, $\beta = 0.35$, $t = 3.49$, p-FDR = 0.020) (*Figure 8*). No other regions survived the p-FDR <0.05 threshold (see *Supplementary file 1b*). Only the left retrosplenial cortex was significant at an uncorrected (i.e., p-uncorrected <0.05) analysis threshold (MNI co-ordinates $-14,-52,8$, $\beta = 0.30$, $t = 2.26$, p-FDR = 0.165, p-uncorrected = 0.033). Local efficiency alterations in left parahippocampal cortex were not correlated with increases in average path length ($r = -0.057$, p=0.769, two-tailed). All other topological properties in the two subsystems and midline core of the DN were spared in the amnesic group compared to the control group (*Supplementary file 1b*), including nodes within regions identified with semantic autobiographical memory, such as the medial PFC, middle and inferior temporal regions (e.g., fusiform gyrus) (*Addis et al., 2004b*; *Levine et al., 2004*; *Martinelli et al., 2013*). In particular, global efficiency, clustering coefficient, betweenness centrality, and degree were not significantly different at p-FDR <0.05 in the amnesic group when compared to the control group (*Supplementary file 1b*).

## CA3 damage has no impact on the topology of other large-scale brain networks

The selectivity of the effects of CA3 damage on network topology was examined by conducting analyses with nodes and edges that corresponded to five other large-scale resting-state functional

networks (RSNs), previously associated with alterations in functional connectivity following hippo-campal and MTL damage; namely, the somatomotor network, the visual network, the salience network (paralimbic structures, involved in externally directed task engagement and in maintaining a 'saliency'/priority map of the visual environment), the ventral attention network, and the dorsal attention network. MNI co-ordinates for these networks were derived from 13-module parcellation of the 264-node groundtruth graph reported by *Power et al. (2011)* (*Supplementary file 1n*). Graph theoretic measures and thresholding criteria were equivalent to those used for the DN analysis. In the amnesic group, none of these networks exhibited altered topological properties in any of the graph theoretic measures relative to the control group at p-FDR <0.05 (see *Supplementary files 1c–1g* for results at p-FDR <0.05 and p-uncorrected <0.05 analysis thresholds).

## Stability of effects

In order to evaluate the extent to which the results from the graph theoretic analyses were sensitive to the chosen threshold, we repeated the analyses using a different approach to thresholding in which the connectivity matrix (i.e., network edges) was based on cost. Cost measures the proportion of connections for each node in relation to all connections in the graph. The results from these re-analyses were evaluated at a corrected p-FDR <0.05 analysis threshold and are reported in *Supplementary files 1h-1m*. In addition, for completeness, we also report all results from these analyses at an uncorrected analysis threshold (p-uncorrected <0.05). In summary, the results revealed overlap with our original analysis in the nodes that were significantly different, when amnesic and controls groups were compared using a cost threshold set at 0.15; that is, where the strongest 15% of possible edges and edge weights in the network were retained. The increase in local efficiency in the left posterior cingulate cortex was, however, was not replicated at the p-FDR corrected threshold (MNI −8,−56,26, $\beta$ = 0.41, $t$ = 2.57, p-FDR = 0.098, p-uncorrected = 0.021). A cost threshold of 0.15 has been shown to yield a high degree of reliability when comparing estimates of graph theoretic measures across repeated sessions or runs (*Whitfield-Gabrieli and Nieto-Castanon, 2012*), and is frequently applied in studies examining large-scale network topology (*Bertolero et al., 2015*), because it is at the center of the ideal cost range where many graph theoretic measures are maximal (*Bullmore and Bassett, 2011*). In addition, the altered topology was expressed on the same graph theoretic measures – that is, average path length and local efficiency – as those observed in the original analysis, which was based on an a priori adjacency matrix threshold set at $z$ > 0.84 (one-sided [positive]) and an analysis threshold set at p-FDR corrected <0.05 (two-sided). This overall stability suggests that thresholding $z$ > 0.84 is representative of the underlying data.

## Seed-to-voxel and ROI-to-ROI functional connectivity analyses

Functional connectivity in hippocampal amnesia has also been studied using seed-based analyses of rs-fMRI, revealing that bilateral hippocampal damage in humans can alter the cortico-hippocampal network. The graph theoretic analyses provided information on how bilateral damage to human CA3 can modulate average path length and local efficiency, primarily in brain regions that reside within the MTL subsystem. In order to examine whether the differences in left and right hippocampal average path length are associated with alterations in functional connectivity (as assessed by generating a time-series correlation-strength map) across the whole brain and/or with specific brain areas, we conducted post hoc seed-based functional connectivity analyses. These analyses were conducted in two-ways: (1) seed-to-whole-brain voxelwise analyses (henceforth referred to as, seed-to-voxel) were conducted to test for significant between-group differences in the correlation of the left and right hippocampal seed regions with the rest of the brain (*Biswal et al., 2010*; *Fox et al., 2005*); and (2) we tested for significant between-group differences in the functional connectivity of the left and right hippocampus seed ROIs with ROIs in the DN (i.e., ROI-to-ROI). The pre-processing parameters and pipelines were the same as those applied to the graph theoretic analyses (see 'Materials and methods'). ROI masks were generated using the SPM toolbox, MarsBaR (*Brett et al., 2002*), in SPM12, and were spheres with a radius size of 8 mm.

Between-group differences in seed-to-voxel functional connectivity were examined by entering a left hippocampal seed region (MNI co-ordinates −24,−22,−16), based on co-ordinates implicated in episodic memory (*Hirshhorn et al., 2012*), because the functional nodes proposed by *Power et al. (2011)* and by *Andrews-Hanna et al. (2010)* do not cover the main body of the

hippocampus. On the grounds that the damage to CA3 was bilateral, we elected to assess functional connectivity with the left and right hippocampus as the seed regions. Therefore, a right hemispheric homotopic region of the hippocampus was also entered into the analysis (i.e., MNI co-ordinates 24,–22,–16). In addition to the left and right hippocampal seed regions, between-group seed-to-voxel functional connectivity was computed for two other ROIs: (i) a region in the occipital pole that occurred within the visual network (MNI co-ordinates 18,–47,–10); and, (ii) a region in primary motor cortex that occurred within the somatomotor network (MNI co-ordinates –40,–19,54). We examined these additional regions because other studies have observed that participants' at the chronic phase of the LGI1-antibody-complex LE phenotype exhibited altered functional connectivity in sensorimotor and visual networks relative to control participants (*Heine et al., 2018*). Thus, the visual and somatomotor seeds originated within networks that have been associated with altered functional connectivity following hippocampal damage involving the same aetiology.

In brief, the average time courses from these seed regions were extracted and every other voxel's time series was correlated against them to generate a correlation-strength map covering the whole-brain of each participant. Nuisance variables were regressed out from the analysis and coefficients were z-transformed. The second-level analysis was assessed by applying a random effects model. A height threshold of p-uncorrected <0.001 (two-sided) and p-FDR <0.05 corrected at the cluster level were applied to assess all of the seed regions. Unlike the effects found with MTL and larger hippocampal lesions (*Hayes et al., 2012*; *Henson et al., 2016*; *Rudebeck et al., 2013*), two sample independent *t*-tests revealed that functional connectivity of the left and right hippocampal seed-regions with the rest of the brain were not significantly different between the amnesic group and control group, when assessed at a corrected p-FDR <0.05 threshold. The additional analyses were consistent with these results, because there were no significant differences in functional connectivity between the amnesic group and the control group for the seed regions in the visual network and somatomotor network with the rest of the brain, when assessed at a corrected p-FDR <0.05 threshold. For completeness, we also report between-group clusters at an uncorrected cluster-size threshold (p-uncorrected <0.05; height threshold p-uncorrected <0.001) in *Supplementary file 1o* and provide a plot of the between-group seed-to-voxel correlation map at an uncorrected cluster-size threshold (p-uncorrected <0.05; height threshold, p-uncorrected <0.001) in *Figure 7—figure supplement 1*. In addition, significant clusters thresholded at p-FDR <0.05 are reported separately for the left and right hippocampal seeds as a function of each group in *Supplementary file 1p,* and corresponding group-wise seed-to-voxel correlation maps are plotted in *Figure 7—figure supplement 2*.

Next, for the ROI-to-ROI analyses, we tested whether there were significant group differences in functional connectivity between the hippocampus and regions in the DN that usually exhibit functional coupling with the hippocampus. Accordingly, we selected a ROI in the dmPFC, because memory-guided behavior is supported by interactions between the hippocampus and dmPFC (*Shin and Jadhav, 2016*). A ROI in the vmPFC was selected because functional coupling between the hippocampus and vmPFC supports various stages of autobiographical memory processing (*Barry and Maguire, 2019*; *Eichenbaum, 2017*; *McCormick et al., 2018a*). Finally, we examined functional connectivity between the hippocampus and the PCC because it is a core hub of the DN and because posterior midline cortical regions such as the PCC support the successful retrieval of autobiographical memories (*Addis et al., 2004a*; *Ryan et al., 2001*; *Svoboda et al., 2006*). MNI co-ordinates for the dmPFC, vmPFC, left PCC, and right PCC correspond to those used in the graph theoretic analyses of functional connectivity. MNI co-ordinates for the left and right hippocampus ROIs were the same as those used in the seed-to-voxel analyses. On the grounds that we were specifically interested in the effects of bilateral hippocampal damage on functional connectivity, we investigated the connectivity of the left and right hippocampus with all of the other ROIs. Accordingly, temporal correlations were calculated for the left hippocampus seed with the dmPFC, vmPFC, left PCC, and right PCC ROIs and corresponding pairings were calculated for the right hippocampus seed. Normalized correlation coefficients (Fisher's z-transformation) were entered into a between-group *t*-test, assessed at a two-sided p-FDR <0.05 criterion (seed-level correction).

Results for the left hippocampus seed ROI revealed that there were no significant between-group differences in functional connectivity with the left PCC ($t_{(27)}$ = 0.77, p-FDR = 0.225), the right PCC ($t_{(27)}$ = 0.86, p-FDR = 0.198), the dmPFC ($t_{(27)}$ = 0.29, p-FDR = 0.387), and the vmPFC ($t_{(27)}$ = –0.36, p-FDR = 0.637). Results for the right hippocampus seed ROI revealed that there were no significant between-group differences in functional connectivity with the right PCC ($t_{(27)}$ = 1.50,

p-FDR = 0.073), the dmPFC ($t_{(27)}$ = 1.08, p-FDR = 0.145), and the vmPFC ($t_{(27)}$ = 1.16, p-FDR = 0.129). A significant group difference in functional connectivity was found between the right hippocampus seed ROI and the left PCC ($t_{(27)}$ = 1.75, p-FDR = 0.046). Hence, the only region that exhibited a between-group difference in functional connectivity with the hippocampus was the left PCC, but this difference did not survive Holm-Bonferroni correction for multiple comparisons.

## A loss of integration in MTL subsystem regions was predictive of autobiographical episodic retrieval

If the loss of integration in the affected DN nodes and CA3 volume are relevant for episodic memory, then these variables may be associated with the internal (episodic) detail remembered on the AI. First, we examined the association between CA3 volume and total internal (episodic) detail. At a single group level, the correlation between total internal detail and CA3 volume was not significant either for the amnesic group (Kendall's $τ_{(15)}$ = −0.018, p=0.961, two-tailed) nor for the control group (Kendall's $τ_{(15)}$ = 0.43, p=0.458, two-tailed; *Figure 4—figure supplement 1*). CA3 volume and total internal detail were significantly correlated (Kendall's $τ_{(30)}$ = 0.283, p=0.028, two-tailed; *Figure 4—figure supplement 1*), such that lower CA3 volumes were associated with remembering less internal detail on the AI, when these variables were collapsed across group.

Second, we evaluated the association between the nodes that exhibited between-group differences in topology and internal (episodic) detail on the AI. A robust multiple regression based analysis was performed, using Huber's method of correction for outliers on the difference in average path length of affected nodes from the mean of the control group and total internal detail (collapsed across five time points of the standard administration). Left and right PHC, left hippocampal formation and right hippocampal formation, left retrosplenial cortex, vmPFC, and left temporal pole were entered as independent variables, whereas total internal detail was entered as a dependent variable. Differences in average path length from the mean of the control group in the left PHC ($β$ = 1.05, $t$ = 4.69, p=0.018), left hippocampal formation and right hippocampal formation ($β$ = 1.81, $t$ = 7.87, p=0.004 and $β$ = −1.55, $t$ = −8.66, p=0.003, respectively), left retrosplenial cortex ($β$ = 26.61, $t$ = 12.55, p=0.001), vmPFC ($β$ = −26.31, $t$ = −11.55, p=0.001), and the left temporal pole ($β$ = −1.70, $t$ = −10.26, p=0.002) significantly predicted the quantity of total internal (episodic) detail remembered by the amnesic group participants on the AI. Right PHC was not predictive of the quantity of internal detail that was remembered (see *Supplementary file 1q*).

Third, a robust multiple regression with Huber's method of correction for outliers was performed on the difference between amnesic group participants from the mean of the control group in the local efficiency of affected nodes and total internal (episodic) detail (see *Supplementary file 1r*). Left PHC, right retrosplenial cortex, and left posterior cingulate cortex were entered as independent variables, whereas total internal detail for each amnesic group participant was entered as a dependent variable. Differences in local efficiency at each of these three nodes from the mean of the control group did not significantly predict the quantity of internal (episodic) detail that was remembered by the amnesic group participants (*Supplementary file 1r*).

## Discussion

Episodic memory is dependent on a large-scale hippocampal-neocortical network of regions (*Káli and Dayan, 2004*; *McClelland et al., 1995*; *Wang and Morris, 2010*; *Winocur et al., 2010*). In the current study, we hypothesized that damage to human CA3 would impair both recent and remote episodic memories for personal events and would affect the topological properties of brain regions implicated autobiographical memory. Hence, both anatomical damage and alterations in functional network topology were hypothesized to be relevant for autobiographical episodic memory performance on the AI. Results from the participants with bilateral damage to CA3 revealed a time-sensitive loss of internal (episodic) detail: the earliest remote memory (0–11 years) was intact, whereas all other remote memories and memory for an event from the past year exhibited a loss of internal (episodic) detail, yet the personal semantic content and narrative structure of these same memories were comparable to those observed in the control group. Hence, a complete loss of hippocampal neurons and/or extensive MTL damage are not necessary conditions for the impairment of both recent and very remote episodic memories. The loss of episodic detail for recent and remote memories (i.e., for all but the earliest remote memory) is consistent with neurobiological

accounts that predict damage to the human hippocampus affects retrieval for as long as the memory retains vivid, episodic detail (*Barry and Maguire, 2019*; *Eichenbaum et al., 2007*; *Hassabis and Maguire, 2007*; *Moscovitch et al., 2016*; *Nadel and Moscovitch, 1997*; *Winocur et al., 2007*; *Yonelinas et al., 2019*). The preservation of the earliest remote memories is consistent with the contention that these memories are remembered more frequently, re-encoded, and can be supported by neocortical representations (*Sekeres et al., 2018*). Second, CA3 damage was associated with a loss of integration, as evinced by an increase in average path length, in nodes of the MTL subsystem of the DN and in the left temporal pole. Alterations in local efficiency were observed in two nodes within the MTL subsystem and in the left posterior cingulate cortex, whereas the other graph theoretic measures were unaffected. Third, perturbations of functional integration in the left parahippocampal cortex, left hippocampal formation, right hippocampal formation, left retrosplenial cortex, vmPFC, and left temporal pole were predictive of the amount of internal (episodic) detail remembered by amnesic group participants, whereas the CA3 volumes of amnesic group participants and the differences in local efficiency were not predictive of the amount of episodic detail remembered. These three main results are addressed in turn.

## Recent and remote memories were disrupted by CA3 damage

The observed dependence of remote episodic memories on human CA3 is inconsistent with lesion and molecular imaging studies of model organisms that implicate CA3 in the retrieval of recent but not remote episodic-like memories (*Daumas et al., 2005*; *Hasselmo, 2005*; *Hunsaker and Kesner, 2008*; *Lee et al., 2005*; *Lux et al., 2016*). Many of these studies instead identified CA1 as being involved in remote retrieval and consolidation, which align with the evidence of a link between amnesia for recent and remote autobiographical memories and damage to human CA1 (*Bartsch et al., 2011*). The discrepancy between our results and those from model organisms may reflect the use of behavioral tasks in model organisms that are based on episodic-like retrieval of associations between an object and the location (where) and/or the occasion (when) it was last encountered, which may not be equivalent to autobiographical episodic retrieval in humans (*Hardt and Nadel, 2018*). In addition, memories involving the context of anxiety or fear conditioning can be expressed without hippocampal involvement at remote time points (*Kim and Fanselow, 1992*), and may lack the qualitative features of autobiographical episodic memory (*Hardt and Nadel, 2018*). Remote memories remain hippocampal-dependent in mice when detailed context fear memories are assessed (*Wiltgen et al., 2010*), whereas in humans, hippocampal and CA3/DG activity measured using fMRI have each been associated with the retrieval of remote autobiographical episodic detail across a wide range of intervals (mean range = −2.5 days to −32.3 years) (*Bonnici et al., 2012*; *Gilboa et al., 2004*; *Rekkas and Constable, 2005*; *Söderlund et al., 2012*; *Steinvorth et al., 2006*; *Viard et al., 2007*).

Hippocampal area CA3 supports computations that distinguish between specific experiences (pattern separation) or that code relationships across items to increase overlap (pattern completion) (*Aly and Turk-Browne, 2016*; *Guzman et al., 2016*; *Lee et al., 2004*; *Leutgeb et al., 2004*; *McClelland et al., 1995*; *Norman and O'Reilly, 2003*; *Treves and Rolls, 1994*; *Yassa and Stark, 2011*), depending on the degree of similarity or dissimilarity between contexts or learning materials (*Guzowski et al., 2004*). Impaired recall of detailed episodic memories in the amnesic group could thus, in part, reflect a diminution in the ability of these participants to discriminate between individual mnemonic patterns and/or to track the precision of episodic retrieval for specific events with partial cueing (*Chadwick et al., 2014*). Notably, the observed deficit was specific, because the loss of internal (episodic) detail was expressed under conditions where personal semantic (external) detail was intact relative to that in the control group, which aligns with the effects of bilateral and unilateral temporal lobe resection (*Noulhiane et al., 2007*; *Viskontas et al., 2000*) and other results reported in studies of hippocampal damage (*Taylor et al., 2007*). It is also worth noting that the impact of deficits in these computations on remote and recent episodic detail retrieval may not be equivalent, because remote memories are likely to have undergone more extensive reconsolidation and reconstruction (*Barry and Maguire, 2019*; *Dudai and Morris, 2013*; *Nader and Hardt, 2009*; *Tronson and Taylor, 2007*; *Wang and Morris, 2010*).

Lesion studies of the hippocampus (hippocampus proper along with the DG and subiculum) and adjacent MTL are associated with loss that is either restricted to recent rather than remote episodic memories (*Bayley et al., 2005*; *Bontempi et al., 1999*; *Dede et al., 2016*; *Kapur and Brooks,*

*1999*; *Kim and Fanselow, 1992*; *Kirwan et al., 2008*; *Squire and Bayley, 2007*; *Takashima et al., 2009*) or consistent with extensive amnesia, independent of remoteness (*Eichenbaum et al., 2007*; *Hassabis and Maguire, 2007*; *Moscovitch et al., 2016*; *Nadel and Moscovitch, 1997*; *Noulhiane et al., 2007*; *Steinvorth et al., 2005*; *Winocur et al., 2007*). In case reports of focal subfield damage, retrograde amnesia in two individuals with bilateral CA1 damage was limited to 1–2 years (*Rempel-Clower et al., 1996*; *Zola-Morgan et al., 1989*). By contrast, retrograde amnesia up to ~60 years has been observed post onset of damage to human CA1, secondary to transient global amnesia (lasting, 8.3 ± 1.9 hr) (*Bartsch et al., 2011*). The duration of amnesia observed here – spanning up to ~50 years prior to the CA3 damage – is compatible with theoretical accounts that implicate the hippocampus in retrieval for as long as the memory retains spatial detail and context-specific episodic content (*Barry and Maguire, 2019*; *Maguire and Mullally, 2013*; *Moscovitch et al., 2016*; *Moscovitch et al., 2005*; *Winocur et al., 2010*; *Yonelinas et al., 2019*). Notably, although remote autobiographical episodic memories are likely to undergo more reconstruction and thus reconsolidation than recent memories, CA3 damage did not affect remote memories to a greater extent.

## Altered topology in the default network

The effects of the CA3 damage on DN topology may help to explain the notional gap between the small, by volume, bilateral subfield damage observed here and persistent episodic memory loss. Average path length was increased, similar to lower global efficiency, in the left and right parahippocampal cortex, the left and right hippocampal formation, vmPFC, and the left retrosplenial cortex of the MTL subsystem and the left temporal pole of the dmPFC subsystem. The reduction in functional integration — that is, the capacity to transmit information with less attenuation — in the amnesic group aligns with the impact of damage to hub regions, such as the hippocampus, having a disproportionate effect on global efficiency (*Hwang et al., 2013 Albert et al., 2000*). Parahippocampal cortices support context reinstatement (*Diana et al., 2013*) and remote memory (*Lux et al., 2016*), thus the loss of integration in parahippocampal cortices is likely to have interfered with the operations necessary for remembering episodic detail. Integration between the affected regions and other parts of the core autobiographical network is particularly important for successful episodic retrieval (*Westphal et al., 2017*), and is likely to impact on the capacity of hippocampal-neocortical ensembles to support the reconstruction and elaboration that underscores autobiographical episodic retrieval (*Barry and Maguire, 2019*; *Greenberg et al., 2005a*; *Piolino et al., 2009*; *St-Laurent et al., 2014*). For example, functional coupling between the hippocampus and retrosplenial cortex has been implicated in the initial reconstruction of the episodic context and detail as a memory is selected and accessed (*Inman et al., 2018*), and supports the consolidation of events (*Bird et al., 2015*; *Staresina et al., 2013*).

Local efficiency of the left posterior cingulate cortex (midline core), left parahippocampal cortex, and right retrosplenial cortex nodes were increased in the amnesic group compared to the control group, which suggests greater capacity to process information (notably, the result at the left posterior cingulate cortex was not robust). Modulation of activity in these regions is associated with the regulation of learning, consolidation, and retrieval. The hippocampus and posterior cingulate cortex are anatomically connected to one another (*Daselaar et al., 2008*; *Kobayashi and Amaral, 2003*), as part of a core retrieval network (*Rugg and Vilberg, 2013*), and damage to the retrosplenial cortex region of the posterior cingulate cortex can lead to deficits that are similar to those associated with MTL damage (*Philippi et al., 2015*; *Valenstein et al., 1987*). Increases in local efficiency may alter connectivity with immediately surrounding regions that support the regulation of learning, consolidation, and retrieval, and could reflect adaptive functional reorganization, possibly involving the reassignment of nodal roles. In model organisms, the inhibition of rodent CA1 can lead to selective compensatory changes in anterior cingulate cortex activity, which are associated with remote contextual memory (*Goshen et al., 2011*). Of note, however, the local efficiency differences in the three affected regions were not associated with the retrieval of episodic detail on the AI. Increases in local efficiency of between 85% and 270% have been observed in structural connectivity studies of left temporal lobe epilepsy, and attributed to a compensatory (perhaps maladaptive) mechanism that maintains connectivity despite the loss of connections in hub regions (*DeSalvo et al., 2014*). For the most part, however, the relationship between functional and structural graph theoretic connectivity measures is not yet well established.

The effects of hippocampal and MTL damage, which are observed across mixed aetiologies, on RSNs have variable scalar extents (*Hayes et al., 2012*; *Heine et al., 2018*; *Henson et al., 2016*; *Rudebeck et al., 2013*). Here, graph theoretic analyses of salience, ventral attention, dorsal attention, somatomotor, and visual RSNs revealed null between-group differences in topology. Only single nodes were affected when a liberal uncorrected threshold was applied to the analyses of these networks, and these nodes did not survive false-discovery rate correction. These results notionally align with how deficits in the amnesic group on standardized neuropsychological assessment did not extend beyond tests of memory. Also in line with the observed scalar extent of altered network topology is evidence that simulated lesions involving hub regions can lead to non-extensive effects on topological organization, whereas damage to regions that connect different network modules (connectors) can result in widespread effects that affect whole-brain network organization (*Gratton et al., 2012*; *He et al., 2009*; *Honey and Sporns, 2008*). Larger-scale effects on graph theoretic measures of functional connectivity have, however, been reported following more extensive damage (*Henson et al., 2016*). In particular, *Henson et al. (2016)* found that functional connectivity was altered in DN, thalamic, and precuneus RSNs and that modularity (clustering and smallworldness) was increased across whole-brain networks (*Henson et al., 2016*), following extra-hippocampal damage to amygdala (3/6 patients), entorhinal and parahippocampal cortices (2–3/6 patients), and in a participant with deficits on nonverbal reasoning, attention, and executive function.

The whole-brain seed-to-voxel analyses revealed null between-group differences in functional connectivity for the left and right hippocampus seed regions, and the seed regions in visual and motor cortices. The ROI-to-ROI analyses revealed that functional coupling was altered only between the right hippocampus and the left posterior cingulate cortex, but this difference did not survive correction for multiple comparisons. Alterations in connectivity between the hippocampus and the posterior cingulate cortex are likely to disrupt autobiographical memory, because functional connectivity between the hippocampus and the posterior cingulate cortex has been implicated in episodic autobiographical remembering (*Sheldon et al., 2016*; *Sheldon and Levine, 2013*). More extensive damage to the hippocampus, parahippocampal gyrus, and temporal pole can lead to significant differences in functional connectivity that are limited to the DN (*Hayes et al., 2012*). Other studies, however, have observed dysfunction in multiple large-scale brain networks following hippocampal damage, such that functional connectivity was decreased in the salience network and increased in the dorsal and ventral DN, sensorimotor network, and higher visual networks relative to that in control participants (*Heine et al., 2018*). These alterations in functional connectivity co-occurred with more extensive behavioral deficits, as compared to those reported here, in episodic and working memory, verbal and visual learning, semantic fluency, and executive function (*Heine et al., 2018*). Variability in the outcomes may also reflect that the correlation-strength map of voxels generated by seed-based approaches is particular to the average time-series correlation with the seed region under analysis. Another consideration is that seed-based analyses do not assess network connections between ROIs simultaneously, which is necessary when studying functional integration and segregation at the scale of RSNs (*van den Heuvel and Hulshoff Pol, 2010*). More generally, because functional connectivity can reflect signaling that unfolds within the underlying structural network (*Betzel et al., 2013*; *Goñi et al., 2014*), future studies will need to examine anatomical pathways. It is important to note, however, that white matter connectivity appears to be unaffected by LGI1-antibody-complex LE (*Finke et al., 2017*).

## Associations between CA3 volume and altered topology with autobiographical episodic memory performance

CA3 volume was not correlated with the amount of internal (episodic) detail remembered by either the amnesic group or the control group. One possible interpretation is that sample size and variability of the data may account for the failure to detect a significant association at the group level, because CA3 volume and internal detail were positively correlated when considered as continuous distributions by collapsing across the two groups. CA3 along with CA2 and the DG are input structures, typically associated with encoding, whereas CA1 is an output structure, implicated in the retrieval of events within their temporal context (*Preston et al., 2010*; *Suthana et al., 2011*; *Zeineh et al., 2003*). In the study by *Bartsch et al. (2011)*, involving transient global amnesia, a link between the residual CA1 volume and autobiographical episodic retrieval was not reported. In

healthy adult participants, there has been a failure to find an association between CA1 volume and episodic retrieval (*Mueller et al., 2011*), whereas CA3 volume correlates with the efficacy with which newly formed memories are differentiated (*Chadwick et al., 2014*). The latter effect was interpreted to reflect either decreases in retrieval confusion that results from an increased number of CA3 neurons or enhanced lateral connectivity which could lead to improvements in pattern separation (*Chadwick et al., 2014*). fMRI studies based on laboratory-learned materials suggest that time and space context are represented across multiple human hippocampal subfields (*Copara et al., 2014*), with differentiation between spatial and temporal context mediated by distinct neural network patterns (i.e., multiplexed) rather than by individual structures (*Kyle et al., 2015*). Hence, links between single subfield volumes and autobiographical episodic retrieval remain challenging to interpret. Volumetric analyses that collapse across larger subregions have found that, for example, residual bilateral MTL volume following brain injury correlates with remote autobiographical memory (*Gilboa et al., 2005*), whereas in a study on participants who underwent temporal lobe resection for epilepsy, right parahippocampal cortical volumes correlated with that capacity to remember remote episodes and bilateral MTL regions predicted memory for recent episodes (*Noulhiane et al., 2007*).

The relevance of the between-group differences in DN topology for the retrieval of internal (episodic) detail were also assessed. Specific perturbations of the functional network topology of the DN were found to predict the amount of episodic detail remembered by the amnesic group. In particular, differences in average path length from the control group in the left parahippocampal cortex, left and right hippocampal formation, left retrosplenial cortex, vmPFC, and left temporal pole were predictive of the amount of episodic detail remembered by amnesic group participants. Increases in the average path length of the right hippocampal formation, vmPFC, and left temporal pole, consistent with a loss of functional integration, were associated with deficits in autobiographical episodic retrieval. vmPFC-hippocampal interactions integrate remote memories with other stored information during retrieval (*Eichenbaum, 2017*; *Preston and Eichenbaum, 2013*). However, the results for the left hippocampal formation, left parahippocampal cortex, and left retrosplenial cortex are again challenging to interpret in a principled manner, given that functional integration is important for remembering internal detail. On face value, this result may reflect aberrant functional connectivity or functional reorganization. As noted, other studies of hippocampal and MTL amnesia have reported positive and negative associations between measures of functional connectivity strength and episodic memory (*Heine et al., 2018*). It will be important to reconcile the results across different studies in order develop a network model of cognitive (dys)function in hippocampal amnesia.

## Concluding remarks

Human CA3 was found to be necessary for the retrieval of internal (episodic) but not external (non-episodic, semantic) detail related to both recent and remote memories. Contrary to the duration of hippocampal involvement predicted by systems consolidation and computational-based theories of episodic memory (*Rolls et al., 1997*; *Squire and Bayley, 2007*), amnesia was observed for episodes that occurred up to ~50 years prior to focal CA3 damage. Furthermore, with the exception of the earliest intact remote memory, CA3 involvement in episodic retrieval did not become less important with increasing remoteness. The duration of CA3 involvement in autobiographical episodic retrieval is at variance with the evidence from model organisms that suggests that remote memories depend on CA1 but not CA3 (*Denny et al., 2014*; *Guzman et al., 2016*; *Kesner and Rolls, 2015*; *Leutgeb et al., 2007*; *Lisman, 1999*; *Lux et al., 2016*; *McNaughton and Morris, 1987*; *Rebola et al., 2017*). CA3 damage is likely to impair normal engagement with neocortical episodic memory traces, because CA3 enables signals arising from different brain regions to be associated together (*Kesner and Rolls, 2015*). Human CA3 may be a default activator of remote episodic memories that participates in their maintenance throughout recall (*Goshen et al., 2011*). Episodic retrieval performance was predicted by the perturbation of specific topological properties of nodes within the MTL subsystem of DN; these affected nodes overlapped with key regions of the core autobiographical network. The current results also extend experimental evidence on the neural basis of episodic memory to highlight the important role of integration in the MTL subsystem (*Geib et al., 2017*; *Westphal et al., 2017*). Future work will need to hone in on how intrinsic hippocampal subfield-based computations contribute to autobiographical memory and how subfield involvement in episodic memory changes with use, especially because different multiscale brain circuits are

recruited during recent and remote memory (*Barry and Maguire, 2019*; *Dudai and Morris, 2013*; *Nader and Hardt, 2009*; *Tronson and Taylor, 2007*).

## Materials and methods

### Participants

The amnesic group was comprised of 16 participants (mean and S.E.M. age: 64.2 ± 4.81 years, female = 3). All participants in the amnesic group had chronic amnesia induced by a single-aetiology, leucine-rich glycine-inactivate-1 antibody-complex limbic encephalitis (LGI1-antibody-complex LE). Fifteen of the participants were LGI1-antibody positive and one participant was LG1I and CASPR2 negative but VGKC-complex antibody positive. LGI1-antibody-complex LE typically presents with amnesia and seizures (*Dalmau and Rosenfeld, 2014*), and leads to non-reversible chronic atrophy that is confined to the hippocampus (*Finke et al., 2017*; *Irani et al., 2013*; *Miller et al., 2017*; *Wagner et al., 2015b*). All participants were seizure-free at the time of testing. In model organisms, LGI1 is predominantly expressed in CA3 and DG subfields (*Herranz-Pérez et al., 2010*). The highly selective bilateral CA3 lesions align with evidence from rodent models that have also revealed neuronal loss in CA3 (*Chabrol et al., 2010*), probably caused by excitotoxic lesions (given that IgG-containing LGI1 antibodies induce population epileptiform discharges in CA3 pyramidal neurons in vitro [*Lalic et al., 2011*]), or by complement-mediated fixation of bound antibodies (*Bien et al., 2012*).

All of the amnesic group participants were considered clinically stable by their consultant neurologist (median = 4 years post-onset, range = 7), and were thus discharged and not undergoing treatment. All participants with amnesia were otherwise self-reported as healthy, with no evidence of secondary gain or active psychopathology. Unlike studies of amnesia involving participants with chronic conditions such as epilepsy that are associated with seizures and hippocampal sclerosis (*Kapur and Prevett, 2003*), the known timing, discrete onset, and monophasic nature of autoimmune encephalitis suggest that hippocampal function was intact in these participants during the encoding of memories before the disease onset. Hence, impaired retrograde memory probably reflects disruption of CA3-mediated retrieval mechanisms rather than anterograde difficulties with encoding and consolidation.

Sixteen healthy age- and education-matched controls were recruited (62.3 ± 3.23 years, female = 2) to the control group. These participants had no history of cognitive, psychiatric, or neurological illness, and were not taking psychoactive medications. With the exception of one participant in the amnesic group who had reduced visual fields, secondary to age-related macular degeneration, all participants had normal or corrected-to-normal vision. All participants were fluent, native English speakers. No significant differences were found between the groups in terms of age or years-of-education. The sample size was selected to be similar to comparable studies in the literature. Informed written consent was obtained from all participants for all procedures and for consent to publish, in accordance with the terms of approval granted by the local research ethics committee and the principles expressed in the Declaration of Helsinki.

### 7.0-Tesla magnetic resonance image acquisition and protocols

All 16 participants in the amnesic group and 15 of the 16 control participants underwent an ultrahigh resolution anatomical neuroimaging protocol and a functional neuroimaging protocol performed using a 7.0-Tesla whole-body MR scanner (Achieva, Koninklijke Philips Electronics), based at the Sir Peter Mansfield Magnetic Resonance Centre, School of Physics and Astronomy, University of Nottingham, operated with a volume-transmit 32-element receive whole-head coil array (Nova Medical, Inc, Wilmington, MA, USA). Information obtained from the 7.0-Tesla MRI anatomical neuroimaging, concerning the nature of the CA3 lesions in the participants in the amnesic group and the absence of damage in the control group has been previously reported (*Miller et al., 2017*).

### Anatomical MRI acquisition

As reported in our previous study (*Miller et al., 2017*), two principal anatomical sequences were acquired to conduct: (a) bilateral quantitative 3-D morphometry of cornu ammonis (CA) subfields 1–3, DG, and subiculum on whole-hippocampal images (a partial volume was focused on the

hippocampi and acquired at 0.39 x 0.39 x 1.0 mm$^3$ spatial resolution); and (b) whole-brain voxel-by-voxel based morphometry (VBM) (a whole-brain T$_1$ was acquired at 0.6 mm$^3$ isotropic spatial resolution).

In particular, anatomical MRI data acquisition involved the followings sequences: (1) Initial parasagittal localizer images in three orthogonal orientations were acquired to verify head position and to guide acquisition, so that the oblique coronal volume of interest could be oriented perpendicular to the anterior-posterior axis of the hippocampus. (2) RF-refocusing sequence optimized for heavy contrast, which provided a three-dimensional T$_2$-weighted fast spin-echo image (0.39 × 0.39 mm$^2$, in plane × 1.0 mm, slice thickness, resolution) in 52 contiguous oblique coronal sections (perpendicular to hippocampal axis), with coverage of both hippocampi. T$_2$-weighting provided the necessary contrast between white and gray matter, allowing visualization of the white matter bands between the CA and DG. (3) Three-dimensional whole-brain T$_1$-weighted magnetization-prepared rapid acquisition gradient-echo images (0.6 × 0.6 × 0.6 mm$^3$ resolution) and non-prepared 3-D images to correct T$_1$-weighted images for B$_0$ intensity field bias, with the following parameters: 176 slices; resolution = 256 × 256; voxel size = 1 mm × 1 mm × 1 mm; time repetition = 1900 ms; time echo = 2.2 ms; flip angle = 9°. These sagittal T$_1$-weighted images provided information on global brain morphology, which enabled us to derive intracranial volume (*Mathalon et al., 1993*; *Nordenskjöld et al., 2013*). These images were also used to guide the 3-D FSE imaging planes, as shown in *Figure 1*, provided the basis for whole-brain voxel-by-voxel based morphometry, and were used to perform anatomical normalization of the EPI sequence.

## Resting-state functional MRI acquisition

Resting-state (i.e., task-free) functional MRI (rs-fMRI) data were also acquired for the amnesic and control group participants alongside a high-resolution T$_1$-weighted and a 3-D FSE sequence. Each rs-fMRI scan required that the participants lay supine in the scanner for 8 min. Head movements were limited by the use of foam padding and an inflatable cuff. Participants were fitted with ear plugs and instructed to remain still, keep their eyes open while viewing a blank black screen, and allow their mind to wander and not think about anything systematically, but not to fall asleep. rs-fMRI data (200 volumes/participant) were acquired using a whole-brain echoplanar imaging (EPI) pulse sequence sensitive to blood oxygen-level–dependent (BOLD) contrast, with the following parameters: TR = 2500 s, TE = 25 ms, acquired resolution, isotropic voxel size = 2 × 2 × 2 mm$^3$, on a base matrix of a x b pixels; field-of-view 192 × 150 × 120 mm, flip angle = 90°, and 60 slices covering the whole-brain, with a slice thickness of 2 mm and 0 interslice gap. EPI scans were oriented to intersect the anterior and posterior commissures.

## MRI analysis pipelines
### Volumetric measures
#### Hippocampal subfield quantitative morphometry

The hippocampal subfield segmentation protocol has been described in our prior study that reported the results from 18 participants with LGI1-antibody-complex LE (*Miller et al., 2017*). Briefly, manual segmentation was performed on the three-dimensional fast-spin echo images using the freehand spline drawing and manual segmentation tools in ITK-SNAP 3.2 (http://www.itksnap.org) (*Yushkevich et al., 2006*). Hippocampal delineation and segmentation were performed on images at native resolution (0.39 × 0.39 × 1 mm$^3$), in a coronal orientation and in the anterior-posterior direction. Quantitative hippocampal subfield morphometry was conducted along the whole left and right hippocampus of participants in the amnesic and control groups, guided by a previously described 7.0-Tesla manual segmentation protocol (*Figure 1*) (*Wisse et al., 2012*), which was modified so that CA2 and the boundary between CA1 and the subiculum were additionally delineated on each slice (*Miller et al., 2017*). Five hippocampal subfields — CA1, CA2, CA3, dentate gyrus, and the subiculum — were identified, thereby deriving 10 volumes (mm$^3$). All hippocampi were fully re-segmented after a month had elapsed in order to evaluate intra-rater reliability, and a subset of amnesic group participant and control participant scans also underwent manual segmentation by a second rater to generate inter-rater reliability indices. Intra-rater and inter-rater reliabilities were calculated using the Dice overlap metric applied across all segmented slices (*Dice, 1945*). Both raters were blinded to the identity of all scans and to the behavioral data.

## Whole-brain voxel-by-voxel brain morphometry

The high-resolution whole-brain $T_1$-weighted anatomical images were used as the basis for whole-brain voxel-by-voxel based morphometry (VBM) and diffeomorphic anatomical registration using an exponentiated Lie algebra (DARTEL) registration method (*Ashburner and Friston, 2009*) (*Figure 3*). The automated VBM analysis was performed using SPM12 (Statistical Parametric Mapping, Wellcome Trust Centre, London, UK; www.fil.ion.ac.uk/spm). $T_1$-weighted anatomical images were first segmented into gray matter (GM), white matter (WM), and cerebrospinal fluid (CSF), as well as into three extra-cerebral tissue classes. Inter-subject iterative registration of the gray and white matter segments was performed using the Dartel toolbox (*Ashburner, 2007*). SPM12 did not register one amnesic group participant, so the scan was removed from further VBM analyses. Gray and white matter maps were normalized to the gray matter population-specific template generated from the complete image set using the DARTEL toolbox. The DARTEL template and deformations were used to normalize gray and white matter probability maps to Montreal Neurological Institute (MNI) stereotactic space, preserving the total amount of signal from each region in the images (i.e., modulation), modulated by the Jacobian determinants derived from the spatial normalization and smoothed with a full-width at half-maximum kernel of $8 \times 8 \times 8$ mm$^3$. We excluded voxels with gray matter values < 0.2 (absolute threshold masking) to avoid edge effects between the tissue types. Global GM volumes for each participant were calculated as the mean value of the voxels within all regions, including, for example, voxels in regions that are adjacent to the hippocampus and other sites enriched in LGI1, such as those reported in a study examining the anatomical localization of gene transcripts of the LGI1 family (*Herranz-Pérez et al., 2010*). These gray matter volumes from participants in the amnesic and control groups were contrasted using a two-sample *t*-test and thresholded at $p<0.05$, with family-wise error correction and a cluster extent of 50 voxels. Total intracranial volumes were included in the model as a covariate of no interest.

Concerns about pathology elsewhere in the brain are not confined to autoimmune encephalitis. Other aetiologies that can lead to hippocampal-mediated amnesia, such as viral encephalitis, hypoxic brain injury secondary to drug overdose, or toxic shock syndrome, are associated with circumscribed hippocampal lesions, but frequently also involve anatomical damage elsewhere (*Heinz and Rollnik, 2015*; *Raschilas et al., 2002*). In addition, these aetiologies lead to co-morbidities and broader cognitive impairment (*Heinz and Rollnik, 2015*; *Hokkanen and Launes, 2007*; *Peskine et al., 2010*; *Thakur et al., 2013*), which were absent from the clinical and neuropsychological profiles of the amnesic group participants that are reported here. Many previous studies have reported patients with circumscribed hippocampal lesions, but have not reported results from whole-brain voxel-by-voxel morphometry or techniques based on related whole-brain segmentation.

## Measurement of intracranial volumes

Total intracranial volumes (TIVs) were derived by applying the sequence of unified segmentation, as implemented in SPM12 (*Malone et al., 2015*), to the $T_1$-weighted 7.0-Telsa images of each participant in order to normalize for inter-participant variation and premorbid head size. Adjustments for TIV also increased the power to detect between-group differences in hippocampal volume (*Nordenskjöld et al., 2013*). Individual raw volumes for each hippocampus were normalized to TIV (*Jack et al., 1992*; *Lehéricy et al., 1994*).

## Functional connectivity measures

### rs-fMRI pre-processing

4-D rs-fMRI images were pre-processed using SPM12 (Statistical Parametric Mapping, Wellcome Trust Centre, London, UK; www.fil.ion.ac.uk/spm), and the default pre-processing steps for volume-based analysis (to Montreal Neurological Institute ((MNI)-standard space) were implemented within the functional connectivity toolbox for correlated and anticorrelated brain networks (CONN) on the MATLAB platform ((v.17.f); https://www.nitrc.org/projects/conn/) (*Whitfield-Gabrieli and Nieto-Castanon, 2012*). The first six volumes were removed to allow the fMRI signal to stabilize and to ensure magnetization equilibrium. Quality assurance involved checks for artifacts in both volume and slice-to-slice variance in the global signal.

Pre-processing of functional images from each participant involved: (1) Realignment, unwarping, and slice-timing correction. (2) Co-registration of the EPI scans to the $T_1$-weighted anatomical scan.

(3) Segmentation of co-registered T$_1$ images into white and gray matter and cerebrospinal fluid (CSF), for use with the diffeomorphic anatomic registration through an exponentiated lie algebra algorithm (DARTEL) toolbox to create structural templates and individual flow fields. The latter were used for normalization of the structural and segmented functional images to the MNI standard stereotaxic anatomical space (MNI-152). (4) Spatial smoothing with a 6.0 mm isotropic full-width at half-maximum isotropic Gaussian kernel. (5) Outlier detection of global signal and motion with the artifact detection toolbox (ART-based scrubbing). fMRI data from one participant in the amnesic group were excluded due to poor quality and a failure to complete all steps of the pre-processing.

## Motion

Even small movement artifacts can contaminate estimates of functional connectivity (*Muschelli et al., 2014*; *Power et al., 2012*). Motion was minimal across all the remaining participants, and all participants included in the final analyses met the minimum mean framewise displacement (i.e., the sum across all volumes of six possible motion parameters (x, y, z, roll, pitch, yaw)), estimated from the reference (i.e., realignment volume) threshold (< 0.5 mm) for inclusion. Movement parameters were compared between the amnesic and control groups using two-tailed *t*-tests and revealed no significant between-group differences in mean framewise displacement, with amnesic and control participants exhibiting means of 0.23 mm (s.e.m. = 0.07 mm) and 0.26 mm (s.e.m. = 0.07 mm), respectively ($t_{(27)}$ = 0.570, p=0.573). There were also no significant correlations between altered topology and mean framewise displacement (p-values >0.4). Functional connectivity analyses were performed on ART 'scrubbed' data and included a realignment based motion correction (six rigid-body) parameter. No significant between-group difference was evident in the proportion of scrubbed volumes. Residual motion effects were also addressed via an anatomical component-based noise correction (aCompCor) method (*Behzadi et al., 2007*) (see below).

## Functional connectivity pipeline

Functional connectivity analyses were conducted using the CONN toolbox (http://www.nitrc.org/projects/conn) (*Whitfield-Gabrieli and Nieto-Castanon, 2012*). Temporal processing of each participant was conducted via the aCompCor method (*Behzadi et al., 2007*). Principal components associated with white matter and CSF voxels — identified for each participant via a segmentation of the anatomical images — were entered as additional nuisance regressors in the denoising step of the aCompCor method, along with scrubbing, six rigid-body head motion parameter values derived from spatial motion correction (x, y, and z translations and rotations) and six first-order temporal derivatives. These steps remove temporal confounding noise from non-neural factors, including motion parameters, cardiac, respiratory, and other physiological noise, and increase the sensitivity and reliability of functional connectivity analysis (*Whitfield-Gabrieli and Nieto-Castanon, 2012*). aCompCor accounts for the effects of participant movement without affecting intrinsic functional connectivity (*Chai et al., 2012*), and has been shown to reduce the impact of motion-induced BOLD signal changes (*Muschelli et al., 2014*). In addition, a conventional temporal band-pass filter based frequency of interest was applied to remove low-frequency drift and high-frequency noise (high pass = 0.008 Hz, low pass = 0.09 Hz). No global BOLD signal regression was applied because it may result in lower reproducibility of network metrics (*Telesford et al., 2013*). Realignment and scrubbing were entered as first-level covariates. Quality assurance measures (maximum inter-scan motion, number of valid/invalid scans per participant, and global correlation coefficient index per participant and condition) that were automatically generated during ART-based pre-processing and denoising steps were added as covariates for the second-level analysis.

## Network definition

Our core hypotheses focused on examining network topology based on functional connectivity effects across the two subsystems and midline core of the DN. Graph theoretic analyses were applied to a network of nodes (predefined ROIs) and edges (functional connections between ROIs), allowing us to quantify and compare topological alterations across these two subsystems and the midline core. Individual ROIs were treated as interconnected modules of nodes connected by edges (i.e., correlations or 'paths' between nodes).

Nineteen DN ROIs, based on spheres with 8 mm radii and centered on co-ordinates obtained from a study by *Andrews-Hanna et al. (2010)*, were generated using the SPM toolbox, MarsBaR (*Brett et al., 2002*), in SPM12. Nodes occurred within the common midline core, the left dorsal medial prefrontal cortex subsystem, or the left MTL subsystem, and within analogous co-ordinates on the right hemisphere for the dorsal medial prefrontal cortex subsystem and MTL subsystem: *midline core* — (a) anterior medial prefrontal cortex (MNI co-ordinates −6,52,−2) (amPFC) and (b) posterior cingulate cortex (−8,−56,26); *dorsal medial prefrontal cortex (dmPFC) subsystem* — (c) dorsal medial prefrontal cortex (0,52,26), (d) temporal parietal junction (−54,−54,28) (TPJ), (e) lateral temporal cortex (−60,−24,−18) (LTC), and (f) temporal pole (−50, 4,−40) (TempP); and, *MTL subsystem* — (g) ventral medial prefrontal cortex (vmPFC) (0,26,−18), (h) posterior inferior parietal lobule (pIPL) (−44,−74,32), (i) retrosplenial cortex (−14,−52,8) (Rsp), (j) parahippocampal cortex (−28,−40,−12) (PHC), and (k) hippocampal formation (−22,−20,−26) (HF).

In order to test whether the observed effects were specific to the DN, the somatomotor network, visual network, salience network, dorsal attention network, and ventral attention network were examined as control networks (derived from co-ordinates reported by *Power et al., 2011*). The meta-analytically derived ROIs were selected because these consist of regions with anatomic homology and account for common RSNs. The somatomotor network, visual network, salience network, dorsal attention network, and ventral attention network were comprised of 35, 31, 18, 11, and 9 respectively, spherical ROIs with 8 mm radii centered on co-ordinates specified in *Supplementary file 1n*. Probabilistic anatomical locations of the examined ROIs are also stated in *Supplementary file 1n*. These ROIs included portions of cuneus, fusiform, occipital and lingual gyri, which are outside of the network hypothesized to be affected by the bilateral CA3 lesions, but have been implicated in the recollection of events (*Addis et al., 2009*), and also included networks that have been associated with damage to the hippocampus and MTL (see discussion).

Results from all analyses were considered significant at p<0.05, with false-discovery rate correction applied for multiple comparisons.

## Network topology analyses

Tools for measuring network properties that are included in the CONN toolbox were used for the construction of graphs (i.e., nodes and edges [the functional connections]), their description, and the mathematical formula of each measure. Three classes of properties were computed for each participant: two measures of functional integration (average path length and global efficiency), two measures of functional segregation (clustering coefficient and local efficiency), and two centrality measures (degree and betweenness centrality). In the final step, we explored the relationship between the graph theoretic measures that were significantly different between the amnesic and control groups and behavioral performance on the AI.

Average path length is defined as the average number of steps along the shortest paths for all possible pairs of network nodes. Global network efficiency also assesses the integrative capacity of complex systems (*Watts and Strogatz, 1998*), and reflects effective information transfer within a network of nodes and edges. Clustering coefficient is related to the functional specificity of regional brain areas, whereas local efficiency is defined as the average global efficiency within a local subgraph consisting only of the neighbors of a given (index) node (i.e., excluding the node itself), and corresponds to a measure of the fault tolerance of the network and, by extension, the extent to which nodes are part of a local cluster of interconnected nodes. Degree is defined as the number of connections for each node to all other nodes in the network, whereas betweenness centrality measures the fraction of all of the shortest paths in a network that contain a given node, reflecting how connected a particular region was to other regions (higher numbers indicate participation in a large number of shortest paths).

In line with prior studies, edges were defined by thresholding the connectivity matrix to include *z*-scores > 0.84 and ignored negative edges (*Power et al., 2011*; *Rubinov and Sporns, 2010*), because there remains little consensus for interpreting negative edge weights, and some graph measures either need to be adapted or are undefined when negative edges are present (*Murphy and Fox, 2017*; *Rubinov and Sporns, 2011*). This threshold was chosen to balance statistical evidence of functional connectivity with the need to minimize less reliable sparse networks (*Braun et al., 2012*; *Rubinov and Sporns, 2011*; *Wang et al., 2011*). Negative edges were excluded

because these reduce the reliability of graph theoretic measures (*Wang et al., 2011*). See results section for stability analyses conducted using cost-based thresholding.

## Seed-to-voxel and ROI-to-ROI functional connectivity analyses

Seed regions-of-interest were comprised of four spherical ROIs with 8 mm$^3$ radii: left hippocampus (MNI co-ordinates −24,−22,−16), right hippocampus (24,−22,−16), occipital pole within the visual network (18,−47,−10), and primary motor cortex (M1) within the somatomotor network (−40,−19,54). The left hippocampus ROI was centered on published functional co-ordinates associated with episodic memory (*Hirshhorn et al., 2012*), whereas the right hippocampus was a homotopic region. Hippocampal ROIs with these co-ordinates were used in the seed-to-voxel and ROI-to-ROI functional connectivity analyses because functional nodes proposed by *Power et al. (2011)* and *Andrews-Hanna et al. (2010)* do not cover the main body of the hippocampus. The occipital pole seed and M1 seed correspond to co-ordinates obtained from visual network and somatomotor network, respectively. All seed ROIs were generated using the SPM toolbox MarsBaR (*Brett et al., 2002*) in SPM12.

Individual correlation maps were generated in the CONN toolbox by extracting mean resting-state BOLD signal time series for each seed ROI. Pairwise regional correlation coefficients between each seed ROI and all other voxels in the volume were computed using bivariate Pearson's product moment correlations. The value of each voxel in the volume represents the relative degree of functional connectivity with each seed (*Whitfield-Gabrieli and Nieto-Castanon, 2012*). Correlation coefficients were z-transformed into normally distributed scores by applying Fisher's transformation to generate maps of voxelwise functional connectivity of seed-ROIs for each participant. Separate analyses using the CONN toolbox were performed for each seed-ROI to whole-brain voxels, with the vectors of average time course data of whole-brain activity, corresponding to the brain map of seed-ROI connectivity for each participant. These maps were entered into a second-level general linear model analyses random effects analysis of relative functional connectivity in the CONN toolbox using a two-sided independent t-test to investigate between-group differences in seed-to-voxel connectivity. ROI-to-ROI analyses were conducted by specifying a bivariate correlation model between ROIs. Correlation coefficients were converted to z-values using Fisher r-to-z transformation to improve the normality of the distribution. Functional connectivity differences between the amnesic group and control group were estimated by specifying a group contrast in the second level analysis of CONN, in order to conduct a t-test on the connections and retain only the significant connections.

As described in prior studies (*Fallon et al., 2016*), whole-brain seed-to-voxel between-group comparisons were assessed at a height threshold of p(uncorrected) <0.001 before FDR correction was applied at the cluster level (p<0.05). Negative correlations (i.e., anti-correlations) were ignored because whole-brain signal normalization changes the correlation distribution to mean near zero, leading to negative correlations even if these correlations are not initially present in the data (*Murphy et al., 2009*). Statistical significance for the ROI-to-ROI analyses was assessed by entering the Fisher z values into a between-group t-test, at a two-sided p-FDR <0.05 threshold (seed-level correction).

## Behavior

### Autobiographical interview

Episodic and context-independent (semantic) memory were investigated under the retrieval conditions of the AI (*Levine et al., 2002*). Data were obtained from the 16 participants in the amnesic group and 16 participants in the control group. Identical intervals were sampled for amnesic and control group participants. Verbal prompts were used to encourage the recovery of spatial, perceptual, and mental state details related to temporally specific recent (within the past year) and remote (extending to ~60 years) event memories. Given the monophasic nature of LGI1-antibody-complex LE, it is reasonable to assume that the recent and remote retrograde memory were acquired prior to the illness, and represent a measure of hippocampus-mediated retrieval mechanisms.

### Scoring and reliability

All verbal responses on the AI were digital-audio recorded and then transcribed for scoring offline. Transcripts were compiled so that identifying personal details or anything that pertained to group

membership were removed. Two independent raters scored 100% of the episodes acquired from each time period for all participants in the amnesic and control groups. The raters were blinded to the identity and group membership of each transcript. Responses were scored according to the standardized method outlined in the AI Scoring Manual (*Levine et al., 2002*). Accounts were segmented into informational bits/details, or those occurrences, observations, or thoughts expressed as a grammatical clause. Details that related directly to a unique event, and which had a specific time and place or were associated with episodic re-experiencing (such as thoughts or emotions), were classified as internal (episodic) details. Information that did not relate to the event was assigned to external details, and then sub-categorized into semantic (factual information or extended events), repetitions (where previous details had been given with no new elaboration), and other (e.g., meta-cognitive statements, editorializing, and inferences).

In line with prior studies of autobiographical memory (*Tranel and Jones, 2006*), each narrative was verified, where possible, against a collateral – a spouse, family member, or friend – for confabulation (differences in personal reflections and thoughts notwithstanding). No evidence of confabulation was found. Therefore, all acquired events were included in the analyses. The quantitative text-based method applied when scoring the AI aligns with standardized tests that examine the retrieval of narrative-based details.

## Neuropsychological assessment

In order to obtain a detailed cognitive profile for the participants in the amnesic group (*Figure 6* and *Supplementary file 1a*), comprehensive neuropsychological assessment was conducted using standardized neuropsychological tests to assess the following domains: intelligence, verbal memory, visual memory, recognition memory, attention, language, executive function, visuomotor skills, and visuoconstruction. One participant in the amnesic group was unable to complete the attention tasks due to reduced visual fields (secondary to age-related macular degeneration), and the subtests that underlie the indice of intelligence were not conducted in one participant. All subtests that were conducted were included in the analyses used to generate the results reported in *Figure 6* and *Supplementary file 1a*.

The neuropsychological composite measures were comprised of the following individual tests: *Intelligence* Wechsler Abbreviated Scale of Intelligence (WASI) – Similarities and Matrix Reasoning (*Wechsler, 1999*); *Verbal memory* — Wechsler Memory Scale–III (WMS-III) – Logical Memory I and II, Logical Memory I and II themes, and Word Lists I and II (*Wechsler, 1997*) and Doors and People – People and People Recall Tests (*Baddeley et al., 1994*); *Visual memory* — Rey complex figure – Immediate-Recall (*Osterrieth, 1944*) and Doors and People – Shapes Test and Visual Forgetting (*Baddeley et al., 1994*); *Recognition memory* — WMS-III – Words Lists II Recognition (*Wechsler, 1997*), Recognition Memory Test – Words and Faces (*Warrington, 1984*), and Doors and People – Names and Doors Tests (*Baddeley et al., 1994*); *Attention* — Test of Everyday Attention – all subtests (*Robertson et al., 1994*); *Language* — Graded Naming Test (*McKenna and Warrington, 1980*), Delis-Kaplan Executive Function System (D-KEFS) – Letter Fluency and Category Fluency from the Verbal Fluency Test (*Delis et al., 2001*), and the Camel and Cactus Test (*Bozeat et al., 2000*); *Executive function* — D-KEFS – Category Switching from the Verbal Fluency Test, Number-Letter Switching from the Trail Making Test and Colour-Word Interference Test (*Delis et al., 2001*), and WMS-III – Digit Span (*Wechsler, 1997*); *Visuomotor skills* — D-KEFS – Visual Scanning, Number Sequencing, Letter Sequencing, and Motor Speed from the Trail Making Test (*Delis et al., 2001*); and *Visuoconstruction skills* — Rey complex figure – Copy (*Osterrieth, 1944*).

Scores on the standardized neuropsychological tests were first transformed into age-corrected standard values, where available, then transformed into $z$-scores, and averaged across the tests within each domain to derive composite scores corresponding to indices for the respective cognitive domains. A group domain was then obtained as a group average across each neuropsychological domain. One-sample $t$-tests (two-tailed) were conducted to determine whether group performance was significantly different from normative data. fMRI and neuropsychological studies indicate that the retrieval of autobiographical and experimentally determined encoded memories are dissociable (*McDermott et al., 2009*; *Palombo et al., 2015*; *Patihis et al., 2013*). The behavioral results from the AI and mild impairment on standardized measures of recall are compatible with the hypothesis that, under some conditions, the retrieval of autobiographical event knowledge is qualitatively

different from other forms of episodic retrieval (*Chen et al., 2017*; *McDermott et al., 2009*; *Roediger and McDermott, 2013*) (*Figures 4*, *5* and *6*; *Supplementary file 1a*). Nonetheless, other studies indicate that remember-know responses and memory judgements on personally familiar stimuli, such as landmarks, can each be associated with activity in brain regions implicated in auto-biographical recall (*Elman et al., 2013*; *Frithsen and Miller, 2014*). More generally, dissociations need, at the very least, to be tested under conditions that strictly align the retrieval demands engaged during laboratory-based versus autobiographical memory tasks. Indeed, even changes in retrieval orientation that emphasize either contextual or conceptual features of an autobiographical memory can modulate the distributed network of brain regions that support autobiographical retrieval (*Gurguryan and Sheldon, 2019*).

## Statistical analyses

Mixed-model omnibus factorial ANOVAs were used to analyze between-group mean differences for hippocampal subfield volumes and for AI internal and external detail composite scores. Mauchly's test was used to assess the assumption of sphericity. Modifications to the degrees of freedom were applied when sphericity was violated so that a valid *F*-statistic could be obtained. Greenhouse-Geisser correction was applied if estimated epsilon ($\varepsilon$) was less than 0.75, whereas Huynh-Feldt correction was applied if the estimated $\varepsilon$ exceeded 0.75. Planned comparisons were used to assess for differences between participants in the amnesia group and the control group for all subfields. Group differences in internal and external detail scores from the AI over time were assessed using subsidiary mixed-model ANOVAs. These were conducted using SPSS Version 24.0 (Armonk, NY; IBM Corp). Significance values for planned comparisons were corrected for multiple comparisons using the Holm-Bonferroni method. Robust multiple regression with correction was used to investigate the relationships between CA3 volume and graph theoretic measures with the cumulative internal detail point scores from the AI. Robust multiple regression with correction was performed with NCSS version 9 (Kaysville, Utah; NCSS LLC). As noted above, analyses of the standardized neuropsychological test data were conducted using one-sample *t*-tests (two-tailed) to determine whether group indices reported in *Figure 6* and *Supplementary file 1a* were significantly different from normative data.

Dice similarity indices (DSIs) of geometric overlap for the five subfields, derived using Convert3D (www.itksnap.org), were obtained to assess the intra- and inter-rater reliability of manual hippocampal segmentation. In addition, the agreement between twice-repeated manual segmentations of all five subfields as a function of side (left, right) and twice-repeated scoring of data acquired on the AI were evaluated using intra-class correlation coefficients (ICCs). A two-way mixed-model design was used to test the degree of absolute agreement. Intra-class correlations were conducted to assess inter-rated reliability (two-way mixed-effects model) for internal and external details between the two raters (*McGraw and Wong, 1996*).

## Acknowledgements

The authors thank the participants and their families for their contributions to this study. We also thank Professor Angela Vincent for her support.

## Additional information

### Competing interests

Sarosh R Irani: Sarosh R Irani, Angela Vincent, and the Department of Clinical Neurology in Oxford receive royalties and payments for antibody assays are both named inventors on patent application WO/2010/046716 entitled 'Neurological Autoimmune Disorders'. The patent has been licensed to Euroimmun AG for the development of assays for LGI1 and other VGKC-complex antibodies. Dr. Irani also reports personal fees from MedImmune, grants from Wellcome Trust, grants from UCB Pharma, grants from BMA (Vera Down and Margaret Temple grants), Fulbright UK-US Commission and Epilepsy Research UK, outside the submitted work. Dr. Irani served on the scientific advisory board for Encephalitis Society and MedImmune. Saiju Jacob: Dr. Jacob has been a scientific advisory board member for Alexion and Alynylam pharmaceuticals. The other authors declare that no competing interests exist.

## Funding

| Funder | Author |
|---|---|
| Medical Research Council | Penny A Gowland |
| Engineering and Physical Sciences Research Council | Penny A Gowland |
| John Fell Fund, University of Oxford | Christopher Kennard<br>Clive R Rosenthal |
| National Institute for Health Research | Anne M Aimola Davies<br>Sarosh R Irani<br>Christopher Kennard<br>Clive R Rosenthal |
| Guarantors of Brain | Thomas D Miller |
| Patrick Berthoud Charitable Trust | Thomas D Miller |
| Encephalitis Society | Thomas D Miller |
| National Health and Medical Research Council | Trevor TJ Chong |
| Australian Research Council | Trevor TJ Chong |
| Wellcome Trust | Sarosh R Irani<br>Masud Husain<br>Christopher Kennard<br>Clive R Rosenthal |
| British Medical Association | Sarosh R Irani |

The funders had no role in study design, data collection and interpretation, or the decision to submit the work for publication.

## Author contributions

Thomas D Miller, Resources, Data curation, Formal analysis, Funding acquisition, Validation, Investigation, Visualization, Methodology, Writing—review and editing; Trevor T-J Chong, Tammy WC Ng, Validation, Investigation, Writing—review and editing; Anne M Aimola Davies, Funding acquisition, Methodology, Writing—review and editing; Michael R Johnson, Sarosh R Irani, Resources, Writing—review and editing; Masud Husain, Resources, Funding acquisition, Writing—review and editing; Saiju Jacob, Paul Maddison, Resources; Christopher Kennard, Resources, Funding acquisition; Penny A Gowland, Resources, Software, Funding acquisition, Investigation, Methodology, Writing—review and editing; Clive R Rosenthal, Conceptualization, Resources, Data curation, Software, Formal analysis, Supervision, Funding acquisition, Validation, Investigation, Visualization, Methodology, Writing—original draft, Project administration, Writing—review and editing

## Author ORCIDs

Trevor T-J Chong https://orcid.org/0000-0001-7764-3811
Clive R Rosenthal https://orcid.org/0000-0002-5960-4648

## Ethics

Human subjects: Informed written consent was obtained from all participants for all procedures and for consent to publish, in accordance with the terms of approval granted by the local research ethics committee (04/Q0406/147) and the principles expressed in the Declaration of Helsinki.

## Decision letter and Author response

Decision letter https://doi.org/10.7554/eLife.41836.sa1
Author response https://doi.org/10.7554/eLife.41836.sa2

## Additional files

### Supplementary files

• Supplementary file 1. Supplementary tables reporting results from neuropsychological assessments, graph theoretic analyses, seed-to-voxel analyses, ROI-to-ROI analyses, and multiple regression based analyses. (**a**) Neuropsychological domain performance in the amnesic group. (**b**) Results from graph theoretic analyses of the default network (DN). Between-group differences in global efficiency, local efficiency, betweenness centrality, average path length, clustering coefficient, and degree were examined. Network edges (adjacency matrix threshold) were thresholded at a $z$-score > 0.84 (one-sided, positive) and assessed at a corrected analysis threshold (p-FDR <0.05, two-sided) and, for completeness, at an uncorrected analysis threshold (p-uncorrected <0.05, two-sided). (**c**) Results from graph theoretic analyses of the somatomotor network. Between-group differences in global efficiency, local efficiency, betweenness centrality, average path length, clustering coefficient, and degree were examined. Network edges (adjacency matrix threshold) were thresholded at a $z$-score > 0.84 (one-sided, positive) and assessed at a corrected analysis threshold (p-FDR <0.05, two-sided), and, for completeness, at an uncorrected analysis threshold (p-uncorrected <0.05, two-sided). (**d**) Results from graph theoretic analyses of the visual network. Between-group differences in global efficiency, local efficiency, betweenness centrality, average path length, clustering coefficient, and degree were examined. Network edges (adjacency matrix threshold) were thresholded at a $z$-score > 0.84 (one-sided, positive) and assessed at a corrected (p-FDR <0.05, two-sided), and, for completeness, at an uncorrected analysis threshold (p-uncorrected <0.05, two-sided). (**e**) Results from graph theoretic analyses of the salience network. Between-group differences in global efficiency, local efficiency, betweenness centrality, average path length, clustering coefficient, and degree were examined. Network edges (adjacency matrix threshold) were thresholded at a $z$-score > 0.84 (one-sided, positive) and assessed at a corrected (p-FDR <0.05, two-sided), and, for completeness, at an uncorrected analysis threshold (p-uncorrected <0.05, two-sided). (**f**) Results from graph theoretic analyses of the ventral attention network. Between-group differences in global efficiency, local efficiency, betweenness centrality, average path length, clustering coefficient and degree were examined. Network edges (adjacency matrix threshold) were thresholded at a $z$-score > 0.84 (one-sided, positive) and assessed at a corrected analysis threshold (p-FDR <0.05, two-sided), and, for completeness, at an uncorrected analysis threshold (p-uncorrected <0.05, two-sided). (**g**) Results from graph theoretic analyses of the dorsal attention network. Between-group differences in global efficiency, local efficiency, betweenness centrality, average path length, clustering coefficient, and degree were examined. Network edges (adjacency matrix threshold) were thresholded at a $z$-score > 0.84 (one-sided, positive) and assessed at a corrected anaylsis threshold (p-FDR <0.05, two-sided), and, for completeness, at an uncorrected analysis threshold (p-uncorrected <0.05, two-sided). (**h**) Results from graph theoretic stability analyses of default network topology. The amnesic group and the control group were assessed using an adjacency matrix threshold based on cost. The analysis threshold (i.e., p-FDR <0.05, two-sided) was same as when the default network was assessed using the $z$-score based adjacency matrix threshold. Nodes that were not significant at the FDR-corrected analysis threshold applied to infer significance are reported at an uncorrected analysis threshold (p-uncorrected < 0.05, two-sided). (**i**) Results from graph theoretic stability analyses of somatomotor network topology. The amnesic group and the control group were assessed using an adjacency matrix threshold based on cost. The analysis threshold (i.e., p-FDR <0.05, two-sided) was same as when the somatomotor network was assessed using the $z$-score based adjacency matrix threshold. Nodes that were not significant at the p-FDR-corrected analysis threshold applied to infer significance are reported at an uncorrected analysis threshold (p-uncorrected <0.05, two-sided). (**j**) Results from graph theoretic stability analyses of visual network topology. The amnesic group and the control group were assessed using an adjacency matrix threshold based on cost. The analysis threshold (i.e., p-FDR <0.05, two-sided) was same as when the visual network was assessed using the $z$-score based adjacency matrix threshold. Nodes that were not significant at the p-FDR-corrected analysis threshold applied to infer significance are reported at an uncorrected analysis threshold (p-uncorrected <0.05, two-sided). (**k**) Results from graph theoretic stability analyses of the salience network. The amnesic group and the control group were assessed using an adjacency matrix threshold based on cost. The analysis threshold (i.e., p-FDR <0.05, two-sided) was same as when the salience network was assessed using the $z$-score based adjacency

matrix threshold. Nodes that were not significant at the p-FDR-corrected analysis threshold applied to infer significance are reported an uncorrected analysis threshold (p-uncorrected <0.05, two-sided). (**l**) Results from graph theoretic stability analyses of the ventral attention network. The amnesic group and the control group were assessed using an adjacency matrix threshold based on cost. The analysis threshold (i.e., p-FDR <0.05, two-sided) was same as when the ventral attention network was assessed using the z-score based adjacency matrix threshold. Nodes that were not significant at the p-FDR-corrected analysis threshold applied to infer significance are reported at an uncorrected analysis threshold (p-uncorrected <0.05, two-sided). (**m**) Results from graph theoretic stability analyses of the dorsal attention network. The amnesic group and the control group were assessed using an adjacency matrix threshold based on cost. The analysis threshold (i.e., p-FDR <0.05, two-sided) was same as when the dorsal attention network was assessed using the z-score based adjacency matrix threshold. Nodes that were not significant at the p-FDR-corrected analysis threshold applied to infer significance are reported at an uncorrected analysis threshold (p-uncorrected <0.05, two-sided). (**n**) MNI co-ordinates for the nodes used in control network analyses. MNI co-ordinates for the nodes that correspond to the somatomotor network, visual network, salience network, dorsal attention network, and ventral attention network were based on the parcellation scheme proposed by *Power et al. (2011)*. Probabilistic anatomical locations of the MNI co-ordinates are defined using the Harvard-Oxford Cortical and Subcortical Probabilistic Atlases. (**o**) Results from between-group seed-to-voxel functional connectivity based analyses. No brain regions exhibited significant differences (height threshold, p-uncorrected <0.001 and an extent threshold of p-FDR <0.05 at the cluster level) in functional connectivity between the amnesic group and the control group, when tested with left and right hippocampal seed regions-of-interest, an occipital pole seed within the visual network, and a seed in primary motor cortex within the somatomotor network. Seed regions were spheres with 8 mm radii. For the left hippocampal seed region, a between-group difference in functional connectivity was found only at a lenient, p-uncorrected <0.05 cluster-size threshold. There were no significant clusters for the right hippocampal seed region, even when between-group differences were assessed at a cluster-size threshold set at p-uncorrected <0.05 (see *Figure 7—figure supplement 1*). (**p**) Results for left and right hippocampus seed-to-voxel functional connectivity based analyses. MNI co-ordinates of brain regions that exhibited significant functional connectivity with left and right hippocampal seed regions-of-interest shown separately for the amnesic group and the control group (height threshold, p-uncorrected <0.001 and an extent threshold of p-FDR <0.05, at the cluster level). Seed regions were spheres with 8 mm radii (see *Figure 7—figure supplement 2*). (**q**) Results from robust multiple linear regression analysis on nodes in the default network with significantly increased average path length (independent variables) relative to the control group and total internal (episodic) detail remembered on the AI. Average path length values entered into the robust multiple regression analysis were based on the difference between each participant and the mean of the control group for each affected ROI/node. (**r**) Results from multiple linear regression analysis examining link between nodes in the DN that exhibited significantly different local efficiency and internal (episodic) detail remembered on the AI. Local efficiency values entered into the robust multiple regression analysis were based on the difference between each participant and the mean of the control group for each affected ROI/node.

- Transparent reporting form

## Data availability

The data used to support the results of this study are not available for open distribution to comply with the restrictions of the local research ethics committee and the terms of consent signed by the human participants. For further details on the restrictions related to data sharing, please email clive.rosenthal@clneuro.ox.ac.uk.

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

**Appendix 1**

## Supplementary Results

### Hippocampal subfield morphometry

Results from the quantitative analyses of the hippocampal subfields compared amnesic group participants' regional subfield volumes (corrected for intracranial volume) to volumes from control participants. A three-way mixed-model ANOVA, with two within-subjects variables (subfield and side) and one between-subjects variable (group), was used to test for between-group differences in hippocampal subfield volumes between participants in the amnesic and control groups (*Table 1* and *Figure 2—figure supplement 1*). The assumption of sphericity was violated for subfield ($\chi^2_{(9)}$ = 52.46, p<0.0001) and for the interaction between subfield and side ($\chi^2_{(9)}$ = 63.48, p<0.0001), so degrees of freedom were corrected using Greenhouse-Geisser correction ($\varepsilon$ = 0.551). There were significant main effects of group ($F_{(1,28)}$ = 5.52, p=0.026, $\eta^2_p$ = 0.165), side ($F_{(1,28)}$ = 32.58, p<0.001, $\eta^2_p$ = 0.538), and subfield ($F_{(2.21,61.74)}$ = 384.01, p<0.0001, $\eta^2_p$ = 0.932). Significant two-way interactions were found between group and subfield ($F_{(5.30, 61.74)}$ = 5.30, p=0.006, $\eta^2_p$ = 0.159) and between side and subfield ($F_{(2.02,56.55)}$ = 14.15, p<0.0001, $\eta^2_p$ = 0.336), but not between group and side ($F_{(1,28)}$ = 1.25, p=0.272, $\eta^2_p$ = 0.043). The three-way interaction was not significant ($F_{(2.02,56.55)}$ = 0.43, p=0.66, $\eta^2_p$ = 0.015).

A series of planned comparisons revealed a significant reduction in total CA3 volume – collapsed across left and right CA3 due to the absence of a significant three-way interaction – in the amnesic group relative to control group participants ($F_{(1,28)}$ = 14.52, p=0.001, Cohen's *d* = 1.39); mean reduction = -29%) (*Figure 2*). The between-group differences in CA1, CA2, subiculum, and dentate gyrus volumes were not statistically significant at the Holm-Bonferroni alpha criterion corrected for multiple comparisons.

### Subgroup analyses

As noted in the main results section, we examined the impact of including a participant with bilateral CA3 volume eight percent greater than the control group mean, but 18% below the matched control participant. The results from this subgroup analyses are outlined next.

A three-way mixed-model ANOVA, with two within-subjects variables (subfield and side) and one between-subjects variable (group), was used to test for group differences in hippocampal subfield volumes between participants in the amnesia and control groups (*Table 1* and *Figure 2—figure supplement 1*). The assumption of sphericity was violated for subfield ($\chi^2_{(9)}$ = 50.41, p<0.0001) and for the interaction between subfield and side ($\chi^2_{(9)}$ = 65.53, p<0.0001), so degrees of freedom were corrected using Greenhouse-Geisser correction ($\varepsilon$ = 0.500). There were significant main effects of group ($F_{(1,27)}$ = 17.11, p<0.0001, $\eta^2_p$ = 0.388), side ($F_{(1,28)}$ = 37.54, p<0.001, $\eta^2_p$ = 0.582), and subfield ($F_{(1.97,53.14)}$ = 503.25, p<0.0001, $\eta^2_p$ = 0.949). Significant two-way interactions were found between group and subfield ($F_{(1.97,53.14)}$ = 8.82, p=0.001, $\eta^2_p$ = 0.246) and between side and subfield ($F_{(2,53.99)}$ = 15.82, p<0.0001, $\eta^2_p$ = 0.370), but not between group and side ($F_{(1,27)}$ = 0.68, p=0.421, $\eta^2_p$ = 0.024). The three-way interaction was not significant ($F_{(2,53.99)}$ = 0.24, p=0.791, $\eta^2_p$ = 0.009).

Also in line with the main results, the planned group comparisons revealed a significant reduction in total CA3 volume – again, collapsed across left and right CA3 due to the absence of a significant group interaction with side and subfield – of the amnesic group participants relative to controls ($F_{(1,27)}$ = 18.34, p<0.0001), whereas the between-group differences in subiculum, CA1, CA2, and dentate gyrus volumes were not statistically significant at the alpha criterion corrected for multiple comparisons (Cohen's *d* all <0.8).

## Autobiographical Interview

For completeness, we report the results from an omnibus mixed-model ANOVA conducted with the full factorial design. A 2 (group: amnesic, controls) x 2 (detail type: internal (episodic), external (non-episodic, semantic)) x 5 (time interval: 0–11, 11–18, 18–30, 30–55, and recent anterograde (past year)) mixed-model ANOVA was performed on the mean AI cumulative point scores, with group (amnesic, control) as a between-subjects variable and detail (internal, external) and time (past year (i.e., anterograde interval); 30–55, 18–30, 11–18, and 0–11 years (i.e., retrograde intervals)) as within-subjects variables. Mauchly's test indicated the assumption of sphericity was violated for time ($\chi^2_{(9)}$ = 35.77, p<0.0001) and for the interaction between detail type and time ($\chi^2_{(9)}$ = 21.91, p=0.009). Therefore, degrees of freedom were corrected using Huynh-Feldt estimates ($\varepsilon$ = 0.876). There was a significant main effect of group ($F_{(1,30)}$ = 9.29, p=0.005, $\eta^2_p$ = 0.237) and detail type ($F_{(1,30)}$ = 296.63, p<0.0001, $\eta^2_p$ = 0.908), but not time ($F_{(2.88,86.53)}$ = 0.94, p=0.413, $\eta^2_p$ = 0.030). Significant two-way interactions were found between group and detail type ($F_{(1,30)}$ = 22.23, p<0.0001, $\eta^2_p$ = 0.426) and between group and time ($F_{(2.88,86.53)}$ = 4.51, p=0.006, $\eta^2_p$ = 0.131), whereas the interaction between time and detail type was not significant ($F_{(3.50,105.11)}$ = 1.40, p=0.244, $\eta^2_p$ = 0.045). The three-way interaction between group, time and detail type was also significant ($F_{(3.50,105.11)}$ = 2.83, p=0.034, $\eta^2_p$ = 0.086; *Figures 4* and *5*). Post hoc two-way mixed model ANOVA's and direct pairwise contrasts investigating between-group main effects and interaction terms as a function of detail type are reported in the main text.

## Subgroup analyses

As noted in the main results section, the results from the AI held both in terms of the main effects and interaction terms when we examined the inclusion of a participant in whom we were unable to obtain quantitative hippocampal subfield volumes as compared to the results conducted with N = 16. In addition, we examined the impact of a participant with bilateral CA3 volume that was eight percent larger than the control group mean, but 18% below the matched control participant. The results from these subgroup analyses are outlined next.

First, re-analysis of the data obtained from the AI without the amnesic group participant in whom we were unable to obtain quantitative hippocampal subfield volumes (i.e., with N = 15 amnesic group participants) indicated that the results held in the main effects and key interaction terms as compared to the results conducted with N = 16. Mauchly's test demonstrated that the assumption of sphericity had been violated for time ($\chi^2_{(9)}$ = 33.46, p<0.0001). Degrees of freedom were corrected using Greenhouse-Geisser estimates ($\varepsilon$ = 0.651). In particular, a two-way mixed-model ANOVA conducted on internal (episodic) detail, with group (amnesic, control) as a between-subjects factor and time (past year, 30–55, 18–30, 11–18, and 0–11 years), as a within-subjects factor, revealed a significant main effect of group ($F_{(1,29)}$ = 14.48, p=0.001, $\eta^2_p$ = 0.333) and a significant two-way interaction between group and time ($F_{(2.60,75.50)}$ = 4.24, p=0.011, $\eta^2_p$ = 0.127), whereas time was not significant ($F_{\{2.60,75.50\}}$ = 0.87, p=0.449, $\eta^2_p$ = 0.029). As was the case in the main analyses of internal detail, exclusion of the earliest remote memory from the within-subjects factor (i.e., past year, 30–55, 18–30, and 11–18 years) revealed a significant main effect of group ($F_{(1,29)}$ = 20.88, p<0.0001, $\eta^2_p$ = 0.419), whereas the two-way interaction between group and time ($F_{(2.58,74.91)}$ = 1.59, p=0.205, $\eta^2_p$ = 0.052) and the main effect of time ($F_{(2.58,74.91)}$ = 0.50, p=0.659, $\eta^2_p$ = 0.017) were not significant. Degrees of freedom were corrected using Huynh-Feldt estimates ($\varepsilon$ = 0.861) because Mauchly's test demonstrated that the assumption of sphericity had been violated for time ($\chi^2_{(5)}$ = 12.39, p=0.030). Re-analysis of external detail with a 2 (amnesic, control) x 5 (past year, 30–55, 18–30, 11–18, and 0–11 years) mixed-model ANOVA revealed that the main effect of group ($F_{(1,29)}$ = 1.95, p=0.173, $\eta^2_p$ = 0.063) and time ($F_{(4,116)}$ = 1.14, p=0.342, $\eta^2_p$ = 0.038) were not significant. In addition, the two-way interaction between group and time was not significant ($F_{(4,116)}$ = 1.28, p=0.283, $\eta^2_p$ = 0.042). Mauchly's test of sphericity was not violated for time ($\chi^2_{(9)}$ = 15.58, p=0.077).

Second, a re-analysis of the data obtained from the AI was conducted without the amnesic group participant with total CA3 volume that was eight percent greater than the control

mean, but 18% below the CA3 volume of the matched control participant. The results held both in terms of the main effects and interaction terms, and with the planned comparisons as compared to the results conducted with N = 16 participants in the amnesic group. In particular, a two-way mixed-model ANOVA conducted on the internal (episodic) detail, with one between-subjects factor (group: amnesic, control) and one within-subjects factor (time: past year, 30–55, 18–30, 11–18, 0–11 years), revealed a significant main effect of group ($F_{(1,29)}$ = 14.56, p=0.001, $\eta^2_p$ = 0.334) and a significant two-way interaction between group and time ($F_{(2.67,77.42)}$ = 3.58, p=0.021, $\eta^2_p$ = 0.110), whereas time was not significant ($F_{(2.67,77.42)}$ = 1.11, p=0.346, $\eta^2_p$ = 0.037). Degrees of freedom were corrected using Greenhouse-Geisser estimates ($\varepsilon$ = 0.667) because Mauchly's test demonstrated that the assumption of sphericity had been violated for time ($\chi^2_{(9)}$ = 30.75, p<0.0001). As was the case in the main analyses on internal detail, exclusion of the earliest remote memory from the within-subjects factor (i.e., past year, 30–55, 18–30, 11–18 years) revealed a significant main effect of group ($F_{(1,29)}$ = 20.90, p<0.0001, $\eta^2_p$ = 0.419), whereas the two-way interaction between group and time ($F_{(2.62,75.97)}$ = 1.37, p=0.260, $\eta^2_p$ = 0.045) and the main effect of time ($F_{(2.62,75.97)}$ = 0.60, p=0.597, $\eta^2_p$ = 0.020) were not significant. Degrees of freedom were corrected using Huynh-Feldt estimates ($\varepsilon$ = 0.873) because Mauchly's test demonstrated that the assumption of sphericity had been violated for time ($\chi^2_{(5)}$ = 11.61, p=0.041). Re-analysis of cumulative external details point scores with a 2 (amnesic, control) x 5 (past year, 30–55, 18–30, 11–18, 0–11 years) mixed-model ANOVA revealed that the main effect of group ($F_{(1,29)}$ = 1.49, p=0.233, $\eta^2_p$ = 0.049) and time ($F_{(4,116)}$ = 1.48, p=0.212, $\eta^2_p$ = 0.049) were not significant. In addition, the two-way interaction between group and time was not significant ($F_{(4,116)}$ = 1.52, p=0.201, $\eta^2_p$ = 0.050). Mauchly's test of sphericity was not violated for time ($\chi^2_{(9)}$ = 4.59, p=0.868).

