## [Decision Letter]

**Acceptance summary:**

Miller and colleagues address an important debate in memory research: Whether the role of the hippocampus in episodic retrieval is time limited. The literature on this topic is riddled with discrepancies, which may be in part due to the fact that past studies have not been able to investigate the contribution of individual subregions. Here, using an uncharacteristically large sample of well-characterized and rare patients, the authors examine the role of selective lesions to the CA3 subregion of the hippocampus and their knock-on effects to functional integration in the default network. First, they observed that patients showed a flat temporal gradient for retrieval of episodic autobiographical information, with disrupted memory in the patients for all but the oldest memories from when the patients were between 0 and 11 years of age. The fact that the patients showed impoverished episodic memory for events that happened over 50 years ago supports the notion that the hippocampus is required for episodic retrieval of even very old memories. The authors then investigated how the CA3 supports autobiographical memory in terms of functional integration with the default network of the brain. Using graph theoretic analyses of 7.0T resting-state fMRI data, the authors found that CA3 damage led to increased path length specifically in the medial temporal of subsystem of the default network. Moreover, both loss of CA3 volume and loss of integration in the retrosplenial cortex were predictive of episodic retrieval performance. These results will be relevant to anyone interested in how the brain consolidates and retrieves autobiographical memory.

**Decision letter after peer review:**

Thank you for submitting your article "Human hippocampal CA3 damage disrupts both recent and remote episodic memories" for consideration by *eLife*. Your article has been reviewed by three peer reviewers, one of whom is a member of our Board of Reviewing Editors, and the evaluation has been overseen by Laura Colgin as the Senior Editor.

The reviewers have discussed the reviews with one another and the Reviewing Editor has drafted this decision to help you prepare a revised submission.

We all agreed that the results are important to the field of memory, and we judged the paper to be of broad interest. However, we agreed on several major concerns, largely regarding how the analyses were conducted and the conclusions that can be drawn from them. These are detailed below.

Essential revisions:

1) The authors' main claim is that, following CA3 lesions, episodic memory is equally susceptible to impairment no matter how long ago the memories were acquired. This claim seems to rest on a comparison of memory for 0-11 years with 30-55 years, and 0-11 years with the preceding year, separately for patients and controls. Why were only those comparisons chosen, and not others? Moreover, we do not think that the chosen analyses are ideal for depicting a flat temporal gradient of amnesia for the patients. Looking at Figure 4, the amnesic patients' deficit seems to grow with time: the patient vs control difference for 0-11 years seems to be much smaller than the patient vs control difference for the last year. Did the authors specifically test for whether the difference between patients and controls changes over time? The 3-way interaction between group, type of detail, and time does not clarify this: instead, the authors can look for a two-way interaction between group and time, just for internal details. In other words, the lack of a change in patients' internal details over time does not indicate that memory was equally impaired across time, because the controls' memory was better for recent vs remote events. If patients' scores are relatively flat, and controls' memory is better for recent events, that suggests that patients' impairment (relative to controls) increases as memories get more recent. This is a speculation based on the figure, but the authors should directly test it by looking for a 2-way interaction between group and time for internal details. If there is a main effect of group and no group x time interaction, it would bolster the claim that memory is equally impaired for all time periods. If there is a group x time interaction, then the main conclusion of the paper is not supported.

In a very similar vein, just put slightly differently: The authors state that "Deficits were hypothesised to affect internal details independently of the age of recalled events" (Introduction) and "Episodic loss of detail in the amnesic group for events did not vary over time (~1- 60 year interval)" (Figure 4 caption). This phrasing led to some confusion, because the difference between patients and controls for internal details does vary with the age of recalled events: controls show a gradient over time and patients don't, thus the difference varies (decreases) with the age of the memory. A "loss" of details in the patients implies that this is relative to the controls. After some closer reading it seems that the intended meaning was something more like "both recent and remote memories were affected" by CA3 damage. The ANOVA associated with Figure 4 reports differences between groups "across all five time intervals", but the between-time-bin comparisons (eg, 0-11 vs. last year) are reported within-group only. For clarity, could the authors please elaborate on these issues, and also state what pattern of results they would expect if CA3 is not needed for remote memories? I.e., what would Figure 4 and Figure 5 look like? We suggest reworking the phrasing to head off similar confusions on the part of readers.

2) Regression model predicting internal details from brain measures:

- The details of the final regression model, where internal details were predicted from path length and CA3 volume, should be clarified. It seems that this model was run on both amnesics and controls; was group included as a factor in the model? Considering path length and CA3 volume were already shown to express group differences between amnesics and controls, the regression model should include group as a factor to account for overall group differences. Alternatively, the regression analysis can be run separately per group. Please clarify how the model was designed.

- If the analysis was done with both groups, how do we know the effect isn't primarily carried by controls? It wouldn't be a problem per se if it was, but the claim in the paper does not seem to be about normal brain variation in the healthy population. For example, in the Discussion section, it is stated, "the contribution of human CA3 to episodic retrieval was explained not only by the variability in CA3 volume but also how the insult perturbed functional network topology of the DN". "How the insult perturbed functional network topology" implies that it is specifically the change from normal, in the patient group, that matters. To support that, shouldn't the analysis be done for the patient group only, on the difference between their path length and the mean of controls?

- On a similar note, please show the scatter plot for internal details ~ CA3 volume. I understand that this scatter plot would not reflect the same regression as the model, but it seems an obvious and direct analysis. Path length is a fairly complex measure and we have little intuition about how it might share variance with CA3 volume; we would like to see the direct relationship between CA3 volume and internal details, even if it isn't a significant correlation.

- Related to the point above: the nodes for the multiple regression analysis were chosen to be nodes that showed between-group differences (subsection “CA3 volume and a loss of integration in retrosplenial cortex were predictive of autobiographical episodic retrieval across the lifespan”). There are also between-group differences in memory. This increases the concern that this analysis might be problematic, because it might be overly influenced by the group difference between patients and controls, rather than variation in the patients, which is what seems to be implied by "how the insult perturbed functional network pathology". Can the authors alleviate this concern? Can the method also be clarified, e.g., when it is stated that the variables of interest were z-scored, how were they z-scored? If both patients and controls were included in the analysis, were the values z-scored based on the mean across both groups? Also, were there outliers, how many, and from which groups did they come?

- Perhaps in Supplemental Material, we would like to see a table depicting the multiple regression model and all the significant and non-significant predictors.

- Please put the degrees of freedom when reporting all statistics (not just those described in this point).

3) In several instances, we wished for clarity regarding how time intervals were used in analyses:

- The time intervals tested for memory contain both retrograde (0-55 years) and anterograde (last year) components. The debate about the involvement of the hippocampus for recent vs remote memories is about its involvement in the retrograde domain. These models typically all assume that the hippocampus is needed to acquire new autobiographical / episodic memories (though there is some evidence that the hippocampus is not always needed for at least some types of new visual memories; see Froudist-Walsh et al., 2018). Because the paper is set up to focus on the retrograde domain (e.g., the first paragraph mentions a debate about the role of the hippocampus in remote memories), then it is not clear why the anterograde time point is often lumped in with the others. For example, it is included with the others in a 3-way ANOVA (subsection “Autobiographical memory: Amnesia for recent and remote episodic detail”) and also in the multiple regression analyses relating CA3 volume and path length measures to memory. Most models will agree that the anterograde timepoint will be impaired; the difference is whether the retrograde memory impairment is flat or graded. If the authors are testing those models, then the reasoning why the anterograde time interval is sometimes lumped in with the others should be clarified.

- Subsection “Autobiographical memory: Amnesia for recent and remote episodic detail”: When describing the absence of a significant effect for "retrograde events", does this refer to all retrograde events, collapsed across interval?

- In subsection “Autobiographical memory: Amnesia for recent and remote episodic detail” the authors state that: "Furthermore, there was an absence of between-group differences in the contrast between internal and external detail for the 0-11 year memories (F(1,30) = 1.68, p = 0.205) that is consistent with the view that early memories are gist-like…" A between-group comparison seems circular here, because it uses the authors' interpretation – that the hippocampus is not required for gist memory – as its premise. Please clarify.

4) Hippocampal functional connectivity: The authors were interested in how damage to CA3 could affect downstream network connectivity. An intuitive place to start, before getting to graph theory, would be to look at hippocampal seeded whole-brain functional connectivity maps. These maps can be examined for each group, and also compared across the amnesics and controls to reveal how damage to the CA3 affects functional coupling between the hippocampus and the rest of the brain. This point is especially important considering the hippocampal formation ROI from the Andrews-Hanna coordinates used for the graph analyses is quite anterior, close to the junction between the hippocampal head and amygdala. Therefore, the present results may not adequately characterize changes in functional connectivity surrounding the hippocampus.

5) Stability of effects: Thresholding choices can have considerable effects on graph metrics. The authors mention having conducted analyses revealing the stability of their results, but provide only general comments ("the results revealed overlap") regarding the analyses. Please report results from the stability analyses (e.g., quantify the overlap), perhaps in the supplemental materials.

6) CA3 damage does not affect network structure outside of the default mode: The authors report that no differences were found across any of their graph metrics when looking at resting state networks beyond the default mode network. No statistical measures or results are provided to accompany this statement. This is also the case for non-significant metrics and ROIs from the DMN. It would be beneficial to include these details (e.g., p-FDR and beta values, perhaps as a table in the Supplementary Materials).

7) Implications of increased local efficiency in amnesics:

- Interestingly, amnesics showed increased local efficiency in a few ROIs (posterior cingulate, parahippocampal cortex, and retrosplenial cortex). In other words, amnesics show more robust local functional network structure, surrounding these specific ROIs, than healthy controls. The results regarding path length are intuitive and are discussed, but the implications for greater local efficiency in the amnesic group warrants some discussion.

- In general, we thought that more clarity could be provided with respect to the graph theoretic analyses. For example, the sentence in subsection “Integration within MTL subsystem was reduced by damage to the human CA3 network” describing the DVs is very hard to follow ("In particular, the following widely used graph metrics were estimated for each of 20 core nodes (ROIs) that comprised the two subsystems and mainline core: global efficiency (i.e., integrative information transfer); average path length (inversely related to the efficiency of information exchange over a network) and clustering coefficient (a.k.a., transitivity) were used to examine network integration and the functional specificity of regional brain areas, respectively whereas, local efficiency, degree (number of connections maintained by a node), and betweenness centrality (number of short communication paths that a node is a member) were computed as regional measures."). I suggest stating more directly what each graph metric was interpreted to reflect. But more importantly, how many metrics/DVs were analysed for each node? Is there a problem of multiple comparisons here?

8) While we appreciated the care taken to characterize their patient sample, a critical claim of the current paper is that this pattern of results is due to selective CA3 lesions, so the bar is very high. To this end, we have a few comments and requests for clarification:

- We would like to see a graph that shows individual participant data (patients and controls) for the individual subfields. Is it the case that one amnesic participant does not have appreciable CA3 volume loss? Note that the y-axis on Figure 2 is obscured by the label, adding to confusion. Does this individual have altered functional connectivity? Does this individual show signs of memory impairment? We assume that repeating the analyses without this patient does not change the results? If we are misunderstanding the axis, please clarify.

- The neuropsychological results, as presented in Figure 6 and Supplementary file 1, indicate that the patients are indistinguishable – or superior – than their controls. A more fulsome commentary in the text on this issue seems appropriate, because at first glance, the patients don't seem to be particularly amnesic. This might be a good opportunity to highlight the fact that the patients were impaired on the assessments of delayed verbal recall. It would be helpful to readers, and to future researchers, if the authors could provide some additional detail regarding the extent and quality of amnesia associated with this patient group. For example, including some sample transcripts that are representative of their recall, relative to the age-matched controls, would be very informative.

- In subsection “Quantitative hippocampal subfield morphometry” the authors indicate that "Results from one participant in the amnesic group was not available because it was not possible to segment the five hippocampal subfield across the entire longitudinal axes of both hippocampi, but the participant exhibited bilateral hippocampal volume loss compared to the control group mean." Additional details regarding why this was the case (e.g., issue with segmentation software?) would be helpful, and further information is necessary to convincingly argue that this participant has selective CA3 damage and thus warrants inclusion in the current sample.

9) We provide the more general suggestion that the authors present the reader with an overarching analysis plan at the outset of the Results section. I believe that this would help to mitigate the impression that the various tests were not motivated or coherently linked.

[Editors' note: further revisions were requested prior to acceptance, as described below.]

Thank you for resubmitting your work entitled "Human hippocampal CA3 damage disrupts both recent and remote episodic memories" for further consideration at *eLife*. Your revised article has been favorably evaluated by Laura Colgin (Senior Editor), a Reviewing Editor, and two reviewers.

The manuscript has been improved but there are some remaining issues that need to be addressed before acceptance, as outlined below.

We appreciated the thoroughness of the revisions and believe that the overall clarity of the manuscript has significantly improved. We continue to be impressed by the careful characterization of this large and rare patient population, as well as the rigorous approach taken in characterizing the autobiographical memories. However, the new analyses raised some additional concerns, which we describe below (note that numbering corresponds to the original review synthesis statement).

1) Our primary concern is with respect to excluding the 0-11 bin. In the original submission this time bin was critical to the main conclusions of the paper, yet in this subsequent revision the authors argue that this time bin should be dropped. For starters, we believe that it is essential that the exclusion of the earliest time bin is clearly described as post hoc and some of the stronger claims be tempered. Visual inspection of Figure 4 suggests that there is a time-sensitive deficit (i.e., the opposite of what the paper argues), and the significant interaction in the new 2 x 5 ANOVA supports this claim. As such, the primary conclusion of the paper hinges on a null interaction from a post hoc 2 x 4 ANOVA that had reduced power due to the dropped time bin.

These concerns aside, we do agree that there is a valid argument that these oldest memories might have been schematized to a state that renders them qualitatively different from the more recent (yet still old) memories. Is there evidence for this 'extreme schematization' in the current data? For example, do controls show an elevated external:internal ratio for this time bin? Any such analysis would be obviously post hoc and should be described as such, but we believe that clearer justification for this analytic decision would strengthen the resulting claims, particularly if your own data show that the oldest memories are indeed more schematized than the others.

Secondly, the finding of reduced internal details for the 11-18 bin offers strong evidence against systems consolidation, yet the only direct comparison reported between groups was for the 0-11 bin. Is there a group difference at the 11-18 bin?

4) Hippocampal functional connectivity.

4.1) The new analyses report that when whole-brain voxel-level functional connectivity maps are calculated using left and right hippocampus separately, there are no differences found between patients and controls anywhere in the brain. This is quite surprising given that the patients are amnesic. We would like to see the left and right hippocampal functional connectivity maps for each group. If hippocampal connectivity is truly completely unaffected in CA3 amnesic patients, that seems like a rather interesting finding -- counter to some previous findings, as the authors note in their Discussion -- that warrants visualization as a map. It also would suggest that graph metric differences between groups arise largely from disruptions in connectivity between regions other than the hippocampus.

A group difference map of left and right hippocampal connectivity would be informative as well; even though no clusters survived FDR correction, a more leniently thresholded difference map might reveal the qualitative pattern of altered hippocampal connectivity.

We believe displaying these maps would add substantial value to the paper, as connectivity maps are comparable and relevant to a large number of studies in the literature. The graph metrics are all derived from the connectivity measures, so insight into the graph metrics can be obtained by examining the connectivity maps.

For example, one possible reason that path length is higher in the patient group is if fewer edges are above threshold overall in the connectivity matrix of that group (not specific to the hippocampus). How extensive are the regions exceeding the z>0.84 threshold (which is taken from another paper) in this dataset? We understand that, as the authors write in the Discussion, "seed-based analyses do not assess network connections between ROIs simultaneously, which is necessary when studying functional integration and segregation at the scale of whole-brain networks". Nonetheless, significant path length increases are reported for left and right hippocampus (Discussion section). It is helpful for the reader be able to see these basic properties of the data.

4.2) We wish for more clarity on the logic of selecting occipital pole and primary motor cortex as "control" regions for the hippocampus-to-ROI analysis. The hippocampus tends to have low correlations with voxels in the visual cortex and somatosensory cortices in healthy controls, so why would one expect a difference between groups in those regions? It's a comparison of noise with noise, and thus uninformative. It would be more useful to see hippocampus-to-ROI functional connectivity with nodes in the DN or MTL. This might be called a "control" analysis in the sense that if the whole-brain voxel-level connectivity map threshold (4.1 above) was quite stringent and that's why no clusters survived, regions in DN or MTL that usually *are* correlated with hippocampus can be tested to see whether there is still no difference between groups. Again, it is surprising (though not impossible) that there are no differences at all between groups in hippocampal functional connectivity, given that (a) the hippocampus is damaged in the patient group, and (b) there are differences between groups in the graph metrics based on that connectivity.

4.3) The right hippocampus is also referred to as a "control" region in the seed-based functional connectivity analysis. We found this confusing, as the patients have bilateral hippocampal damage. It is unclear exactly what analyses were done in subsection “Seed-based functional connectivity analyses”: are these correlations between (a) left hippocampus and (b) the three "control" regions (right hippocampus, visual network, and somatomotor network), or correlations between (a) the right hippocampus and (b) the visual network, and somatomotor network?

8) In Figure 3 there appears to be substantial CA1 volume loss in the patients, in addition to the CA3 volume loss which is the main focus of the paper. We understand that CA1 volume was not statistically different between groups. However, given that the claims of the paper with respect to retrograde amnesia hinge on the CA1 being preserved while CA3 is selectively damaged, perhaps further discussion of CA1 volume loss is warranted. At a minimum the mean reduction% , F, p, and d should be reported, as they are for CA3.

---

## [Author Response]

We all agreed that the results are important to the field of memory, and we judged the paper to be of broad interest. However, we agreed on several major concerns, largely regarding how the analyses were conducted and the conclusions that can be drawn from them. These are detailed below.

We thank the Reviewing Editor, Senior Editor, and the three reviewers for their detailed and constructive feedback on our manuscript. We have endeavoured to address the issues that were raised and provide point-by-point responses that indicate where revisions have been made to the main text, Appendix 1, Supplementary files, and Figure Supplements.

Essential revisions:1) The authors' main claim is that, following CA3 lesions, episodic memory is equally susceptible to impairment no matter how long ago the memories were acquired. This claim seems to rest on a comparison of memory for 0-11 years with 30-55 years, and 0-11 years with the preceding year, separately for patients and controls. Why were only those comparisons chosen, and not others? Moreover, we do not think that the chosen analyses are ideal for depicting a flat temporal gradient of amnesia for the patients. Looking at Figure 4, the amnesic patients' deficit seems to grow with time: the patient vs control difference for 0-11 years seems to be much smaller than the patient vs control difference for the last year. Did the authors specifically test for whether the difference between patients and controls changes over time? The 3-way interaction between group, type of detail, and time does not clarify this: instead, the authors can look for a two-way interaction between group and time, just for internal details. In other words, the lack of a change in patients' internal details over time does not indicate that memory was equally impaired across time, because the controls' memory was better for recent vs remote events. If patients' scores are relatively flat, and controls' memory is better for recent events, that suggests that patients' impairment (relative to controls) increases as memories get more recent. This is a speculation based on the figure, but the authors should directly test it by looking for a 2-way interaction between group and time for internal details. If there is a main effect of group and no group x time interaction, it would bolster the claim that memory is equally impaired for all time periods. If there is a group x time interaction, then the main conclusion of the paper is not supported.

In light of the considerations raised, we agree that the original approach did not include adequate analyses to examine the loss (amnesic group vs control group) of internal and external detail over time. In brief, the requested new analyses indicate that a key conclusion expressed in the title, “Human hippocampal CA3 damage disrupts both recent and remote episodic memories" holds for a duration lasting up ~50 years.

Crucially, the new analyses demonstrate that the deficit in the amnesic group (relative to controls) increases with recency of the memories, but the greater loss of detail for recent as compared to remote memories was driven by the most remote (and intact) memory (0-11 years). The latter is notable given that, as discussed in the original manuscript and extensively elsewhere, very early remote memories are arguably qualitatively different from other remote memories (e.g., Akhtar et al., 2018; Barclay and Wellman, 1986; Cermak, 1984; Sekeres, Winocur and Moscovitch, 2018; Winocur and Moscovitch, 2011), and are less vulnerable to the effect of hippocampal damage due to the transformation of the events into generalised schema representations.

The observed temporal extent of internal detail loss in the remote memories is at variance with the extant experimental data on CA3 from model organisms and exceeds the predicted durations of involvement based on standard systems consolidation and computational frameworks, where the hippocampus is hypothesised to have a time-limited role in episodic memory (e.g., Dudai, 2004; Frankland and Bontempi, 2005; McClelland et al., 1995; Squire et al., 2015; Treves and Rolls, 1994). Hence one key conclusion that emerged from these new analyses is that, with the exception of the earliest intact remote memory, human CA3 is no less essential for the retrieval of autobiographical episodic memories as time passes after learning.

More specifically, retrograde amnesia was observed in the amnesic group up to ~50 years prior to the CA3 damage, whereas, for example, durations up to ~15 years and 1-5 years have been considered as consistent with systems consolidation (Bayley, Hopkins, and Squire, 2006; Rempel-Clower et al., 1996, respectively). Furthermore, retrograde amnesia beyond these intervals – e.g., 25 years – has been argued to require extra-hippocampal damage that includes regions such as the entorhinal cortex (Rempel-Clower et al., 1996), whereas we observed that retrograde amnesia of up to ~50 years was associated with highly focal hippocampal damage.

To summarise the results of the requested analysis (subsection " Autobiographical memory: amnesia for recent and remote episodic detail”), we conducted a 2 (group: amnesic, control) x 5 (time: past year (i.e., anterograde interval); 30-55, 18-30, 11-18, and 0-11 years (i.e., retrograde intervals)) mixed-model ANOVA on the internal details to test for main effects and the significant two-way interaction between group and time. In addition to a significant main effect of group (*F*_(1,30)_ = 16.37, p<0.0001, η^2^_p_ = 0.353), there was a significant two-way interaction between group and time (*F*_(2.67, 80.22)_ = 3.91, p=0.015, η^2^_p_ = 0.115). Therefore, the loss of internal (episodic) detail in the amnesic group decreased with the age of recalled memory when all five sampled intervals were considered. This effect is consistent with the effects for, example, of larger lesions associated with temporal lobe resection (Noulhiane et al., 2007), where loss is still observed for recent and very remote episodic memories.

However, this analysis prompted us to think about the same question with an additional test that adds necessary nuance when interpreting the loss of internal detail relative to the control group. First, unlike Noulhaine et al., (2007) and the effects of human CA1 damage (Bartsch et al., 2011), internal (episodic) detail was preserved for the earliest remote memory (0-11 year interval, *F*_(1,30)_=0.250, *p*=0.621); put another way, retrograde amnesia was absent for the earliest remote memory. Thus, if the earliest remote memory is, as suggested in prior experimental and theoretical work, qualitatively different and not “hippocampal (CA3)-dependent”, then a re-analysis of the between-group difference in internal (episodic) detail for the remaining memories may yield a different pattern of loss associated with CA3 damage, whereby the loss of internal (episodic) detail does not reduce over time for hippocampal-dependent memories and the result involving five intervals is driven by the earliest, intact remote memory.

The next set of analyses demonstrate that the change in loss of internal detail was driven by the earliest (intact) remote memory. Specifically, 2 (group: amnesic, control) x 4 (time: past year (i.e., anterograde interval); 30-55, 18-30, and 11-18 years (i.e., retrograde intervals)) mixed-model ANOVA was conducted to determine whether or not the loss of internal (episodic) detail decreased with the age of the memories when the earliest remote memory was excluded. Notably, there was a significant main effect of group consistent with a loss of internal detail in the amnesic group (*F*_(1,30)_ = 23.25, p<0.0001, η^2^_p_ = 0.437), whereas the two-way interaction between group and time and main effect of time were *not* significant (*F*_(2.62, 78.44)_ = 1.51, p=0.222, η^2^_p_ = 0.048; *F*_(2.62, 78.44)_ = 0.604, p=0.592, η^2^_p_ = 0.020, respectively). Hence, the internal (episodic) detail loss was equal and ungraded for recent and remote memories when the earliest remote memory was excluded.

As outlined below in our response to a later comment about the need to equate the analysis strategy between internal and external details, we note here that a companion 2 (group: amnesic, control) x 5 (time: past year (i.e., anterograde interval); 30-55, 18-30, 11-18, and 0-11 years (i.e., retrograde intervals)) mixed-model ANOVA conducted on external details demonstrated the specificity of the deficit in internal detail, because there was no significant between-group difference in the amount of external (semantic) detail over the five intervals (*F*_(1,30)_ = 1.24, p=0.275, η^2^_p_ = 0.040) (Figure 5). Moreover, the interaction between group and time for external detail was not significant (*F*_(4,120)_ = 1.46, p=0.218, η^2^_p_ = 0.046) nor was the main effect of time (*F*_(4,120)_ = 1.46, p=0.275, η^2^_p_ = 0.046).

Considered together, these revised analyses have fewer factors than applying the original pairwise contrast analysis strategy to examine both within-group and between-group differences over time, while also directly speaking to the issue of the specific of pattern of loss in internal detail relative to the control group. We thus hope that reporting these more focused analyses better capture the specific nature of the loss over time associated with focal CA3 damage relative to the control group (subsection "Autobiographical memory: amnesia for recent and remote episodic detail”).

In addition, as detailed below, we also show that the ungraded loss for all but the earliest remote memory holds even if we confine the analyses to the remote memories, so the results are not driven by inclusion of the recent memory in the analyses. More generally, we have revised the manuscript throughout to reference the specific pattern of loss found in these new analyses and consider it against prior reports involving more extensive hippocampal damage and damage involving the hippocampus and other MTL subregions.

In a very similar vein, just put slightly differently: The authors state that "Deficits were hypothesised to affect internal details independently of the age of recalled events" (Introduction) and "Episodic loss of detail in the amnesic group for events did not vary over time (~1- 60 year interval)" (Figure 4 caption). This phrasing led to some confusion, because the difference between patients and controls for internal details does vary with the age of recalled events: controls show a gradient over time and patients don't, thus the difference varies (decreases) with the age of the memory. A "loss" of details in the patients implies that this is relative to the controls. After some closer reading it seems that the intended meaning was something more like "both recent and remote memories were affected" by CA3 damage. The ANOVA associated with Figure 4 reports differences between groups "across all five time intervals", but the between-time-bin comparisons (eg, 0-11 vs. last year) are reported within-group only. For clarity, could the authors please elaborate on these issues, and also state what pattern of results they would expect if CA3 is not needed for remote memories? I.e., what would Figure 4 and Figure 5 look like? We suggest reworking the phrasing to head off similar confusions on the part of readers.

We apologise for the lack of clarity. We have revised the text to state that both recent and remote memories were hypothesised to be affected by the focal and bilateral CA3 damage. In addition, we should have qualified the tests that were conducted in light of our original discussion about the earliest remote memory (i.e., 0-11 years), where we noted the, “view that early memories are gist-like (Hardt et al., 2013; Richards and Frankland, 2017; Sadeh et al., 2014), involving a loss of contextual specificity over time, such that these memories become more ‘semanticised’ than the other memories. … and can be supported by extra-hippocampal regions (Wiltgen et al., 2010; Winocur et al., 2010, 2007).”

In light of the results from the new analyses, we have extensively reworked the Figure 4 caption to align with the results in the main text. In particular, the phrasing has been re-worded to first state that remote and recent memories were affected by CA3 damage (up to ~50 years prior to the CA3 damage) and then explain the specific nature of the loss of internal detail in terms of how it was absent for the oldest remembered events, when five intervals were considered, but is consistent with ungraded loss when the earliest (oldest), intact remote memory was excluded. We hope that this more qualified summary for the Figure 4 caption captures the nature of episodic detail loss found in the revised analyses.

We also now state in the introduction that there would be no difference in internal details between controls and amnesic group participants for the retrograde interval if CA3 is not required for remote episodic memories (Introduction), along the lines reported in studies involving model organisms (e.g., Lux et al., 2016) and predicted by systems consolidation and computational frameworks. The legend associated with Figure 5 has been updated with the results of the companion 2 x 5 mixed-model ANOVA that was conducted to assess between group differences in external detail over time.

2) Regression model predicting internal details from brain measures:- The details of the final regression model, where internal details were predicted from path length and CA3 volume, should be clarified. It seems that this model was run on both amnesics and controls; was group included as a factor in the model? Considering path length and CA3 volume were already shown to express group differences between amnesics and controls, the regression model should include group as a factor to account for overall group differences. Alternatively, the regression analysis can be run separately per group. Please clarify how the model was designed.

The original multiple regressions analyses were conducted without including group as a factor in the model. The reviewers make an excellent point that it is important to know whether links between average path length and internal detail hold when analysed either with group as a factor or at the level of group.

In line with the suggestion below, “To support that, shouldn't the analysis be done for the patient group only, on the difference between their path length and the mean of controls?” we conducted a multiple regression analysis for the difference in average path length for the affected nodes from the mean of the controls and internal details. The results indicate that average path length in the left PHC (*β* = 1.05, *t* = 4.69, p=0.018), left hippocampal formation and right hippocampal formation (*β* = 1.81, *t* = 7.87, p=0.004, *β* = -1.55, *t* = -8.66, *p*=0.003, respectively), left retrosplenial cortex (*β* = 26.61, *t* = 12.12, p=0.001), vmPFC (*β* = -26.31, *t* = -11.55, p=0.001), and left temporal pole (*β* = -1.70, *t* = -10.26, p=0.002) were predictive of the amount of internal detail that was remembered. These results and the other non-significant independent variables are reported in subsection "Seed-based functional connectivity analyses” and in Supplementary file 1p.

- If the analysis was done with both groups, how do we know the effect isn't primarily carried by controls? It wouldn't be a problem per se if it was, but the claim in the paper does not seem to be about normal brain variation in the healthy population. For example, in the Discussion section, it is stated, "the contribution of human CA3 to episodic retrieval was explained not only by the variability in CA3 volume but also how the insult perturbed functional network topology of the DN". "How the insult perturbed functional network topology" implies that it is specifically the change from normal, in the patient group, that matters. To support that, shouldn't the analysis be done for the patient group only, on the difference between their path length and the mean of controls?

The original analysis was conducted without group as a variable and thus the results could, indeed, have been carried by the control group. As per suggestion, we have re-assessed the affected ROIs/nodes on the difference between their average path length and the mean of the control group against internal detail. We believe that this analysis examines whether or not the differences in average path length for the affected nodes can be used to predict (the diminished) internal detail. Crucially, the results indicate that the average path length in the left PHC (*β* = 1.05, *t* = 4.69, p=0.018), left hippocampal formation and right hippocampal formation (*β* = 1.81, *t* = 7.87, *p*=0.004, *β* = -1.55, *t* = -8.66, p=0.003, respectively), left retrosplenial cortex (*β* = 26.61, *t* = 12.55, p=0.001), vmPFC (*β* = -26.31, *t* = -11.55, *p*=0.001), and left temporal pole (*β* = -1.70, *t* = -10.26, p=0.002) were significant predictors of internal detail. The same approach was applied to analyse whether or not the differences in local efficiency of the affected nodes from the means of the control group predicted internal detail. The results indicate that local efficiency differences from the mean of the controls in the affected nodes did not predict internal detail. We have updated the relevant section of the results with these new analyses (subsection "Seed-based functional connectivity analyses”), extensively revised the Discussion section, and have included the regression model for each of these analyses in Supplementary file 1p (average path length) and Supplementary file 1q (local efficiency).

- On a similar note, please show the scatter plot for internal details ~ CA3 volume. I understand that this scatter plot would not reflect the same regression as the model, but it seems an obvious and direct analysis. Path length is a fairly complex measure and we have little intuition about how it might share variance with CA3 volume; we would like to see the direct relationship between CA3 volume and internal details, even if it isn't a significant correlation.

We have revised the manuscript and have incorporated the requested scatter plot for total internal detail against CA3 volume (Figure 4—figure supplement 1). It is a known issue that sample size and variability are key considerations when looking for associations between variables. Thus, we feel that a fair and complete way to present these data are to show two scatterplots: (a) plot each group separately using coded points (Individual data points: dark grey diamond = amnesic group participant; grey circle = control group participant) and (b) plot participants from both groups in order to include a linear regression line with 95% confidence interval. The solid black line shows the best-fitting linear regression line and the dashed lines show 95% confidence intervals (*β* = 0.18, *t* = 2.05, p=0.049). This set of plots allows the reader to determine whether the variability in CA3 represents two distinct groups vs continuous variability in a structure that still supports, albeit quantitatively reduced (for 4 intervals), remembering/reconstruction of internal detail relative to the control group. This question goes beyond the scope of the current study and will be an important consideration in future work, but we believe that is it instructive to plot both graphs. Reference to this new figure is included in the Results section and the figure caption for Figure 4—figure supplement 1.

To summarise the inferential analyses, “At a single group level, the correlation between internal details and CA3 volume was not significant either for the amnesic group (Kendall’s *τ*_(15)_ = -0.018, p=0.961, two-tailed) nor for the control group (Kendall’s *τ*_(15)_ = 0.43, p=0.458, two-tailed). CA3 volume and total internal detail were significantly correlated (Kendall’s *τ*_(30)_ = 0.283, p=0.028, two-tailed; Figure 4—figure supplement 1), such that lower CA3 volumes were associated with remembering less internal (episodic) detail on the AI, only when these variables were collapsed across group.” As a result of conducting the requested plots and analyses of the link between CA3 volume and internal details, the multiple regression analyses related to the graph theoretic measures do not include CA3 as an independent variable, which avoids introducing additional considerations about handling anatomical and functional variables in the same regression model.

In perhaps the most immediate comparison available from the relevant literature, the relationship between autobiographical episodic retrieval and human CA1 damage associated with transient global amnesia was not tested/reported by Bartsch et al., (2011). We have also extensively revised the discussion to contextualise the results from these new results:

“CA3 volume was not correlated with the amount of internal (episodic) detail remembered either by the amnesic group nor the control group. One possible interpretation is that sample size and variability of the data may account for the failure to detect a significant association at the group level, because these variables were positively correlated if considered as continuous distributions and collapsed across the two groups. CA3 along with CA2 and the DG are input structures, typically associated with encoding, whereas CA1 is an output structure, implicated in retrieval of events within their temporal context (Preston et al., 2010; Suthana et al., 2011; Zeineh et al., 2003). It is unknown whether residual CA1 volume in participants with transient global amnesia was linked with autobiographical episodic retrieval because the association was not reported (Bartsch et al., 2011). In healthy adult participants, there has been a failure to find an association between CA1 volume and episodic retrieval (Mueller et al., 2011), whereas CA3 volume correlates with the efficacy with which newly formed memories are differentiated (Chadwick et al., 2014). The latter effect was interpreted to reflect either decreases in retrieval confusion due to an increased number of CA3 neurons or enhanced lateral connectivity that could lead to improvements in pattern separation (Chadwick et al., 2014). fMRI studies based on laboratory-learned materials suggest time and space context are represented across multiple human hippocampal subfields (Copara et al., 2014), with differentiation between spatial and temporal context mediated by distinct neural network patterns (i.e., multiplexed) rather than individual structures (Kyle et al., 2015). Hence links found with single subfield volumes and autobiographical episodic retrieval are likely to be challenging to interpret. Volumetric analyses that collapse across large subregions have found that, for example, residual bilateral MTL volume following brain injury correlates with remote autobiographical memory (Gilboa et al., 2005), whereas in a study on participants who underwent temporal lobe resection for epilepsy, right parahippocampal cortical volumes correlated with remembering remote episodes and bilateral MTL regions predicted memory for recent episodes (Noulhiane et al., 2007).”

- Related to the point above: the nodes for the multiple regression analysis were chosen to be nodes that showed between-group differences (subsection “CA3 volume and a loss of integration in retrosplenial cortex were predictive of autobiographical episodic retrieval across the lifespan”). There are also between-group differences in memory. This increases the concern that this analysis might be problematic, because it might be overly influenced by the group difference between patients and controls, rather than variation in the patients, which is what seems to be implied by "how the insult perturbed functional network pathology". Can the authors alleviate this concern? Can the method also be clarified, e.g., when it is stated that the variables of interest were z-scored, how were they z-scored? If both patients and controls were included in the analysis, were the values z-scored based on the mean across both groups? Also, were there outliers, how many, and from which groups did they come?

As noted, we agree that the observed effects in original multiple regression analysis may have been carried by the control group because group was not included as a variable. As a consequence and following the suggestion above, we have revised the analyses to examine the affected nodes against internal detail for the amnesic group to alleviate the concern, basing the analysis on the difference from the mean of the controls rather than z-scores.

We now report the results of a robust multiple regression analysis on “patient group only, on the difference between their path length and the mean of controls?” No outliers were identified in the revised analysis based on the difference between average path length of the amnesic group participants and the mean of the controls. The original analysis was conducted using a *z*-transformation was based on the control group mean and standard deviation, such that mean of the control group was subtracted from the average path length of each participant and divided by the standard deviation of the control group (Jones et al., 2016). The main text has been revised to include the results from the suggested new approach for each multiple regression analysis are now also reported (subsection "Seed-based functional connectivity analyses”).

- Perhaps in Supplemental Material, we would like to see a table depicting the multiple regression model and all the significant and non-significant predictors.

Supplementary Materials/files have now been updated to include the requested tables (Supplementary file 1p-1q). Results from the multiple regression analyses report values for the significant and non-significant predictors.

- Please put the degrees of freedom when reporting all statistics (not just those described in this point).

We have revised the main text, Supplementary files and Appendix 1 to report degrees of freedom.

3) In several instances, we wished for clarity regarding how time intervals were used in analyses:- The time intervals tested for memory contain both retrograde (0-55 years) and anterograde (last year) components. The debate about the involvement of the hippocampus for recent vs remote memories is about its involvement in the retrograde domain. These models typically all assume that the hippocampus is needed to acquire new autobiographical / episodic memories (though there is some evidence that the hippocampus is not always needed for at least some types of new visual memories; see Froudist-Walsh et al., 2018). Because the paper is set up to focus on the retrograde domain (e.g., the first paragraph mentions a debate about the role of the hippocampus in remote memories), then it is not clear why the anterograde time point is often lumped in with the others. For example, it is included with the others in a 3-way ANOVA (subsection “Autobiographical memory: Amnesia for recent and remote episodic detail”) and also in the multiple regression analyses relating CA3 volume and path length measures to memory. Most models will agree that the anterograde timepoint will be impaired; the difference is whether the retrograde memory impairment is flat or graded. If the authors are testing those models, then the reasoning why the anterograde time interval is sometimes lumped in with the others should be clarified.

We emphasised the retrograde interval because this is, indeed, where dominant accounts of memory consolidation diverge, but it was not intended to be at the expense of considering the role of CA3 in recent autobiographical episodic memory. Indeed, we included several papers in the introduction that have only looked at the role of human CA3 in newly formed and recent memories (e.g., Chadwick et al., 2014), and, the neuroimaging studies of autobiographical memory that we noted included contrasts between recent and remote memory.

Analyses of all intervals acquired on the AI using an omnibus ANOVA aligns with the factorial design of our experiment and with prior studies that have asked related questions about recent and remote autobiographical memory (e.g., Noulhiane et al., 2017). We agree that the structure of the planned comparisons unintentionally gave a misleading view, but our analysis strategy has now been revised to examine (1) the loss of detail over time relative to the control group and (2) we report below the reliability of these results by examining the loss and specificity for the remote memories alone; results from these latter analyses demonstrate that the effects are not driven by including the anterograde/recent (hippocampal-dependent) memory:

As noted in our response to point 1, we have added revised follow-up analyses in line with the suggestion to test for a 2 x 5 interaction between group and time that includes the anterograde interval. Notably, the conclusions that can be inferred from these revised analyses are the same as those found in an exploratory mixed ANOVA analysis that excludes the last year. To examine the impact of including the anterograde memory, a 2 (group: amnesic, control) x 4 (time: 30-55, 18-30, 11-18, and 0-11 years (i.e., retrograde intervals)) ANOVA on interval details for only retrograde details revealed a significant main effect of group (*F*_(1,30)_ = 9.27, p=0.005, η^2^_p_ = 0.236) and a significant two-way interaction between group and time (*F*_(3,60)_ = 3.11, p=0.030, η^2^_p_ = 0.094), whereas time was not significant (*F*_(3,90)_ = 0.67, p=0.573, η^2^_p_ = 0.022) – degrees of freedom were not adjusted because Mauchly’s test revealed that sphericity was not violated (χ^2^_(5)_ = 6.87, p=0.231). Further re-analysis of internal details to determine the impact of the earliest, intact remote memory on the group by time interaction revealed that the two-way interaction between group and time (*F*_(2,60)_ = 0.19, p=0.825, η^2^_p_ = 0.006) and main effect of time (*F*_(2,60)_ = 0.10, p=0.898, η^2^_p_ = 0.003) were not significant, whereas the main effect of group (*F*_(1,30)_ = 13.19, p=0.001, η^2^_p_ = 0.305) was significant. Put another way, the impact CA3 damage on the ability to remember spatio-temporal contextual features of autobiographical events did not vary with remoteness.

We have revised the main text to state in the introduction that most accounts of memory consolidation predict deficits in anterograde/recent memory following hippocampal damage. In the data, we are able to demonstrate that human CA3 damage impairs recent and even very remote autobiographical episodic but not semantic memory. This result is not compatible with systems consolidation or evidence from extant experimental work on CA3 involving model organisms.

There are several other reasons to include the recent memory in all of the main analyses:

First, it would be difficult to understand the nature of the remote memory pathology without the post-lesion baseline – in the form of the anterograde/recent memory – for recently formed autobiographical memories that have also undergone consolidation. Given the consensus about the hippocampal-dependence for the anterograde/recent memory, the inclusion of this memory enables us to determine, by inference, if the magnitude of loss at each remote interval resembles that of a hippocampal(CA3)-dependent memory, particularly given the discrete known onset of pathological insult. Objective evidence on the group difference for the recent memory is the only interval in which loss could be benchmarked, particularly given how recent and remote memories were probed in the same way on the AI, and thus the design was setup to acquire similar memories, as far as possible, for all time intervals.

Second, the results from the anterograde/recent memory enabled us to demonstrate the necessity of *human* CA3 for episodic memories after the pathological insult, under conditions where we demonstrate that the loss of episodic memory is not secondary or conjoint with deficits in cognitive domains such as attention or executive function. In addition, several other studies that have examined the link between MTL or hippocampal damage have combined recent and remote intervals when examining the link between, for example, volumetric measures and autobiographical episodic memory performance and included remote and recent memories in all other analyses (e.g., Noulhiane et al., 2007).

Third, the approach aligns with studies on CA1 and CA3 involving experimental animals, such that by Lux et al., (2016), whereby the retrieval of recent memories were reported when tackling the question about the role of particular subfields for remote memory. By extrapolating the same overall approach to our analyses, we are able to determine if the results from model organisms overlap with those found following damage to human CA3.

Fourth, the anterograde/recent memory is sampled from an interval that spans the last year, which given that the memories can be up to one year old, provides insight into memory consolidation based processes for recent memories. Even though the duration of consolidation remains a topic for debate (Nadel and Moscovitch, 1997), changes in the neural bases of episodic memories can occur after a day has elapsed (Takashima et al., 200; cf. Stark and Squire, 2000) or even more rapidly (Marr, 1971), and recent evidence suggests that the recent memories sampled using the protocol applied here can undergo memory consolidation within 4 months (Barry, Chadwick, and Maguire, 2018). Indeed, systems consolidation based interpretations are applied in experimental work to account for the retrieval of memories after short and long delays.

For these reasons and the objective evidence for equivalence demonstrated when the inferential analyses are confined to the retrograde/remote interval, we would prefer to reported the main results that are based on including the anterograde/recent interval.

- Subsection “Autobiographical memory: Amnesia for recent and remote episodic detail”: When describing the absence of a significant effect for "retrograde events", does this refer to all retrograde events, collapsed across interval?

We apologise for the lack of clarity. Retrograde events did, indeed, refer to all retrograde events collapsed across the entire retrograde – remote memory – interval. However, as noted in our response above, we have conducted the requested analyses on internal details to examine how the loss varied across the sampled intervals and these results together with other new analyses are reported in revisions to the main text.

- In subsection “Autobiographical memory: Amnesia for recent and remote episodic detail” the authors state that: "Furthermore, there was an absence of between-group differences in the contrast between internal and external detail for the 0-11 year memories (F(1,30) = 1.68, p = 0.205) that is consistent with the view that early memories are gist-like…" A between-group comparison seems circular here, because it uses the authors' interpretation – that the hippocampus is not required for gist memory – as its premise. Please clarify.

We agree with the circular nature of the interpretation and the analysis has instead been conducted to test (1) whether the amount of internal detail retrieved for the earliest remote memory was comparable between the amnesic group and control group because this memory has been argued to be qualitatively different from other remote memories. In addition, we conducted an extension to the suggested revisions and tested whether earliest, intact remote memory was driving the effect of variable loss in internal detail over time. As noted earlier, the results indicate that this was the case.

4) Hippocampal functional connectivity: The authors were interested in how damage to CA3 could affect downstream network connectivity. An intuitive place to start, before getting to graph theory, would be to look at hippocampal seeded whole-brain functional connectivity maps. These maps can be examined for each group, and also compared across the amnesics and controls to reveal how damage to the CA3 affects functional coupling between the hippocampus and the rest of the brain. This point is especially important considering the hippocampal formation ROI from the Andrews-Hanna coordinates used for the graph analyses is quite anterior, close to the junction between the hippocampal head and amygdala. Therefore, the present results may not adequately characterize changes in functional connectivity surrounding the hippocampus.

In contrast to model-dependent seed-based functional connectivity and model-independent methods such as independent component analyses (Calhoun et al. 2001; Greicius et al. 2003), our intention with the graph-based network analyses was not only to visualize the overall connectivity pattern between brain regions (Smith et al., 2013, TiCS), but also to characterize the global organization and topological reconfiguration of brain networks following focal CA3 damage (Bassett and Bullmore, 2009; Bullmore and Sporns, 2009, Fornito et al., 2015; He et al., 2009).

As requested, we have conducted a hippocampal seed-based voxel functional connectivity analysis for a left hippocampal seed region that occurs within the main body rather than anterior region of the hippocampus. Specifically, between-group differences in functional connectivity were examined by entering a left hippocampal seed region (MNI co-ordinates -24, -22, -16), based on co-ordinates that have been shown to be implicated in episodic memory (Hirshorn et al., 2012), because the nodes proposed by Power et al., (2011) do not cover main body of the hippocampus. In addition, the analyses report results from three seed regions, a homotopic seed region in the right hippocampus, one in the occipital lobe of the visual network (as defined by Power et al.,), and one in the primary motor cortex of the somatomotor network (as defined by Power et al.). To summarise the new results, “… two sample independent I-tests revealed functional connectivity for the left and right hippocampal seed-regions were not significantly different between the amnesic group and control group, when assessed at a corrected p-FDR <0.05 threshold. The additional control analyses were consistent with these results because there were no significant differences between the amnesic group and control group for the right hippocampus and seed regions in the visual network and somatomotor network at p-FDR<0.05. For completeness, we also report all clusters at an uncorrected cluster-size threshold (p-uncorrected>0.05) in Supplementary file 1o.”

These results have been added to a new subsection in the Results and, as mentioned, in a new table in the Supplementary files (Supplementary file 1o). The detailed methods that we applied using the CONN toolbox to conduct these analyses are described in a new subsection of the Materials and methods section. We have also added a paragraph to the Discussion section to contextualise and interpret these new results:

“The seed-based analyses revealed null group differences in functional connectivity for left or right hippocampus seed regions and seed regions in visual and motor cortices. More extensive damage to hippocampus, parahippocampal gyrus and temporal pole can lead to significant differences in functional connectivity that are limited to the DN (Hayes et al., 2012). Other studies have observed that hippocampal damage can be associated with dysfunction in multiple large-scale brain networks, such that functional connectivity was decreased in the salience network and increased in the dorsal and ventral DN, sensorimotor, and higher visual networks relative to controls (Heine et al., 2018). These differences co-occurred with behaviour deficits in episodic and working memory, verbal and visual learning, semantic fluency, and executive function (Heine et al., 2018). The variability in outcomes may reflect that the correlation-strength map of voxels generated by seed-based approaches are particular to the average time series correlation with the seed region under analysis. Furthermore, seed-based analyses do not assess network connections between ROIs simultaneously, which is necessary when studying functional integration and segregation at the scale of whole-brain networks (van den Heuvel and Hulshoff Pol, 2010). More generally, since functional connectivity can reflect signaling that unfolds within the underlying structural network, future studies will need to examine anatomical pathways, but it is important to note that white matter connectivity appears to be unaffected following LGI1-antibody-complex LE (Finke et al., 2017).”

5) Stability of effects: Thresholding choices can have considerable effects on graph metrics. The authors mention having conducted analyses revealing the stability of their results, but provide only general comments ("the results revealed overlap") regarding the analyses. Please report results from the stability analyses (e.g., quantify the overlap), perhaps in the supplemental materials.

The results from the analyses conducted to test the stability of the effects are now reported in the Supplementary files, as a series of six new Tables for the DN and the five control networks (Supplementary files 1h-1m). These detailed results demonstrate that DN regions identified in the original analysis exhibited altered topology on the same graph theoretic measures (Supplementary file 1h). The stability analyses can now be compared directly against the results from the original threshold that are reported in six additional new supplementary Tables (Supplementary files 1b-1g), revealing that the same nodes are significant at p-FDR <0.05, when assessed using an adjacency matrix threshold for the network edges based on cost. The results in all of these new tables are reported at corrected (p-FDR <0.05) and uncorrected (p-uncorrected <0.05) thresholds for completeness. The subsection “Stability of effects” has been updated to point to the availability of these data.

6) CA3 damage does not affect network structure outside of the default mode: The authors report that no differences were found across any of their graph metrics when looking at resting state networks beyond the default mode network. No statistical measures or results are provided to accompany this statement. This is also the case for non-significant metrics and ROIs from the DMN. It would be beneficial to include these details (e.g., p-FDR and beta values, perhaps as a table in the Supplementary Materials).

We apologise for this oversight. As noted above, we now report complete results from the original a priori threshold in six additional new supplementary Tables for the DN and five control resting-state networks as a function of all of the graph theoretic measures that were tested (Supplementary file 1b-1g). The results in these new tables are reported at corrected (p-FDR<0.05) and uncorrected (p-uncorrected <0.05) thresholds for completeness and can be compared against the companion results obtained from the stability analyses (Supplementary file 1h-1m).

7) Implications of increased local efficiency in amnesics:- Interestingly, amnesics showed increased local efficiency in a few ROIs (posterior cingulate, parahippocampal cortex, and retrosplenial cortex). In other words, amnesics show more robust local functional network structure, surrounding these specific ROIs, than healthy controls. The results regarding path length are intuitive and are discussed, but the implications for greater local efficiency in the amnesic group warrants some discussion.

We have endeavoured to provide additional discussion about results for the difference in local efficiency, as part of the more general requested revisions to our descriptions and discussion of the results involving graph theory based measures (Discussion section):

“Local efficiency of the left posterior cingulate cortex (midline core), left parahippocampal cortex, and right retrosplenial cortex nodes were increased in the amnesic group compared to the control group, which suggests greater capacity to process information – the result at the left posterior cingulate cortex was not robust. Modulation of activity in these regions is associated with the regulation of learning, consolidation, and retrieval. The hippocampus and posterior cingulate cortex are anatomically connected to one another (Daselaar et al., 2008; Kobayashi and Amaral, 2003), as part of a core retrieval network (Rugg and Vilberg, 2013), and damage to the retrosplenial cortex region of the posterior cingulate cortex can lead to similar deficits to those associated with MTL damage (Philippi et al., 2015; Valenstein et al., 1987). Increases in local efficiency may alter connectivity with immediately surrounding regions that support the regulation of learning, consolidation, and retrieval, and reflect adaptive functional reorganization, possibly involving the reassignment of nodal roles. In model organisms, the inhibition of rodent CA1 can lead to selective compensatory changes in anterior cingulate cortex activity that are associated with remote contextual memory (Goshen et al., 2011). Notably, however, the local efficiency differences within the three affected regions were not associated with the retrieval of episodic detail on the AI. Increases in local efficiency between 85%-270% have been observed in structural connectivity studies of left temporal lobe epilepsy, and attributed to a compensatory, perhaps maladaptive, mechanism that maintains connectivity despite the loss of connections in hub regions (DeSalvo et al., 2014). For the most part, though, the relationship between functional and structural graph theoretic based connectivity measures is not yet established.”

- In general, we thought that more clarity could be provided with respect to the graph theoretic analyses. For example, the sentence in subsection “Integration within MTL subsystem was reduced by damage to the human CA3 network” describing the DVs is very hard to follow ("In particular, the following widely used graph metrics were estimated for each of 20 core nodes (ROIs) that comprised the two subsystems and mainline core: global efficiency (i.e., integrative information transfer); average path length (inversely related to the efficiency of information exchange over a network) and clustering coefficient (a.k.a., transitivity) were used to examine network integration and the functional specificity of regional brain areas, respectively whereas, local efficiency, degree (number of connections maintained by a node), and betweenness centrality (number of short communication paths that a node is a member) were computed as regional measures."). I suggest stating more directly what each graph metric was interpreted to reflect.

In line with the comments, we have extensively revised the descriptions of the graph theoretic measures applied to examine between-group differences in network topology before reporting the result from the analyses (subsection "Integration within MTL subsystem was reduced by damage to the human CA3 network”):

“The rs-fMRI data were interrogated to describe topological organisation of the two subsystems and mainline core of the DN, defined by parcellation scheme proposed by Andrews-Hanna et al. (2010). Network topological properties (i.e., the arrangement of nodes and edges) of the nodes that comprised the two subsystems and mainline core were examined by computing each graph theoretic measure for each network node (8-mm spherical regions-of-interest and their pairwise edges). In line with prior studies, we computed measures of functional integration (average path length and global efficiency), measures of functional segregation (clustering coefficient and local efficiency), and local measures that consider the centrality of nodes (degree and betweenness centrality). Functional integration examined the capacity of nodes within a network to combine information from distributed regions, whereas the measures of functional segregation were a proxy of the capacity for specialised processing within densely interconnected groups of regions. It is important to consider graph theoretic measures together, since, for example, an increase in global efficiency accompanied by a reduction in clustering coefficient could reflect an imbalance between functional integration and segregation.

Average path length expresses the average value of the shortest path lengths in a graph and is inversely related to the (integrative) global efficiency of information exchange over a network. A smaller path length thus represents greater integration. Removal of connections in functional hub regions reduces global efficiency (Hwang et al., 2013), reflecting a loss of network integration (i.e., efficient communication). Clustering coefficient (a.k.a., transitivity) estimates the extent to which connectivity is clustered around a node, independently of its membership of a particular module, and reflects the functional specificity of regional brain areas. Degree (the number of edges maintained by a node) and betweenness centrality (the number of short communication paths that a node is a member) were computed as local measures. For a detailed interpretation of these graph theoretic measures, see (Bullmore and Bassett, 2011) and (Rubinov and Sporns, 2010).”

The relevant section of the Materials and methods section has also been revised to align with the Results section:

“Tools for measuring network properties included in the CONN toolbox were used for the construction of graphs (i.e., nodes and edges (the functional connections)), their description, and the mathematical formula of each measure. Three classes of properties were computed for each participant: two measures of functional integration (average path length and global efficiency), two measures of functional segregation (clustering coefficient and local efficiency), and two centrality measures (degree and betweenness). In the final step, we explored the relationship between the graph theoretic measures that were significantly different between the amnesic and control groups and behavioural performance on the AI.”

But more importantly, how many metrics/DVs were analysed for each node? Is there a problem of multiple comparisons here?

Although, for example, grey matter volume loss may be predicted to influence average path length, the lack of specific precedents in the extant literature for examining the impact of highly focal hippocampal lesions on resting-state networks underscored the reasons we elected not to restrict the analyses to a subset of graph theoretic measures. In particular, of the four or so studies in the current literature that have examined the effects of various forms episodic amnesia associated with MTL and/or larger scale hippocampal damage on resting-state networks, only one has conducted graph theoretic analyses. More generally, it has not yet been established which graph theoretic measures are most appropriate for the analysis of brain networks. On the basis of these considerations, we systematically tested three classes of graph theoretic measures for all nodes given the precedents in the literature for assessing several graphtheoretic measures that are sensitive to three key topological properties: functional integration (average path length and global efficiency), functional segregation (clustering coefficient and local efficiency), and local measures that consider the centrality of nodes (degree and betweenness).

We applied the approach used in previous studies where no correction for multiple comparisons across the graph theoretic measures of network properties (Cao et al., 2014; Rubinov and Sporns, 2010, 2011). Indeed, different topological metrics can exhibit a non-trivial mechanistic correlation related to the way they are mathematically defined (Joyce et al., 2010; Costa et al., 2007; Lynall et al., 2010; Ekman et al., 2012). Thus, we also tested for correlations in the node that overlapped for average path length and local efficiency differences when drawing inferences from the results of conducting analyses of the non-independent graph measures and have reported this result (Page 16). Crucially, and as noted in the manuscript, FDR-based correction for multiple comparisons were applied when testing the ROIs/nodes within the networks (Schröter et al., 2012; Bassett et al., 2012; Lynall et al., 2010). More generally, it is important to consider graph theoretic measures together, since for example, an increase in global efficiency accompanied by a reduction in clustering coefficient could, for example, reflect an imbalance between functional integration and segregation.

8) While we appreciated the care taken to characterize their patient sample, a critical claim of the current paper is that this pattern of results is due to selective CA3 lesions, so the bar is very high. To this end, we have a few comments and requests for clarification:- We would like to see a graph that shows individual participant data (patients and controls) for the individual subfields. Is it the case that one amnesic participant does not have appreciable CA3 volume loss? Note that the y-axis on Figure 2 is obscured by the label, adding to confusion. Does this individual have altered functional connectivity? Does this individual show signs of memory impairment? We assume that repeating the analyses without this patient does not change the results? If we are misunderstanding the axis, please clarify.

A graph with individual participant data for each subfield is now included with the Supplementary files section of the manuscript (Figure 2—figure supplement 1). Reference to this new graph is made in the relevant subsection of the results and in the legends associated with Figure 2 and Table 1. Figure 2 has been revised so that the y-axis is no longer obscured by the axis label.

With respect to the participant in the amnesic group with a total CA3 volume that was in excess of the control group mean, we note that this participant met the inclusion criteria for the study: positive for the LGI1 antibody, an antibody titre that was in excess of 400 pmol/l, exhibited characteristic hippocampal hyperintensities at the *acute* phase on T2 clinical neuroimaging that over time resulted in permanent hippocampal volume loss, a clinical presentation consistent with the onset of LGI1-antibody-complex LE, received a formal clinical diagnosis of LGI1-antibody-complex LE, underwent clinical treatment for the symptoms associated LGI1-antibody-complex LE, and in the chronic phase of the disease at the time of data acquisition for the study. As a consequence, the participant was recruited into the study to conduct clinical, behavioural, and neuroimaging assessments, and we thereby contend that there is no a priori basis on which to exclude the data obtained from this participant.

Crucially, we also note that (1) the participant exhibited *CA3 volume loss* (-18%) when considered against the matched control participant. This magnitude of the volume loss is within the range of CA3 volume loss for the other participants; (2) the participant exhibited memory impairment on the AI that was characteristic of the amnesic group participants; in particular, internal detail was 52.9% *below* control group mean and 62% *below* the internal detail remembered by the matched control participant; and (3) the participant exhibited altered functional connectivity. A re-analysis of the results from the AI is now reported in the revised Supplementary Materials/re-named as, Appendix 1 under a new subsection entitled, “Subgroup analyses”. In brief, the results of the re-analyses of data obtained from the Autobiographical Interview did not change when the participant was excluded. Likewise, for the results from 3-D manual volumetry.

Thus, in summary, we contend that since this participant exhibits CA3 volume loss relative to the matched participant, has memory and functional connectivity deficits, and, crucially, met all a priori inclusion criteria for the study, it is appropriate to include data obtained from this participant in the main study.

- The neuropsychological results, as presented in Figure 6 and Supplementary file 1, indicate that the patients are indistinguishable – or superior – than their controls. A more fulsome commentary in the text on this issue seems appropriate, because at first glance, the patients don't seem to be particularly amnesic. This might be a good opportunity to highlight the fact that the patients were impaired on the assessments of delayed verbal recall. It would be helpful to readers, and to future researchers, if the authors could provide some additional detail regarding the extent and quality of amnesia associated with this patient group. For example, including some sample transcripts that are representative of their recall, relative to the age-matched controls, would be very informative.

In order to elaborate on the detail of the specific pathology associated with LGI1-antibody-complex LE, we have (1) cited other studies that have examined the neuropsychological profile of individuals at the chronic phase of post-onset LGI1-complex-antibody LE to contextualise our results, “Prior studies have shown that the chronic phase of LGI1-antibody-complex LE is associated with persistent memory deficits (Bettcher et al., 2014; Butler et al., 2014; Malter et al., 2014), particularly verbal memory (Finke et al., 2017)” (subsection "Other cognitive functions: general neuropsychological assessment”); (2) added information explaining how the participants in the amnesic group did not confabulate; (3) added performance on delayed visual recall (*z* = -0.08, s.e.m. = 0.20, *t*_(15)_ = 0.41, p=0.685); and, (4) to improve contextualisation of the current results within the wider literature, added a paragraph summarising the neuropsychological profile of participants included in small group studies on amnesia that have (a) reported results from rs-fcMRI data and (b) focal damage to CA1. We also draw attention to the deficits on delayed verbal recall in the Figure 6 and Supplementary file 1a legends and preserved delayed visual recall.

We thank the reviewers for the suggestion and agree that representative excerpts from the transcripts would be informative. However, the terms of Local Research Ethics approval and Oxford Research Hospitals Research and Development approval do not permit these individual primary data to be shared even in anonymised form. These transcripts are uniquely identifying of the individual participants, with the potential for identification by the original participant, their family, friends, and colleagues. In addition, we are legally precluded from publicly sharing the requested transcripts outside of the immediate research group. Therefore, we unable to make the data available in order to comply with the restrictions of the local research ethics committee and the terms of consent signed by the human participants.

We note that two independent raters scored 100% of the episodes acquired from each time period for all participants in the amnesic and control groups, which exceeds the level of second scoring reported in many previous studies that have administered the AI. More generally, we also note that in addition to the rarity of the selective bilateral subfield damage and uncommon size of the amnesic group, the study is atypical in the amnesia literature since all participants were systematically characterised in terms of anatomical damage at 7.0-T, resting-state functional connectivity, performance on the AI, and on multiple cognitive domains of recall and recognition memory, attention, executive function, language, visuomotor, and visuoconstruction.

- In subsection “Quantitative hippocampal subfield morphometry” the authors indicate that "Results from one participant in the amnesic group was not available because it was not possible to segment the five hippocampal subfield across the entire longitudinal axes of both hippocampi, but the participant exhibited bilateral hippocampal volume loss compared to the control group mean." Additional details regarding why this was the case (e.g., issue with segmentation software?) would be helpful, and further information is necessary to convincingly argue that this participant has selective CA3 damage and thus warrants inclusion in the current sample.

We have revised the main text as follows to state the reason why we were unable to segment the hippocampal volumes of this participant, “Results from one participant in the amnesic group were not available because it was not possible to segment the five hippocampal subfield across the entire longitudinal axes of both hippocampi, due to insufficient contrast for delineation between the subfield boundaries on each coronal slice. The same participant exhibited bilateral hippocampal volume loss compared to the control group mean and met all a priori inclusion criteria for participation in the study. Re-examination of the results from the AI revealed that the findings held when this participant with amnesia was removed from the analyses (see Appendix 1).” (subsection "Quantitative hippocampal subfield morphometry”).

In the same way as all other participants in the amnesic group, the participant was positive for the LGI1 antibody, with an antibody titre that was in excess of 400 pmol/l, exhibited characteristic hippocampal hyperintensities at the *acute* phase on T2 clinical neuroimaging that over time resulted in permanent hippocampal volume loss, exhibited a clinical presentation that was consistent with the onset of LGI1-antibody-complex LE, received a formal clinical diagnosis of LGI1-antibody-complex LE, underwent clinical treatment for the symptoms associated LGI1-antibody-complex LE, and was stable, in the chronic phase of the disease at the time of data acquisition for the study. As a consequence, the participant was recruited into the study in order to conduct clinical, behavioural, and neuroimaging assessments, and we would thus contend that there is no a priori basis on which to exclude the data obtained from this participant.

It is also not unreasonable to predict selective deficits associated with LGI1-antibody-complex LE in this participant – we were unfortunately unable to obtain sufficient contrast despite trying with a repeat of 3-D FSE scan. Selective CA3 damage is consistent with anatomical localization of LGI1 gene transcripts enrichment in CA3 of the adult human brain (Hawrylycz et al., 2012) and in DG and CA3 in the mouse (Herranz-Perez et al., 2010), and evidence of greater neuronal loss in CA3 than CA1 following seizures in homozygous LGI1 knockout mice (Chabrol et al., 2010). Perhaps most compellingly, in our larger cohort of 18 participants with LGI1-antibody-complex LE (Miller et al., 2017), we found that all participants with the same laboratory and clinical characteristics as the participant with hippocampal damage exhibited bilateral volume loss involving CA3. Furthermore, like the other participants in the amnesic group, no other lesions were detected in this participant.

Our study is atypical in restricting an amnesic group cohort to a single aetiology and characterising the anatomical pathology at 7.0-Tesla, so we feel that it is important to include all of the participants who met the study inclusion criteria. Indeed, unlike several prior studies, we obtained contemporaneous ultra-high field neuroimaging that still enabled us to determine that the anatomical damage was selective to the hippocampus, bilateral, and within a range of volume loss exhibited by the participants in whom were able to conduct hippocampal subfield segmentation. In terms of memory performance, the participant exhibited selective episodic impairment on the AI and the profile of neuropsychological performance was comparable to other participants included in the amnesic group.

In summary, we believe that inclusion of this participant is informative because the damage was selective to the hippocampus and was likely confined to CA3. We have also reviewed the AI results and demonstrate that this participant did not drive the effects observed in the data. For transparency, we have reported this re-analysis of the AI data to demonstrate that the conclusions are unaffected by the inclusion of this participant. The new results from these analyses are included in Appendix 1 subsection “Subgroup analyses”.

9) We provide the more general suggestion that the authors present the reader with an overarching analysis plan at the outset of the Results section. I believe that this would help to mitigate the impression that the various tests were not motivated or coherently linked.

Thank you for this constructive suggestion. We have revised the Results section to incorporate a précis of the analysis plan. As requested, the précis appears at the outset of the Results section and is structured to align with the later subsections of the Results section:

“All data were collected at the chronic stage of LGI1-antibody-complex LE (median time between symptoms onset and study examination (median=4 years post-onset, range=7), suggesting that the outcomes were unlikely to be mediated by short‐term compensatory processes. Clinical and laboratory characteristics, neuropsychological assessment, and quantitative measures of damage based on anatomical 7.0-Tesla MRI data have been previously published (Miller et al., 2017). New data on autobiographical memory for remote events and functional connectivity based assessments of resting-state fMRI data are reported here.

In brief, we first report autobiographical memory performance by examining the data obtained from administering the AI to participants in the amnesic and control groups. Differences in the retrieval of internal detail over time between the amnesic and control group participants were examined to assess (a) if there was a loss of internal detail in the amnesic group and (b) if present, how the loss changed over time relative to the control group (Figure 4 and Figure 5). Second, we report the results from standardised neuropsychological tests administered to the amnesic group in order to assess intelligence, attention, executive function, language, visuomotor skills, visuoconstructive skills, verbal memory, visual memory, and recognition memory (Figure 6 and Supplementary file 1a). Third, anatomical MRI data acquired at 7.0-Tesla field strength were used to assess (a) which hippocampal subfield volumes were affected in the amnesic group relative to the control group (Figure 1, Figure 2 and Figure 2—figure supplement 1 and Table 1) and (b) whole-brain grey matter volume in the amnesic group relative to control group (Figure 3). Fourth, we characterised the impact of CA3 damage on functional connectivity of the DN by testing for between-group differences in the topological properties of DN nodes defined by the Andrews-Hanna et al. (2010) parcellation scheme. Specifically, the graph theoretic based measures were applied to investigate functional integration, functional segregation, and local measures of centrality (Figure 7 and Figure 8). Fifth, in order to determine the scalar extent of altered topology, the same graph theoretical based measures were applied to examine the topological properties of five other large-scale brain networks (somatomotor network, visual network, dorsal attention network, ventral attention network, and salience network). Sixth, functional connectivity was additionally assessed using seed-based analyses involving a left hippocampal seed region and seeds in the right hippocampus, the occipital pole, and primary motor cortex. Finally, the relevance of differences in CA3 volume and topological properties of the affected nodes for autobiographical memory performance were assessed using robust multiple regression based analyses. Inferences were two-sided at an α level of 0.05, with correction for multiple comparisons.”

[Editors' note: further revisions were requested prior to acceptance, as described below.]Thank you for resubmitting your work entitled "Human hippocampal CA3 damage disrupts both recent and remote episodic memories" for further consideration at eLife. Your revised article has been favorably evaluated by Laura Colgin (Senior Editor), a Reviewing Editor, and two reviewers.The manuscript has been improved but there are some remaining issues that need to be addressed before acceptance, as outlined below.We appreciated the thoroughness of the revisions and believe that the overall clarity of the manuscript has significantly improved. We continue to be impressed by the careful characterization of this large and rare patient population, as well as the rigorous approach taken in characterizing the autobiographical memories. However, the new analyses raised some additional concerns, which we describe below (note that numbering corresponds to the original review synthesis statement).

We are grateful to the Senior Editor, Reviewing Editor, and reviewers for the favourable evaluation of our first round revisions and for the additional insightful comments. Further revisions have been made to the main text and supplementary materials, informed by these remaining comments. We hope that our point-by-point responses, revisions, new analyses, and the inclusion of additional figures in the manuscript have addressed the remaining issues.

1) Our primary concern is with respect to excluding the 0-11 bin. In the original submission this time bin was critical to the main conclusions of the paper, yet in this subsequent revision the authors argue that this time bin should be dropped. For starters, we believe that it is essential that the exclusion of the earliest time bin is clearly described as post hoc and some of the stronger claims be tempered. Visual inspection of Figure 4 suggests that there is a time-sensitive deficit (i.e., the opposite of what the paper argues), and the significant interaction in the new 2 x 5 ANOVA supports this claim. As such, the primary conclusion of the paper hinges on a null interaction from a post hoc 2 x 4 ANOVA that had reduced power due to the dropped time bin.

Thank you for identifying the need to elaborate on these issues. We did not intend to suggest that the earliest remote memory should be dropped from consideration; instead, we sought to understand the profile of time-sensitive loss, in terms of the duration of ungraded loss of internal detail and the inflection point when the between-group difference is null. These features of the time-sensitive loss of internal detail cannot be addressed by limiting the analyses to the three-way (identified in the omnibus ANOVA) and post hoc two-way interaction terms (observed for the 2 x 5 subsidiary/follow-up ANOVAs on internal and external detail). We agree this was not communicated adequately in our initial revisions and have thus extensively revised the text accordingly.

Visual inspection of Figure 4 does indeed suggests that there is a time-sensitive deficit, and the significant interaction found in the 2 x 5 ANOVA analysis (requested in the first round) supports this interpretation. We noted this in our earlier revision to the text, “Crucially, these results suggest that the loss of internal (episodic) detail in the amnesic group relative to the control group decreased with the age of recalled memory, when all five sampled intervals are considered.” (subsection "Autobiographical memory: amnesia for recent and remote episodic detail”) and in the text associated with Figure 4, “Internal (episodic) detail loss in the amnesic group decreased with the age of recalled memory, when all five sampled intervals were considered *(F*_(2.67,80.22)_ = 3.91, p=0.015, η^2^_p_ = 0.115).” This phrasing did not, however, explain that the “decrease” was confined to the earliest remote memory and otherwise ungraded across the remaining remote and recent memories. Accordingly, the text has now been revised to state explicitly that the deficit was time-sensitive and explain the specific features of the deficit over time (please see below for details).

In particular, visual inspection of Figure 4 also suggests that the null group difference in internal detail is confined to the earliest remote memory, which was confirmed by a post hoc direct group comparison (*F*_(1,30)_ = 0.25, p=0.621); the loss of internal detail at all other intervals was, in fact, ungraded over the remaining sampled intervals (group: *F*_(1,30)_ = 23.25, p<0.0001, η^2^_p_ = 0.437; group x time interaction: *F*_(2.62,78.44)_ = 1.51, p=0.222, η^2^_p_ = 0.048). We now also report the results from the requested between-group contrast for the 11-18 interval on internal detail (*F*_(1,30)_ = 6.428, p=0.017), confirming that the loss of internal detail persisted up to ~50 years prior to CA3 damage.

We have now revised the Results section and Discussion section to state explicitly that (1) the loss of internal detail was time-sensitive and (2) state the post hoc nature of the analyses that examined the significant three-way interaction observed in omnibus ANOVA (i.e., group, time, and detail type (*F*_(3.50,105.11)_ = 2.83, p=0.034, η^2^_p_ = 0.086; see Appendix 1 for full three-way ANOVA results) reported at the outset of the results on the AI, which included all of the five intervals. Revisions to the Results section are indicated immediately below. Other revisions are noted later.

- “To preface the main results, a significant interaction between group and time (across all five sampled intervals) indicates that the loss of internal (episodic) detail loss was time-sensitive (*F*_(2.67,80.22)_ = 3.91, p=0.015, η^2^_p_ = 0.115). Post hoc analyses revealed that the earliest remote memory (0-11 years) was intact (*F*_(1,30)_ = 0.250, p=0.621), whereas there was a loss of internal detail across the remaining remote and recent memories that did not change over time (group: *F*_(1,30)_ = 23.25, p<0.0001, η^2^_p_ = 0.437; group x time: *F*_(2.62,78.44)_ = 1.15, p=0.222, η^2^_p_ = 0.048; time: *F*_(2.62,78.44)_ = 0.604, p=0.592, η^2^_p_ = 0.020), and extended up to ~50 years prior to the CA3 damage (11-18 year interval for internal detail, *F*_(1,30)_ = 6.43, p=0.017) (Figure 4).” (subsection "Autobiographical memory: amnesia for recent and remote episodic detail”).

- “In order to explore the between-group differences in internal (episodic) detail as a function of the age of the memory, we conducted a post hoc 2 (group: amnesic, control) x 5 (time: past year (i.e., anterograde interval); 30-55, 18-30, 11-18, and 0-11 years (i.e., retrograde intervals) mixed-model ANOVA on the cumulative internal detail scores. Mauchly’s test of sphericity was significant for time (χ^2^_(9)_ = 31.84, p<0.0001). Degrees of freedom were corrected using Greenhouse-Geisser estimates (ɛ=0.669). In addition to a significant main effect of group (*F*_(1,30)_ = 16.37, p<0.0001, η^2^_p_ = 0.353), there was a significant two-way interaction between group and time (*F*_(2.67,80.22)_ = 3.91, p=0.015, η^2^_p_ = 0.115). The main effect of time was not significant (*F*_(2.67,80.22)_ = 1.13, p=0.337, η^2^_p_ = 0.036). Crucially, these results suggest that the loss of internal (episodic) detail in the amnesic group relative to the control group changed across the five sampled intervals.” (subsection "Autobiographical memory: amnesia for recent and remote episodic detail”).

- “Visual inspection of Figure 4 suggests a null difference in internal (episodic) detail generation for earliest remote memory (i.e., 0-11 year interval). A post hoc direct group comparison revealed a null difference at the 0-11 year interval (*F*_(1, 30)_ = 0.25, p=0.621).” (subsection "Autobiographical memory: amnesia for recent and remote episodic detail”).

- “Figure 4 also points to loss of internal (episodic) detail at all of the other sampled intervals (i.e., the last year, 30-55, 18-30, and 11-18). To assess profile of retrograde amnesia across these intervals, we conducted a post hoc 2 (group: amnesic, control) x 4 (time: past year; 30-55, 18-30, and 11-18 years) mixed-model ANOVA on the cumulative internal detail scores. Mauchly’s test of sphericity was significant for time (χ^2^_(5)_ = 11.93, p=0.036), so degrees of freedom were corrected using Huynh-Feldt estimates (ɛ = 0.872). The loss of internal detail was evident in the main effect of group (*F*_(1,30)_ = 23.25, p<0.0001, η^2^_p_ = 0.437), whereas the two-way interaction between group and time and main effect of time were not significant (*F*_(2.62,78.44)_ = 1.51, p=0.222, η^2^_p_ = 0.048; *F*_(2.62, 78.44)_ = 0.604, p=0.592, η^2^_p_ = 0.020, respectively). Hence, the post hoc analysis revealed that the loss of internal detail in the amnesic group did not change across the recent and remote memories (i.e., ungraded), spanning up to ~50 years prior to CA3 damage (11-18 year interval for internal detail, *F*_(1,30)_=6.43, *p*=0.017), when the loss was assessed without the earliest, intact remote memory. Together, these post hoc analyses revealed that the loss of internal detail was time-sensitive: retrograde amnesia was ungraded across recent and remote memories and spanned up to ~50 years prior to CA3 damage, whereas the earliest remote memory was intact.” (subsection "Autobiographical memory: amnesia for recent and remote episodic detail”).

The legend associated with Figure 4 has been revised:

- “A significant interaction between group and time (across all five sampled intervals) suggests that the loss of internal (episodic) detail loss was time-sensitive (*F*_(2.67,80.22)_ = 3.91, p=0.015, η^2^_p_ = 0.115). Post hoc analyses revealed that the earliest remote memory (0-11 years) was intact (*F*_(1, 30)_ = 0.250, p=0.621), whereas there was ungraded loss of internal detail across the remaining remote and recent memories (group: *F*_(1, 30)_ = 23.25, p<0.0001, η^2^_p_=0.437; group x time: *F*_(2.62, 78.44)_ = 1.51, p=0.222, η^2^_p_ = 0.048; time: *F*_(2.62, 78.44)_ = 0.604, p=0.592, η^2^_p_ = 0.020), extending up to ~50 years prior to the CA3 damage (11-18 year interval for internal detail, *F*_(1,30)_ = 6.43, p=0.017).”

At the outset of the Discussion section, the main text has been revised to reiterate the time-sensitive loss of internal detail:

“Results from participants with damage to CA3 revealed a time-sensitive loss of internal (episodic) detail in the amnesic group when their performance was compared against the control group: the earliest remote memory (0-11 years) was intact, whereas all other remote memories and memory for an event from the last year exhibited a loss of internal (episodic) detail, yet the personal semantic content and narrative structure of these same memories were comparable to the control group.” (subsection "Seed-to-voxel and ROI-to-ROI functional connectivity analyses”).

Crucially, our post hoc analysis strategy is designed to reduce the number of follow-up contrasts required to assess the 2 x 2 x 5 and 2 x 5 interaction terms observed in our data and enabled us to measure the duration of ungraded amnesia (by testing the interaction term over intervals where visual inspection indicates a deficit). To underscore that the results from post hoc comparison of the earliest remote memory and 2 x 4 ANOVA on internal detail are representative of (1) when the inflection point occurs and (2) the duration of loss, we note that, alternatively, five between-group pairwise contrasts on the 0-11, 11-18, 18-30, 30-55 intervals, and the past year revealed significant retrograde amnesia at the 11-18, 18-30, 30-55 intervals, and the past year and a null group difference at the 0-11 interval. Furthermore, the significant loss of episodic detail observed at 11-18, 18-30, and 30-55, and past year survived Bonferroni-Holm correction for multiple comparisons. Notably, in the manuscript, we mainly focus on the duration of the observed deficit in episodic detail (please see illustrative excerpts taken from the manuscript below).

Importantly, the loss of internal detail was not described as time insensitive in our revisions. In our revisions and in the original manuscript, we instead refer to a loss of internal detail in recent and remote memory, with qualification that it persisted up to ~50 years prior to the CA3 damage (as noted, the latter is supported/confirmed by the new analysis that reveals a significant loss in internal detail at the 11-18 years interval as well as by the results from the 2 x 4 ANOVA, please see below). For example, in the abstract we stated, “Here, we demonstrate that recent and remote memories were susceptible to a loss of episodic detail in human participants with focal bilateral damage to CA3.” Later, we stated, “We conclude that human CA3 is necessary for episodic memory retrieval long after initial acquisition.”

We also believe our intended aim and hypothesis for the behavioural data in the introduction were measured given the complex profiles of retrograde amnesia in the existing literature: “Here, we tested the contribution of human hippocampal area CA3 to the retrieval of remote and recent autobiographical memories” … “we hypothesised that CA3 damage would lead to a loss of internal details for both remote and recent memories”, motivated, in part, by evidence from the experimental animal literature that shows, “Lesion and molecular imaging studies in rodents indicate that CA1 and CA3 enable the rapid storage and retrieval of recent (<1 month) contextual fear memories, whereas remote memories depend on CA1 but not CA3 (Denny et al., 2014; Guzman et al., 2016; Kesner and Rolls, 2015; Leutgeb et al., 2007; Lisman, 1999; Lux et al., 2016; McNaughton and Morris, 1987; Rebola et al., 2017)”.

In the Discussion section we stated, “… The loss of episodic detail for remote and recent memories, for all but the earliest remote memory … The duration of amnesia observed here – spanning up to ~50 years prior to the CA3 damage – … Human CA3 was necessary for the retrieval of internal (episodic) but not external (non-episodic, semantic) detail related to both recent and remote memories… Contrary to the duration of hippocampal involvement predicted by systems consolidation and computational based accounts of episodic memory (Rolls et al., 1997; Squire and Bayley, 2007), amnesia was observed for episodes that occurred up to ~50 years prior to focal CA3 damage.” Hence, as in other subsections, we do not explicitly state that the loss of internal episodic detail was time insensitive, merely that there was a loss of internal detail for recent and remote intervals and that the earliest remote memory was intact.

In summary, we have now extensively revised the manuscript to state explicitly that the loss of internal detail was, indeed, time-sensitive and explain the temporal profile of the loss, in terms of the duration of ungraded loss and the inflection point at which the null group difference emerged. As is convention in the literature, pinpointing where the significant differences reside when an interaction term is observed required follow-up pairwise contrasts and analysis. We have also shown here that the pattern of time-sensitive loss is robust to different post hoc analysis decisions. The significant loss of internal detail at the 11-18 interval revealed by the direct group contrast, together with the null difference observed at the 0-11 interval, demonstrate that deficits in episodic memory persisted up to ~50 years prior to the CA3 damage, which as per the comment below, “offers strong evidence against systems consolidation.”

These concerns aside, we do agree that there is a valid argument that these oldest memories might have been schematized to a state that renders them qualitatively different from the more recent (yet still old) memories. Is there evidence for this 'extreme schematization' in the current data? For example, do controls show an elevated external:internal ratio for this time bin? Any such analysis would be obviously post hoc and should be described as such, but we believe that clearer justification for this analytic decision would strengthen the resulting claims, particularly if your own data show that the oldest memories are indeed more schematized than the others.

Our comments about semanticisation were intended to provide a candidate post hoc explanation as to why only the earliest remote memory was intact in the amnesic group. As such, these comments are independent of the post hoc analyses decisions designed to follow-up on the two- and three-way interaction terms and assess how long the deficit in internal detail persisted and the duration of ungraded loss of internal details, with the minimal number of post hoc contrasts. In addition, the analytic decisions related to following up the interaction terms are compatible with the request below for a further direct group contrast at the 11-18 year interval, which provides an important data point that is not compatible with systems consolidation, computational accounts, and the role of CA3 suggested in the experimental animal literature. We apologise that this was not communicated adequately.

We have examined the AI data in order to obtain values for the suggested external detail:internal detail ratio. The results do not suggest evidence of extreme semanticisation, when examining the change in external:internal ratio in the control group over time (*F*_(4,60)_=1.04, p=0.396, η^2^_p_=0.065). Results from the requested post hoc analyses of semanticisation are now reported in the main text and we have formalised the assessment of semanticisation as a candidate post hoc ‘cognitive/behavioural’ explanation to explain the null difference in the earliest remote memory:

“Visual inspection of Figure 4 suggests a null difference in internal (episodic) detail generation for earliest remote memory (i.e., 0-11 year interval). A post hoc direct group comparison revealed a null difference at the 0-11 year interval (*F*_(1,30)_ = 0.25, p=0.621). Early memories have been described as gist-like (Hardt et al., 2013; Richards and Frankland, 2017; Sadeh et al., 2014), and are arguably qualitatively different from other remote memories along several dimensions (Barclay and Wellman, 1986; Cermak, 1984; Sekeres et al., 2018; Winocur and Moscovitch, 2011), lose contextual specificity over time, and can be supported by extra-hippocampal regions such as medial prefrontal cortex (Clewett et al., 2019; Wiltgen et al., 2010; Winocur et al., 2010, 2007). By inference, the earliest, intact remote memory may not be hippocampal (CA3)-dependent. In order to assess if the earliest remote memory was qualitatively difference from the other remote memories, a post hoc one-way ANOVA with time (past year; 30-55, 18-30, 11-18, and 0-11 years) as the repeated measures variable was conducted to assess whether the ratio of external detail was elevated relative to internal detail in the control group for the earliest remote memory. Mauchly’s test of sphericity was not significant for time (χ^2^_(5)_ = 15.86, p=0.071). The main effect of time was not significant (*F*_(4,60)_ = 1.04, p=0.396, η^2^_p_ = 0.065). Hence, the earliest remote memories were not detectably schematized to a state that rendered them qualitatively different from the more recent (remote) memories, at least when assessed by examining the ratio between external and internal detail.”

We have tempered the conclusions, stating merely that CA3 does not appear to be necessary to support the retrieval of the earliest remote memories and, as suggested, by others, in the Discussion section we state, “The preservation of the earliest remote memories is consistent with the contention that these memories are remembered more frequently, re-encoded, and can be supported by neocortical representations (Sekeres et al., 2018).” This neurobiologically centred interpretation may help explain why the amnesic group generated comparable internal details for the earliest remote memory. In addition, the putative changes in the neural substrate associated with very early remote memories may not necessarily be coincidental with a change in the profile of the memory that can be detected by a change in the ratio between external and internal detail.

Secondly, the finding of reduced internal details for the 11-18 bin offers strong evidence against systems consolidation, yet the only direct comparison reported between groups was for the 0-11 bin. Is there a group difference at the 11-18 bin?

As noted in the earlier revision, the post hoc 2 (group: amnesic, control) x 4 (time: past year; 30-55, 18-30, and 11-18 years) mixed-model ANOVA on cumulative internal detail revealed a non-significant effect of time and a non-significant interaction term between group and time (*F*_(2.62,78.44)_ = 1.51, p=0.222, η^2^_p_ = 0.048; *F*_(2.62,78.44)_ = 0.604, p=0.592, η^2^_p_ = 0.020, respectively), whereas the group main effect was significant (*F*_(1,30)_ = 23.25, p<0.0001, η^2^_p_ = 0.437). In line with the significant main effect of group without a significant interaction between group and time, there was, indeed, a significant loss of internal detail in amnesic group compared to the control group for 11-18 interval (*F*_(1,30)_ = 6.428, p=0.017).

Together, these results are consistent with a loss of internal detail persisting up to ~50 years and, as suggested, do not align with predictions from systems consolidation, computational accounts, or the role of CA3 suggested in the experimental animal literature. For completeness, we have reported results of the internal detail between group contrast for the 11-18 interval to the main text:

“Hence, the post hoc analysis revealed that the loss of internal detail in the amnesic group did not change across the recent and remote memories (i.e., ungraded), spanning up to ~50 years prior to CA3 damage (11-18 year interval for internal detail, *F_(_*_1,30)_ = 6.43, p=0.017), when the loss was assessed without the earliest, intact remote memory.” (subsection "Autobiographical memory: amnesia for recent and remote episodic detail”).

4) Hippocampal functional connectivity.4.1) The new analyses report that when whole-brain voxel-level functional connectivity maps are calculated using left and right hippocampus separately, there are no differences found between patients and controls anywhere in the brain. This is quite surprising given that the patients are amnesic. We would like to see the left and right hippocampal functional connectivity maps for each group. If hippocampal connectivity is truly completely unaffected in CA3 amnesic patients, that seems like a rather interesting finding -- counter to some previous findings, as the authors note in their Discussion -- that warrants visualization as a map. It also would suggest that graph metric differences between groups arise largely from disruptions in connectivity between regions other than the hippocampus.

The requested functional connectivity maps for the left and right hippocampal seed to whole-brain voxelwise analyses have been generated for the amnesic group and control groups, and are incorporated into a new figure supplement (Figure 7—figure supplement 2. Left and right hippocampal seed-to-voxel functional connectivity, as function of group). The corresponding new results are reported in Supplementary file 1p. Results from seed-to-voxel functional connectivity based analyses depicted separately for left and right hippocampal seeds, and these new data are referenced in the main text:

“In addition, significant clusters thresholded at p-FDR <0.05 are reported separately for the left and right hippocampal seeds as a function of group in Supplementary file 1p and corresponding group-wise seed-to-voxel correlation maps are plotted in Figure 7—figure supplement 2.” (subsection "Seed-to-voxel and ROI-to-ROI functional connectivity analyses”).

“Figure 7—figure supplement 2. Left and right hippocampal seed-to-voxel functional connectivity, as function of group. Left panel: in the amnesic group, functional connectivity with the left hippocampal seed region was associated with significant activity in two clusters, whereas the right hippocampal seed was associated with significant activity in seven clusters; Right panel: in the control group, functional connectivity with the left hippocampal seed region was associated with significant activity in three clusters, whereas the right hippocampal seed was associated with significant activity in nine clusters. 3-D rendered brains depict activity at a cluster-size p-FDR threshold set at <0.05 and height threshold set at p-uncorrected <0.001 (two-sided) (see Supplementary file 1p).”

“Supplementary file 1p. Results from left and right hippocampus seed-to-voxel functional connectivity based analyses. MNI co-ordinates of brain regions that exhibited significant functional connectivity with left and right hippocampal seed regions-of-interest shown separately for the amnesic group and for the control group (height threshold, p-uncorrected <0.001 and an extent threshold of p-FDR < 0.05 at the cluster level). Seed regions were spheres with 8 mm radii (see Figure 7—figure supplement 2).” (subsection "Seed-to-voxel and ROI-to-ROI functional connectivity analyses”, Supplementary file 1).

A group difference map of left and right hippocampal connectivity would be informative as well; even though no clusters survived FDR correction, a more leniently thresholded difference map might reveal the qualitative pattern of altered hippocampal connectivity.We believe displaying these maps would add substantial value to the paper, as connectivity maps are comparable and relevant to a large number of studies in the literature. The graph metrics are all derived from the connectivity measures, so insight into the graph metrics can be obtained by examining the connectivity maps.

For the left hippocampal seed region, a small single cluster difference in functional connectivity was evident at a lenient, p-uncorrected<0.05 threshold (Figure 7—figure supplement 1. Between group hippocampal seed-to-voxel functional connectivity, at a lenient, p-uncorrected < 0.05 cluster-size threshold.). We note that, as indicated in, “Supplementary file 1o. Results from between-group seed-to-voxel functional connectivity based analyses.”, there were no significant clusters for the right hippocampal seed to whole-brain voxel functional connectivity, even when between-group differences were assessed at cluster-size threshold that was set at p-uncorrected <0.05. Therefore, we are unable to plot a map for the between group right hippocampal seed even at a lenient cluster-size threshold set at p-uncorrected<0.05.

“For completeness, we also report between-group clusters at an uncorrected cluster-size threshold (p-uncorrected >0.05; height threshold p*-*uncorrected <0.001) in Supplementary file 1o and provide a plot of the between group seed-to-voxel correlation map at an uncorrected cluster-size threshold (p-uncorrected >0.05; height threshold p*-*uncorrected < 0.001) in Figure 7—figure supplement 1.” (subsection "Seed-to-voxel and ROI-to-ROI functional connectivity analyses”).

“Figure 7—figure supplement 1. Between group hippocampal seed-to-voxel functional connectivity, at a lenient, *p*-uncorrected<0.05 cluster-size threshold. Left and right hippocampal seed regions were not associated with significant group differences in functional connectivity, when assessed at a cluster-size p-FDR set at <0.05 and height threshold set at p-uncorrected <0.001 (two-sided). Axial image depicts activity in a cluster that exhibited functional connectivity with the left hippocampal seed region at a lenient, p-uncorrected <0.05 cluster-size threshold. There were no significant clusters for the right hippocampal seed region even when between-group differences were assessed at a cluster-size threshold set at p-uncorrected<0.05 (see Supplementary file 1o).”

“Supplementary file 1o. Results from between-group seed-to-voxel functional connectivity based analyses. No brain regions exhibited significant differences (height threshold, p-uncorrected <0.001 and an extent threshold of p-FDR <0.05 at the cluster level) in functional connectivity between the amnesic group and the control group, when tested with left and right hippocampal seed regions-of-interest, an occipital pole seed within the visual network, and a seed in primary motor cortex within the somatomotor network. Seed regions were spheres with 8 mm radii. For the left hippocampal seed region, a between-group difference in functional connectivity was found only at a lenient, p-uncorrected <0.05 cluster-size threshold. There were no significant clusters for the right hippocampal seed region even when between-group differences were assessed at a cluster-size threshold set at p-uncorrected <0.05 (see Figure 7—figure supplement 1).” (subsection "Seed-to-voxel and ROI-to-ROI functional connectivity analyses”, supplementary file 1).

For example, one possible reason that path length is higher in the patient group is if fewer edges are above threshold overall in the connectivity matrix of that group (not specific to the hippocampus). How extensive are the regions exceeding the z>0.84 threshold (which is taken from another paper) in this dataset? We understand that, as the authors write in the Discussion, "seed-based analyses do not assess network connections between ROIs simultaneously, which is necessary when studying functional integration and segregation at the scale of whole-brain networks". Nonetheless, significant path length increases are reported for left and right hippocampus (Discussion section). It is helpful for the reader be able to see these basic properties of the data.

There is no difference between the two groups in the number of edges that exceeded the z>0.84 threshold. Of note, since we also conducted a full repeat assessment of the analyses obtained at *z* > 0.84 using an alternative, cost-based adjacency matrix threshold to assess the robustness of results, the results are not attributable to the choice of a particular edge estimation threshold. As noted, there was (topographic) overlap in the nodes that exhibited significant group differences when functional connectivity was assessed at a cost threshold set at 0.15 as compared to *z* > 0.84; i.e., where the strongest 15% of possible edges and edge weights in the network were retained. A cost threshold of 0.15 has been shown to yield high degree of reliability when comparing estimates of graph theoretic measures across repeated sessions or runs (Whitfield-Gabrieli and Nieto-Castanon, 2012), and is frequently applied in studies examining large-scale network topology (Bertolero et al., 2015), because it is at the centre of the ideal cost range where many graph theory metrics are maximal (Bullmore and Bassett, 2011). In addition, the altered topology was expressed on the same graph theory based measures – i.e., average path length and local efficiency – as observed in the original analysis that was based on an a priori adjacency matrix threshold set at z > 0.84 (one-sided (positive)) and an analysis threshold set at p-FDR corrected <0.05 (two-sided). Thus, the full repeat assessment objectively demonstrated that the increased average path length in the patient group was not dependent/peculiar to thresholding the connectivity matrix at *z* > 0.84.

4.2) We wish for more clarity on the logic of selecting occipital pole and primary motor cortex as "control" regions for the hippocampus-to-ROI analysis. The hippocampus tends to have low correlations with voxels in the visual cortex and somatosensory cortices in healthy controls, so why would one expect a difference between groups in those regions? It's a comparison of noise with noise, and thus uninformative.

These analyses did not test for group differences in functional connectivity between the hippocampus seeds and ROIs in the visual cortex and sensorimotor cortex; rather, as with the left and right hippocampus seeds, we assessed visual and sensorimotor seeds to whole-brain functional connectivity maps in order to reveal how damage to the CA3 affects functional coupling between the these two additional seeds and the rest of the brain. Accordingly, we have revised the main text to state more clearly that these analyses assessed seed-to-voxel functional connectivity and have also revised the main text to explain the rationale for including seed ROIs in the visual cortex and somatosensory cortex:

“Between-group differences in seed-to-voxel functional connectivity were examined by entering a left hippocampal seed region (MNI co-ordinates -24, -22, -16), based on co-ordinates implicated in episodic memory (Hirshhorn et al., 2012), because functional nodes proposed by Power et al. (2011) and Andrews-Hanna et al. (2010) do not cover the main body of the hippocampus. On the grounds that the damage to CA3 was bilateral, we elected to assess functional connectivity with the left and right hippocampus as the seed regions. Therefore, a right hemispheric homotopic region of the hippocampus was also entered into the analysis (i.e., MNI co-ordinates 24, -22, -16). In addition to the left and right hippocampal seed regions, between-group seed-to-voxel functional connectivity was computed for two other ROIs: (i) a region in the occipital pole that occurred within the visual network (MNI co-ordinates 18, -47, -10); and, (ii) a region in primary motor cortex that occurred within the somatomotor network (MNI co-ordinates -40, -19, 54). We examined these additional regions because other studies have observed that participants diagnosed with LGI1-antibody-complex LE exhibited altered functional connectivity in sensorimotor and visual networks relative to control participants (Heine et al., 2018). Thus, the visual and sensorimotor seeds originated within networks that have been associated altered functional connectivity following hippocampal damage due to the same aetiology.” (subsection "Seed-to-voxel and ROI-to-ROI functional connectivity analyses”).

It would be more useful to see hippocampus-to-ROI functional connectivity with nodes in the DN or MTL. This might be called a "control" analysis in the sense that if the whole-brain voxel-level connectivity map threshold (4.1 above) was quite stringent and that's why no clusters survived, regions in DN or MTL that usually *are* correlated with hippocampus can be tested to see whether there is still no difference between groups. Again, it is surprising (though not impossible) that there are no differences at all between groups in hippocampal functional connectivity, given that a) the hippocampus is damaged in the patient group, and b) there are differences between groups in the graph metrics based on that connectivity.

We appreciate the suggestion to conduct additional analyses. Results from the requested hippocampal ROI-to-DN ROI analyses are now provided in the main text:

“Next, for the ROI-to-ROI analyses, we tested whether there were significant group differences in functional connectivity between the hippocampus and regions in the DN that usually exhibit functional coupling with the hippocampus. Accordingly, we selected a ROI in the dmPFC, because memory-guided behaviour is supported by interactions between the hippocampus and dmPFC (Shin and Jadhav, 2016). A ROI in the vmPFC was selected because functional coupling between the hippocampus and vmPFC supports various stages of autobiographical memory processing (Barry and Maguire, 2019; Eichenbaum, 2017; McCormick et al., 2018a). Finally, we examined functional connectivity between the hippocampus and the PCC because it is a core hub of the DN, and posterior midline cortical regions such as the PCC support the successful retrieval of autobiographical memories (Addis et al., 2004a; Ryan et al., 2001; Svoboda et al., 2006). MNI co-ordinates for the dmPFC, vmPFC, left PCC, and right PCC correspond to those used in the graph theory based analyses of functional connectivity. MNI co-ordinates for the left and right hippocampus ROIs were the same as those used in the seed-to-voxel analyses. On the grounds that we were specifically interested in effects of bilateral hippocampal damage on functional connectivity, we investigated the connectivity of the left and right hippocampus with all of the other ROIs. Accordingly, temporal correlations were calculated for the left hippocampus seed with the dmPFC, vmPFC, left PCC, and right PCC ROIs and corresponding pairings were calculated for the right hippocampus seed. Normalised correlation coefficients (Fisher's z-transformation) were entered into a between-group *t*-test, assessed at two-sided p-FDR < 0.05 (seed-level correction).

Results for the left hippocampus seed ROI revealed that there were no significant group differences in functional connectivity with the left PCC (*t*_(27)_ = 0.77, p-FDR = 0.225), the right PCC (*t*_(27)_ = 0.86, p-FDR = 0.198), the dmPFC (*t*_(27)_ = 0.29, p-FDR = 0.387), and the vmPFC (*t*_(27)_ = -0.36, p-FDR = 0.637). Results for the right hippocampus seed ROI revealed that there were no significant differences in functional connectivity with the right PCC (*t*_(27)_ = 1.50, p-FDR = 0.073), the dmPFC (*t*_(27)_ = 1.08, p-FDR = 0.145), and the vmPFC (*t*_(27)_ = 1.16, p-FDR = 0.129). A significant group difference in functional connectivity was found between the right hippocampus seed ROI and the left PCC (*t*_(27)_ = 1.75, p-FDR = 0.046). Hence, the only region that exhibited a group difference in functional connectivity with the hippocampus was the left PCC, but it did not survive Holm-Bonferroni correction for multiple comparisons.” (subsection "Seed-to-voxel and ROI-to-ROI functional connectivity analyses”).

We have also added comments on these new analyses in the Discussion section:

“The ROI-to-ROI analyses revealed that functional coupling was altered only between the right hippocampus and left posterior cingulate cortex, but this did not survive correction for multiple comparisons. Alterations in connectivity between the hippocampus and posterior cingulate cortex is likely to disrupt autobiographical memory, because functional connectivity between the hippocampus and posterior cingulate cortex has been implicated in episodic autobiographical remembering (Sheldon et al., 2016; Sheldon and Levine, 2013).” (subsection "Altered topology in the default network”).

4.3) The right hippocampus is also referred to as a "control" region in the seed-based functional connectivity analysis. We found this confusing, as the patients have bilateral hippocampal damage. It is unclear exactly what analyses were done in subsection “Seed-based functional connectivity analyses”: are these correlations between (a) left hippocampus and (b) the three "control" regions (right hippocampus, visual network, and somatomotor network), or correlations between (a) the right hippocampus and (b) the visual network, and somatomotor network?

We apologise for this oversight. The text has been revised, so that the right hippocampus seed region is no longer described as a control region in the manuscript to avoid confusion; we now state that, we tested left and right hippocampal seed regions:

“Between-group differences in seed-to-voxel functional connectivity were examined by entering a left hippocampal seed region (MNI co-ordinates -24, -22, -16), based on co-ordinates implicated in episodic memory (Hirshhorn et al., 2012), because functional nodes proposed by Power et al. (2011) and Andrews-Hanna et al. (2010) do not cover the main body of the hippocampus. On the grounds that the damage to CA3 was bilateral, we elected to assess functional connectivity with the left and right hippocampus as the seed regions. Therefore, a right hemispheric homotopic region of the hippocampus was also entered into the analysis (i.e., MNI co-ordinates 24, -22, -16).”

In addition, we have extensively revised the relevant subsection in the main text to clarify the nature of the analyses that were conducted, as per the methods section. Each seed region was assessed by conducting a seed-to-whole-brain voxelwise between-group analysis for the left and right hippocampus seeds and seed regions within visual network and somatomotor network. The text has also been updated to refer to the new ROI-to-ROI analyses:

“Functional connectivity in hippocampal amnesia has also been studied using seed-based analyses of rs-fMRI, revealing that bilateral hippocampal damage in humans can alter the cortico-hippocampal network. The graph theoretic analyses provided information on how bilateral damage to human CA3 can modulate average path length and local efficiency primarily in brain regions that reside within MTL subsystem. In order to examine if the differences in left and right hippocampal average path length are associated with alterations of functional connectivity (as assessed by generating time-series correlation-strength map) with the whole brain and/or specific brain areas, we conducted post hoc seed-based functional connectivity analyses. Post hoc seed-based functional connectivity analyses were conducted in two-ways: (1) seed-to-whole-brain voxelwise analyses (henceforth referred to as, seed-to-voxel) were conducted to test for significant group differences in the correlation of left and right hippocampal seed regions with the rest of the brain (Biswal et al., 2010; Fox et al., 2005); and (2) we tested for significant between group differences in the functional connectivity of left and right hippocampus seed ROIs and ROIs in the DN (i.e., ROI-to-ROI)” (subsection "Seed-to-voxel and ROI-to-ROI functional connectivity analyses”).

8) In Figure 3 there appears to be substantial CA1 volume loss in the patients, in addition to the CA3 volume loss which is the main focus of the paper. We understand that CA1 volume was not statistically different between groups. However, given that the claims of the paper with respect to retrograde amnesia hinge on the CA1 being preserved while CA3 is selectively damaged, perhaps further discussion of CA1 volume loss is warranted. At a minimum the mean reduction% , F, p, and d should be reported, as they are for CA3.

Thank you for the suggestion. We have updated the text in the Results section to report the % loss value in the main text (derived from the volumes reported in Table 1) and the *F,* p, and Cohen’s *d* for CA1 volume loss. We have also updated the text regarding in the relevant subsection to elaborate on what is known about the pathophysiological mechanisms that underlie the acute phase of LGI1-antibody-complex LE:

“Also in line with our prior study of 18 amnesic participants with LGI1-antibody-complex LE, CA1 volume loss was not significant when the alpha criterion was corrected for multiple comparisons (mean volume loss = 16%, *F*_(1,28)_ = 5.25, p-uncorrected = 0.019, Cohen’s *d* = 0.91).

Auto-antibodies to the two principal antigenic components of voltage-gated potassium channel (VGKC)-complex—LGI1 and CASPR2 proteins—are preferentially expressed in CA3 and CA1 (Irani et al., 2010). As noted, unlike CA3, volume loss in CA1 contrast did not reach statistical significance when corrected for multiple comparisons. The observed selectivity of CA3 volume loss is consistent with anatomical localization of LGI1 gene transcripts enrichment in CA3 of the adult human brain (Hawrylycz et al., 2012), the expression of LGI1 gene transcripts in mouse CA3 (Herranz-Perez et al., 2010), and evidence of greater neuronal loss in CA3 compared to CA1 following seizures in homozygous LGI1 knockout mice (Chabrol et al., 2010). CA3 exhibits particular vulnerability versus CA1 to excitotoxic lesions associated with seizures, given that IgG containing LGI1 antibodies induce population epileptiform discharges in CA3 pyramidal neurons in vitro (Lalic et al., 2011), or from complement-mediated fixation of bound antibodies (Bien et al., 2012). No other lesions were detected in any of the amnesic group participants. Evidence of selective anatomical damage associated with LGI1 pathogenesis suggests that the chronic phase of the disease represents a compelling lesion model with which to study the causal role of human CA3 in the hippocampal network.” (subsection "Quantitative hippocampal subfield morphometry”).

The relevant text in the legend associated with Figure 2 has also been updated to state:

“Volume loss in CA1 was not significant (mean volume loss = 16%, *F*_(1,28)_ = 5.25, p*-*uncorrected = 0.019, Cohen’s *d* = 0.91) when the alpha criterion was corrected for multiple comparisons. All other subfields were not significant at the alpha criterion corrected for multiple comparisons.”